

**Title Page**

Title: Carbon balance of a restored and cutover raised bog: Comparison to global trends

Running Head: C BALANCE Of A RESTORED AND CUTOVER BOG

Authors: Michael M. Swenson[1]; Shane Regan[1]; Dirk T. H. Bremmers[1]; Jenna Lawless[1]; Matthew Saunders[2]; Laurence W. Gill[1]

Institution:

1. Department of Civil, Structural, and Environmental Engineering, Trinity College Dublin, College Green, Dublin 2, Ireland

2. Department of Botany, Trinity College Dublin, College Green, Dublin 2, Ireland

Corresponding Author: Michael Swenson, Phone: +353 892013544, Email: swensonm@tcd.ie

Key Words: carbon dioxide, peatland restoration, methane, global warming potential, bogs

Paper Type: Primary Research



**Abstract**
All major aspects of the carbon balance – net ecosystem exchange (NEE), $CH_4$ flux, losses of
dissolved organic carbon (DOC) and dissolved inorganic carbon (DIC), and open water $CO_2$ evasion
– were measured for several distinct ecotypes in a restored unharvested raised bog and an adjacent
historically abandoned cutover bog over a two year period. The average annual ecotype carbon
balance at the Sub-Central ecotype, with eco-hydrological characteristics most similar to a high
quality raised bog, was the largest net carbon sink of -32 ±65 g C $m^{-2}$ $yr^{-1}$ while the Calluna Cutover
ecotype, with the characteristics of a well-drained peatland site was the largest net carbon source
of 239 ±83 g C $m^{-2}$ $yr^{-1}$. The annual carbon balance from all ecotype study locations was found to be
controlled by mean annual water table (MAWT). Also, significant negative correlation was observed
between the plot global warming potential and percent *Sphagnum* moss cover, highlighting the
importance of regenerating this keystone genus as a climate change mitigation strategy in peatland
restoration. The data from this study was then compared to the rapidly growing number of
peatland carbon balance studies across Boreal and Temperate regions. The trend in NEE and $CH_4$
flux with respect to MAWT was compared for the five ecotypes in this study was and literature data
from degraded/restored peatlands, intact peatlands, and bare peat sites.
**1. Introduction**
Peatlands are important to the global carbon cycle as they act as significant stores of carbon (C) and
sources or sinks of carbon dioxide ($CO_2$) and methane ($CH_4$) (Gorham 1991). Despite covering only
~3% of the earth's terrestrial surface, it is estimated that between 500 and 700 billion tonnes of C
are stored as organic soil within the global peatland expanse (Yu et al., 2010; Paige and Baird, 2016;
Leifeld and Menichetti. 2018). However, at present, human activity is either draining or mining
~10% of global peatlands, transforming them from long-term C sinks into sources (Joosten, 2010;
Leifeld and Menichetti. 2018). In Europe, a high percentage (~46%) of the remaining peatlands are
degraded to the point whereby peat is no longer actively being formed (Tanneberger et al. 2017),



and in Ireland whilst ~20% of the land area is peatland, over 95% of it has been degraded through
anthropogenic activities such as drainage for agriculture, forestry and peat extraction (Connolly
and Holden, 2009; Connolly and Holden, 2017).
The carbon cycle and greenhouse gas (GHG) dynamics of degraded peatlands is often substantially
different compared to pristine peatlands (Blodau, 2002; Baird et al., 2009) making them significant
with respect to national and global GHG budgets and emission reporting (Wilson et al. 2013; Billet
et al. 2010). Moreover, degraded peatlands can continue to emit C for decades to centuries
following drainage, and current estimates are that degraded peatlands store globally ~80.8 Gt soil C
and emit ~1.91 (0.31–3.38) Gt $CO_2$-eq. $yr^{-1}$ (Leifeld and Menichetti. 2018). Soil carbon
sequestration through peatland restoration is increasingly recognized as an important strategy to
tackle climate change (Dise, 2009; Leifeld and Menichetti. 2018), and in recent years there has been
a substantial increase in money being invested in peatland projects across the world (Anderson *et*
*al.*, 2017). With the increase in global active peatland management, there is a need for more studies
examining how drainage and restoration alters the eco-hydrology of degraded peatlands systems
and their carbon balances (Baird et al., 2009; Young et al., 2017).
The land atmosphere $CO_2$ flux, or net ecosystem exchange (NEE) in peatlands is related to water
table level, as inundation creates anaerobic conditions which suppresses the decomposition of soil
organic matter (Lain et al., 1996). This can result in a net $CO_2$ sink (negative NEE) whereas a low
water table can result in a net $CO_2$ source (positive NEE). Thus, water table has been correlated to
spatial (Strack et al., 2014; Junkurst and Fielder, 2007; Silvola et al., 1996) and temporal (McVeigh
et al. 2014; Peichl et al., 2014; Lund et al., 2012; Strachan et al., 2016; Helftler et al., 2015) variation
in the NEE of both intact and degraded peatlands. However, anaerobic conditions due to a high
water table can also increase the land atmosphere $CH_4$ flux (Frenzel and Karofeld 2000). Both NEE





and $CH_4$ flux are also affected by plant ecology, as the extent of aerenchymatous vegetation cover
such as *Eriophorum spp.* is correlated with increased $CH_4$ flux (Cooper et al. 2014; Frenzel and
Karofeld 2000, Waddington and Day 2007, McNamera et al. 2008, Gray et al. 2013). *Sphagnum spp.*,
however, often exhibit lower $CH_4$ fluxes (Frenzel and Rudolph et al. 1998) due to a symbiotic
relationship with methanotrophic bacteria (Raghoebarsing et al. 2005). Also, *Sphagnum spp.*
coverage may correspond to an increase in the $CO_2$ sink function of natural sites (Strack et al. 2016)
as much of the peat in northern peatlands is derived from this genus (Vitt et al., 2000, Bacon et al.,
2017). Furthermore, the extent of vegetation cover is an important factor affecting the NEE
(Waddington and Day, 2010; Tuitili et al., 1999, Strack et al., 2016). This is relevant to degraded and
restored peatlands because harvested peatlands can have large areas of bare peat (Wilson et al.,

61    2015).


Climatic variables such as the frequency of cloudiness, temperature, and length of growing season
have also been found to be important controlling factors of NEE (Charman et al., 2013; Zhaojun et
al., 2011; McVeigh et al., 2014; Helftler et al., 2015). However, climate variables cannot be
controlled at a specific site, and therefore, may not be as relevant when considering climate change
mitigation actions.

Although $N_2O$ emissions can be an important aspect of the GHG emissions from organic soils (Pärn
et al. 2018), this study focuses only on aspects of the carbon balance. In low nutrient, semi-natural
sites like in this study, $N_2O$ emissions are typically low (Haddaway et al., 2014) but can be higher
for deeply drained (Vanselow-Algan et al., 2015) or high nutrient sites (Danevčič et al., 2010). The
radiative impact of different GHGs can be normalized by converting them into a $CO_2$ equivalents in
terms of the 100-year global warming potential (GWP) in tonnes $CO_2$-eq $ha^{-1}$ $yr^{-1}$: over a hundred
year horizon, $CO_2$ = 1, $CH_4$ = 34, and $N_2O$ = 298 (after Wilson et al., 2016b from IPCC 2014).






Intact peatlands are a net $CO_2$ sink [typical annual average NEE range -31.9 to -66 g C-$CO_2$ m$^{-2}$ yr$^{-1}$,
from Helftler et al., (2015)] and a $CH_4$ source [average of 9.2 g C-$CH_4$ m$^{-2}$ yr$^{-1}$, (95% CI 0.3 to 44.5 g C-
$CH_4$ m$^{-2}$ yr$^{-1}$) for low nutrient temperate peatlands from Wilson et al., (2016a)]. By contrast, drained
peatlands are a $CO_2$ source [the average annual NEE of +81 to +151 g C-$CO_2$ m$^{-2}$ yr$^{-1}$ reported in
Renou-Wilson et al., (2018a) is typical] with very low $CH_4$ emissions (Baird et al., 2009). However, it
should be noted that this can be offset by high methane emissions from active drains of ~60 g $CH_4$
m$^{-2}$ yr$^{-1}$ (Evans et al., 2016). Degraded/drained peatlands typically have a larger GWP compared to
natural sites or rewetted sites because a large positive NEE outweighs the reduced $CH_4$ emissions
(Renau-Wilson et al., 2018a). The NEE and $CH_4$ fluxes from restored peatlands can be similar to
pristine peatlands, but exhibit greater variability (Wilson et al., 2016a; Strack et al., 2016).

Several studies have suggested the hypothesis that time since restoration is an important factor in
the GWP of peatlands (Augustin & Joosten, 2007; Bain et al., 2011; Waddington and Day 2007). In
particular, the restored sites may go through an initial period of high methane production and high
GWP because restored peatlands are often rapidly colonized by aerenchymatous vegetation, such as
*Eriophorum spp.* (Waddington and Day 2007, Cooper et al. 2014). This is followed by a period of
decreasing GWP as mosses and other peatland species become established (Augustin & Joosten,
2007; Bain et al., 2011). To test this hypothesis, more data is needed for peatlands "restored more
than 10 years previously" (Bacon et al., 2017). Also, it is valuable to have studies which directly
compare adjacent raised bog and cutover bog with different site histories.

Aquatic losses of carbon include dissolved organic carbon (DOC) and dissolved inorganic carbon
(DIC) in runoff as well as $CO_2$ evasion from open water. These have not been measured as
frequently as NEE and $CH_4$ flux (Dinsmore et al., 2010), but can represent a key component of the



net ecosystem carbon budget (NECB) (Kindler et al., 2011). Ignoring the aquatic carbon losses
would result in an overestimate of the carbon sink function of peatlands (Billet et al. 2010). The
DOC losses from temperate peatlands range from 5-36 g C $m^{-2}\,yr^{-1}$ and are lower for boreal
peatlands (Range: 4-13 g C $m^{-2}\,yr^{-1}$) (Evans et al., 2016). Few studies have simultaneously measured
a complete NECB for a peatland including the DIC flux (Nilsson et al. 2008) and $CO_2$ evasion from
open water (Dinsmore et al. 2010), even though $CO_2$ evasion has been found to be important to the
overall carbon balance, with a reported 2-year average of 12.8 g C $m^{-2}\,yr^{-1}$ for an intact peatland in
Scotland (Dinsmore et al. 2010). Further, these studies have focused on intact rather than restored
or recovering peatlands.

The growing body of scientific research on the GHG and carbon balance of peatlands and the
importance to global climate change means that it is increasingly important to consider new data in
the context of global studies. Often, boreal and temperate peatlands have similar conditions:
hydrological (consistently high water table), chemical (high carbon, often acidic peat soils),
ecological (often ground cover of *Sphagnum* mosses, with low nutrient sedges and ericaceous
shrubs). As similar factors (i.e. water table, plant ecology, growing season length, soil temperature,
etc.) are often cited as controlling factors for greenhouse gas fluxes, it may be possible to identify
global trends across boreal and temperate peatlands (e.g. Junkurst and Fiedler 2007).

The goal of this work is to quantify all of the major aspects of the carbon balance (NEE, $CH_4$ flux, and
aquatic losses as DOC, DIC, and $CO_2$ evasion), for a historically (ca. 1960) abandoned cutover bog
compared to an adjacent more recently restored (2009) raised bog. This study also presents the
measurements in the context of global studies on boreal and temperate peatlands with the aim of
identifying trends in NEE and $CH_4$ flux based on land condition (drained, restored, pristine), mean
annual water table, and vegetation cover (presence/lack of vegetation).




## 2. Materials and Methods

*2.1 Site Description*

Abbeyleix Bog (N 52.89714, W 7.35022, elevation approx. 90 m) is a natural peatland area in Co.

Laois, Ireland containing both un-harvested raised bog and historically harvested cutover bog (Fig.

1). This site is located in a temperate, oceanic climate with a mean annual rainfall of 844 mm and a

mean annual temperature of 9.9° C. Acidic, low nutrient, histosol, peat soils remain throughout the

raised and cutover bog with 5.0-8.5 m depth on the raised bog and 1-3 m depth on the cutover bog.

The areas of raised bog were impacted by surface drainage in the 1980's in preparation for

industrial extraction. Surface drains were installed at 15 m spacing to a depth of 1 m, and connected

with older, and deeper drains along a historic railway track and the margins of the cutover bog. The

plans for industrial extraction of the peat were abandoned due to resistance from the local

community, and the surface drains were blocked in 2009, 7 years before the start of this study. The

raised bog is mostly surrounded by cutover bog, which was domestically harvested for peat

between the 1870's and 1960's, and then abandoned (Ryle, 2013).




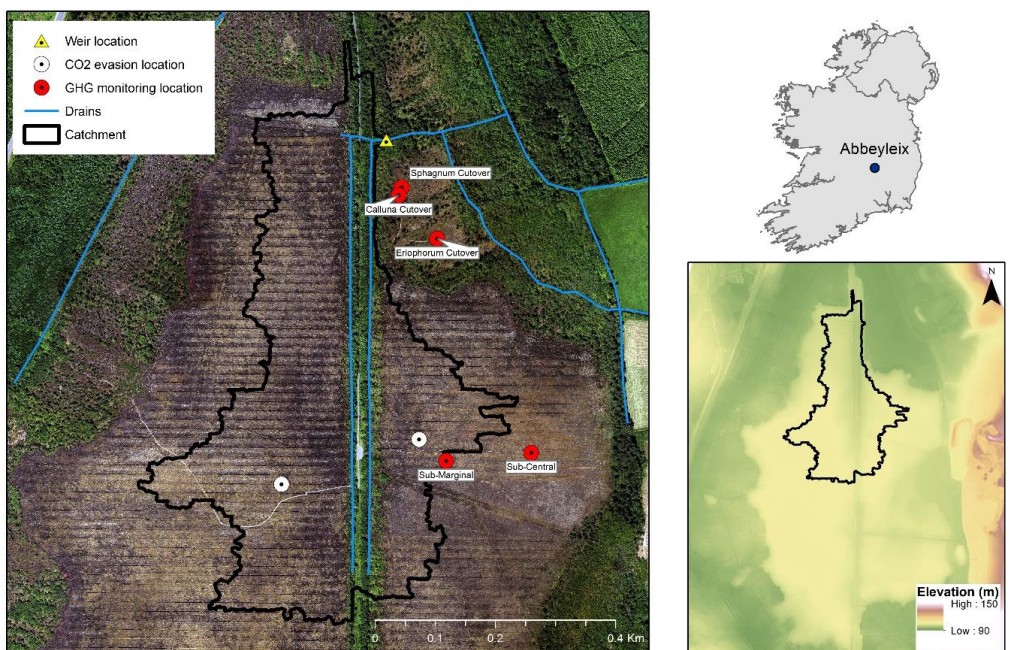

**Figure 1**. Location of the study site in Ireland; elevation map of Abbeyleix Bog (bottom right) showing the uncut raised bog surrounded by lower cutover bog and the higher esker complex to the east; an aerial photograph of the study site showing the weir catchment area, major drains, and sampling locations. In the aerial photograph the blocked surface drainage network on the raised bog can be seen as a set of horizontal lines and the historic railroad track can be seen as a vertical line through the middle of the photograph.

*2.2 Sampling Locations*
Five sampling locations were chosen to quantify GHG emissions, two on the uncut raised bog and
three on the cutover bog. These locations were chosen to represent 5 ecotypes, where the ecotype
refers to a distinct set of hydro-physical and ecological conditions. These 5 areas were chosen to
represent common ecotypes on raised and cutover bogs in Ireland with the help of ecologists from
the National Parks and Wildlife Service (NPWS).

On the raised bog, one study location was chosen in a Sub-Central ecotype, which is defined as
having a continuous *Sphagnum spp.* cover and continuously high water table but lacking the micro-
topography of hummocks and hollows. The Sub-Central ecotype is the highest quality bog
conditions found at this site. Another study location was chosen in a Sub-Marginal ecotype, which is
defined as having a discontinuous *Sphagnum spp.* moss cover and a mixed presence of both



relatively wet and dry bog vegetation (Table 1). Further description of raised bog ecotypes can be
found in Schouten et al. (2002).

On the cutover bog, three sampling locations were chosen based on distinctions in the plant
ecology. The Sphagnum Cutover ecotype contains a continuous *Sphagnum spp.* cover (primarily as
hummocks of *Sphagnum capilifolium* with some *Sphagnum subnitens and Sphagnum magellanicum*)
and a mixture of plant species similar to the Sub-Central ecotype. The Calluna Cutover ecotype
contains a low diversity of plant species characteristic of a well-drained peat soil, dominated by
heather (*Calluna vulgaris*), bare peat, and lichens (mostly *Cladonia portenosa*) similar to a faceband
ecotype on a raised bog. The Eriophorum Cutover ecotype is dominated by *Eriophorum*
*angustifolium*, and contains a moderate percent (21-54% in this study) cover of *Sphagnum spp.*
(Table 1). All sampling locations were chosen in open areas, excluding any large trees, shrubs or
other vegetation that could not fit under the gas sampling chambers (see Section 2.3). Six collars
were installed for each ecotype except for the Calluna Cutover ecotype where 5 collars were
installed. Collar locations were chosen to represent ecological variability within each ecotype. Plant
ecology was characterized for all collars in June 2016 and again in June 2017 with the help of
ecologists from the NPWS. The plant ecology was determined in terms of the percent cover of every
species present, averaged over the two years.

**Table 1.** Summary of the plant ecology for each ecotype in this study. Data is reported as the mean (range) of the collars within each ecotype.

| Ecotype | Percent *Spahgnum spp.* cover | Percent *Eriophorium spp.* cover | Percent *Calluna vulgaris* cover | Percent Total Plant Cover |
|---|---|---|---|---|
| Sphagnum Cutover | 94 (78 to 100) | 8 (3 to 23) | 16 (5 to 30) | 119 (103 to 134) |
| Calluna Cutover | 0 | 2 (0 to 3) | 35 (8 to 50) | 51 (18 to 68) |
| Eriophorium Cutover | 35 (21 to 54) | 51 (21 to 80) | 6 (2 to 15) | 103 (77 to 140) |

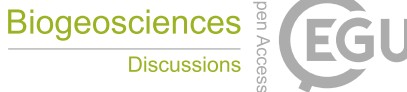



| | | | | |
|---|---|---|---|---|
| Sub-Marginal | 57 (15 to 89) | 13 (4 to 37) | 9 (2 to 15) | 100 (69 to 114) |
| Sub-Central | 98 (93 to 100) | 8 (1 to 39) | 2 (0 to 8) | 124 (107 to 151) |


*2.3 Meteorological Field Data*
On site, hourly measurements of air temperature and humidity (CS215 probe, Campbell Scientific,
Loughborough, UK), rainfall (ARG100 Tipping Bucket Raingauge, Campbell Scientific), barometric
pressure (PTB110 Barometer, Vaisala, Oyj, Finland), and soil temperature at 5 and 10 cm (PT100
temperature probes, Campbell Scientific) were recorded by a CR1000 Data logger (Campbell
Scientific). Soil temperature was also recorded at ecotypes by two LogBoxAA data loggers (Novus,
Miami, USA). Hourly phreatic water table was recorded in 5 cm diameter stilling wells located at
each of the five ecotypes by an Orphius Mini Level Logger (vented transducer, 0.1% error, OTT
Hydromet, Kempten, Germany). The ground elevation at the center of each collar was surveyed and
compared to the stilling well using an RTK GPS with ± 2mm accuracy (TDL 450L, Trimble,
Sunnyvale, CA), and the hourly water table at each collar was offset by this difference in elevation.
All collars were located within 8 m of the ecotype water table logger.

The hourly light intensity was measured in the field in units of W/m$^2$ using an LP02 Pyranometer
(Hukseflux Thermal Sensors, Delft, Netherlands). This sensor was calibrated to the
photosynthetically active radiation (PPFD) sensor (TPR-2, PP Systems) in units of ($\mu$mol m$^{-2}$ s$^{-1}$)
used during the field chamber measurements. A linear calibration between these two sensors was
found for both sunny and overcast days (n=27, r$^2$=0.82), which was used to convert hourly light
intensity to hourly PPFD.



*2.4 Greenhouse Gas Flux Measurements*
The closed static chamber method was used to measure greenhouse gas fluxes from all plots,
comparable to methods used in a large number of other studies, particularly on peatlands in Ireland
(e.g. Wilson et al. 2016b). A stainless steel collar was permanently installed 20 cm into the ground
at least two weeks before the start of sampling. This collar had a water trough to ensure a suitable
seal with the chamber. The chambers (60 x 60 x 30 cm equipped with a fan) were constructed in
house of clear polycarbonate for $CO_2$ measurements and opaque polystone$^{tm}$ for $CH_4$. A system of
wooden platforms was constructed 6-7 weeks before the start of sampling so that each collar could
be accessed without putting pressure on the ground surface adjacent to it. Platforms were placed
on piles to the base of the peat in the Sub-Central ecotype to prevent sinking into the bog. For $CO_2$
flux measurements, chambers were gently set on the collar and any pressure differential between
the chamber headspace and the ambient atmosphere was vented using a 5 cm$^2$ hole set in the side
of the chamber. The chamber was then sealed and the $CO_2$ concentration was recorded in the field
every 15 seconds for a period of 105 seconds using an EGM-4 infra-red gas analyser (PP Systems,
Amesbury, USA). $CO_2$ flux was calculated from the slope of the linear increase in $CO_2$ flux over time.
In order to maintain a constant temperature, particularly under high irradiance, a cooling system
was installed in this chamber which pumped water from an ice bath through a small radiator
located behind the fan to keep the variance of the chamber temperature to within 1°C during the
measurement. The $CO_2$ flux measurement was repeated under a range of light levels by artificially
shading the chamber. Ecosystem respiration is assumed to be the $CO_2$ flux where the light
transmitted into the chamber was zero. $CO_2$ flux measurements were conducted over 63 field days
between January 2016 and August 2017. A total of 3358 quality checked chamber measurements
for $CO_2$ flux were conducted over 29 collar locations.



For $CH_4$ flux measurements, gas samples of 20 mL each were extracted from the chamber every 10
minutes beginning 5 minutes after the chamber had been placed on the collar and sealed. These
samples were later analyzed in the lab on an Agilent Gas Chromatograph instrument with a flame
ionization detector and a 30 m long Elite-plot Q GC column. Samples were collected over 17 field
days between April 2017 and January 2018.

Additionally, the soil temperature at 5 and 10 cm depth, water table adjacent to the collar, air
temperature, and light level inside the chamber (for $CO_2$ flux measurements) were recorded for
each chamber closure at the time of sampling.
*2.5 NEE Modelling*
The NEE was modelled on an hourly basis to account for the expected diurnal variations, which is
driven by PPFD and temperature for the daytime uptake and night time release, respectively. Field
measurements of $CO_2$ flux were used to build collar specific empirical models of gross primary
production (GPP) and ecosystem respiration (ER). Hourly measurements of field variables were
input into these empirical models to calculate hourly GPP and ER, which were then summed to
calculate NEE.

Several models of GPP and ER were tested to fit the data (see Supplemental Section 1). These
models were judged based on the sum of the squares of the residuals and $r^2$ values. Models were
also checked to ensure that there was no bias or trend in the residuals with respect to independent
variables. The GPP model in Eq. (1) was found to best explain the variance in the field data for all of
the 29 collars.

$$GPP = -(a + c * \sin((JDAY + 215)/365 * 2\pi)) * \frac{PAR}{PAR+b} * exp(T_{5cm} * d) * (1 + WT * e) \qquad (1)$$



where $a$, $b$, $c$, $d$, and $e$ are collar specific empirical fitted model parameters and JDAY is the Julian
day of the year, PPFD is the light level in (µmol m$^{-2}$ s$^{-1}$), $T_{5cm}$ is the soil temperature at 5 cm, and WT
is the water level in cm below ground surface at the collar. The r$^2$ value of the modelled versus
measured data using Eq. (1) ranged between 0.77 and 0.94 for each of the 29 collars (Table S3).

For ER, the model in Eq. (2) was found to best explain the variance in the field data for all of the 29
collars. For this ER model, the r$^2$ values ranged from 0.74 to 0.94 for each of the collars.
$$ER = \left( a + d * \sin \left( \frac{JDAY + 215}{182.5} * 2\pi \right) + e * WT \right)$$
$$* \exp \left( c * \left( \frac{1}{(283.15 - 227.13)} - \frac{1}{(TK5cm - 227.13)} \right) \right) + b * WT \qquad (2)$$

where $a$, $b$, $c$, $d$, and $e$ are fitting parameters, and other variables are as above. Fitting parameters
and more information on the GPP and ER models tested can be found in Supplemental Section 1.

Hourly water level, $T_{5cm}$, PAR, and Julian day data were input into Eq. (1) and Eq. (2) (with the collar
specific fitting parameters) to calculate hourly GPP and ER at each collar.

*2.6 Methane Modelling*
In contrast to $CO_2$ flux, $CH_4$ fluxes are expected to be much more constant throughout the day
(Pypker et al. 2013) with apparently random variation. Therefore, $CH_4$ fluxes from each collar could
be calculated from the average measured flux over a given time period (as in Strack et al. 2014).
However, in this case, methane flux measurements were not conducted over the entire 2 year time
period because of equipment issues. Thus, a model was constructed for the purpose of
extrapolating the field data to the entire study period. The field data of $CH_4$ flux from all collars
were normalized by the collar average $CH_4$ flux and lumped together to model the average temporal



variation in CH$_4$ flux. The variations in CH$_4$ flux was modelled according to the Julian day of year and
soil temperature (Eq. S11). Due to limited data, methane flux variations were assumed to follow the
same temporal trend across all ecotypes. The overall average temporal variation was then
multiplied by the average measured methane flux at a given collar. The model gave little difference
between 2016 and 2017, and as field data was only collected in 2017, it was assumed that the
methane flux from both years was the same for the purposes of calculating annual carbon balance
and GWP.
*2.7 Aquatic Carbon Losses*
A thin plate V-notch weir was installed to measure hourly discharge from a 249,000 m$^2$ catchment
area onsite (as shown in Fig. 1). The weir catchment area was delineated in ARC-GIS using a digital
terrain map based on LiDAR survey data from 2013. The majority of this catchment area was
composed of marginal and sub-marginal uncut raised bog (>90%) as well as lightly forested drains
along a bog road (<10%). Aquatic carbon losses as DOC and DIC were quantified at this location
only, and assumed to be the same for all ecotypes (even those adjacent to but outside of this
catchment area) given the difficulty in resolving the relative contributions of each ecotype to the
total DOC flux. The DOC concentration was measured weekly in 2016 and every 12 hours (with a
few gaps) from January through November 2017. DOC samples were filtered in the field using a
0.45 μm cellulose syringe filter after rinsing the syringe and filter with 20 mL of sample. Samples
were then acidified to pH 2 using 10% HCl to preserve them and stored under refrigeration at 4° C
and analysed within two months. The DOC concentration was measured by UV absorbance as in
other studies (e.g. Jager et al. 2008, Koehler et al., 2009) at wavelength 254 nm. A site specific
calibration curve was determined between 254 nm UV absorbance and DOC concentration
measured using a Vario Total Organic Carbon (TOC) Select Analyzer (Elementar, Langenselbold,
Germany). This was undertaken on samples collected from January 2016 to April 2016, July 2016,
and July 2017 (r$^2$ =0.997, n=76). The error of this method was ± 1.1 mg C/L based on the standard





deviation of the residuals. The hourly discharge at the weir was multiplied by the most recent DOC
concentration measurement to calculate a carbon flux as DOC from the catchment. This value was
then divided by the catchment area to calculate the aquatic carbon loss as DOC per $m^2$.

The DIC concentration at the weir was calculated from the aqueous partial pressure of $CO_2$ as well
as the pH and temperature using equations from Gelbrecht et al. (1998) as in Nillson et al. (2008)
where dissolved $CO_2$ was included as part of DIC. Partial pressure of $CO_2$, was measured onsite in
triplicate by filling, then sealing a 250 mL bottle with 200 mL of water sample. Circulated air was
bubbled through the sample and the change in $CO_2$ concentration in the headspace was measured
over time using an EGM-4 infra-red gas analyser (PP Systems, Amesbury, USA) until the
concentration was constant (10-12 minutes). The initial partial pressure of dissolved $CO_2$ in the
sample was then back calculated from the total change in $CO_2$ concentration in the headspace. A
total of 7 DIC measurements were taken at the weir between November 2016 and October 2017.
The average DIC concentration was multiplied by the hourly discharge and divided by the
catchment area to calculate the aquatic carbon loss as DIC per $m^2$.

$CO_2$ evasion occurred from the open water areas of blocked drains on the raised bog and from the
functioning drain network upstream of the weir. $CO_2$ evasion was measured in triplicate with a
CPY-4 (PP systems, Amesbury, USA) chamber fitted to a small floating raft and EGM-4 gas analyser.
A total of 15 measurements of $CO_2$ evasion were conducted between two locations of blocked
drains on the raised bog (Fig. 1), and 8 measurements were conducted just upstream of the weir
from November 2016 to July 2017.

For the calculation of the global warming potential, 90% of the DOC loss is assumed to be converted
to $CO_2$ and 10% to longer term storage (after Evans et al. 2016), while 100% of the DOC flux is



included in the calculation of the carbon balance for the system. All of the DIC loss is assumed to be
converted to atmospheric $CO_2$ as DIC is almost entirely composed of dissolved supersaturated $CO_2$.

*2.8 Statistical Analysis*
The ecotype variance in the NEE can be calculated as the sum of the within collar variance and the
between collar variance. The within collar variance was calculated from the sum of model error and
the error of input field variables. The annual model error was calculated from the standard
deviation of the residuals for GPP and ER models for each collar on an hourly time step and
propagated for the entire year. Similarly, the field inputs into the NEE models were assumed to
have a hourly random variation of ± 1°C, ±1 cm WT, and ± 5% PAR. The effect of which on the NEE,
was calculated from sensitivity analysis, which was run for all models and propagated for the entire
year. The variance in the ecotype $CH_4$ flux was also calculated from the sum of the within collar
variance and the between collar variance. The annual within collar standard deviation of $CH_4$ flux
was assumed to be ± 30% of the collar average annual $CH_4$ flux, or 2.8 g C-$CH_4$ m$^{-2}$ yr$^{-1}$, which was
applied to all collars. For the carbon balance and GWP, the variance in NEE and $CH_4$ flux was
summed with the variance due to measurement error in the DOC flux, DIC flux, and $CO_2$ evasion.
Significant differences between ecotype annual carbon balance, and GWP was determined using 1-
way ANOVA and Bonferroni confidence intervals. The significance of the linear regressions was
determined with Minitab 18 Statistical Software.

## 3. Results

*3.1 Environmental Monitoring*
The annual rainfall at Abbeyleix Bog was 746 mm in 2016 and 840 mm in 2017, compared to the
2001-2016 annual average of 838 mm at the Ballyroan (Oatlands) daily rainfall station, located
approximately 5 km NE of the site. The mean annual temperature at Abbeyleix bog was 9.6° C and



9.7° C in 2016 and 2017, similar to the (1978-2007) average of 9.9° C. Mean daily PPFD, air
temperature, and monthly rainfall are shown in Figure 2 over the study period. The mean annual
water table (MAWT) was within 2 cm at all ecotypes between the two years. The winter (Oct-Mar)
water table was higher than summer (Apr-Sep) water table, as expected (Fig. 3). The average soil
pore water pH was 4.7 (range: 4.4-5.1) for all ecotypes.

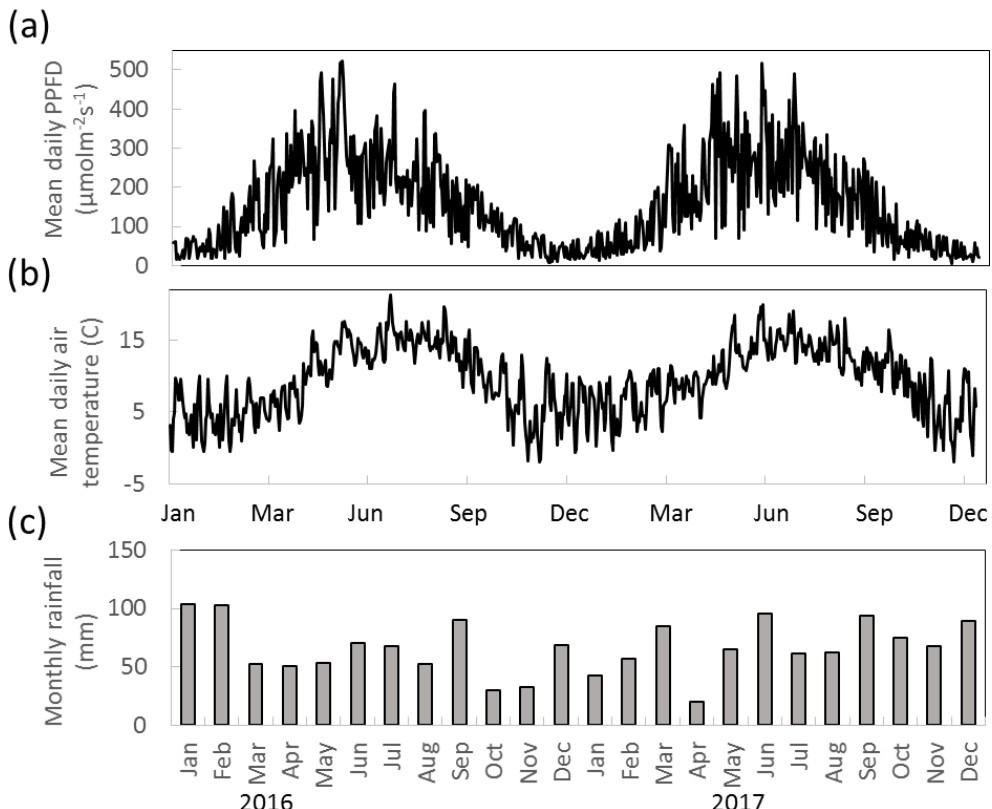

**Figure 2**. (a) Mean daily PPFD, (b) mean daily temperature, and (c) monthly rainfall at Abbeyleix Bog in 2016 and 2017.





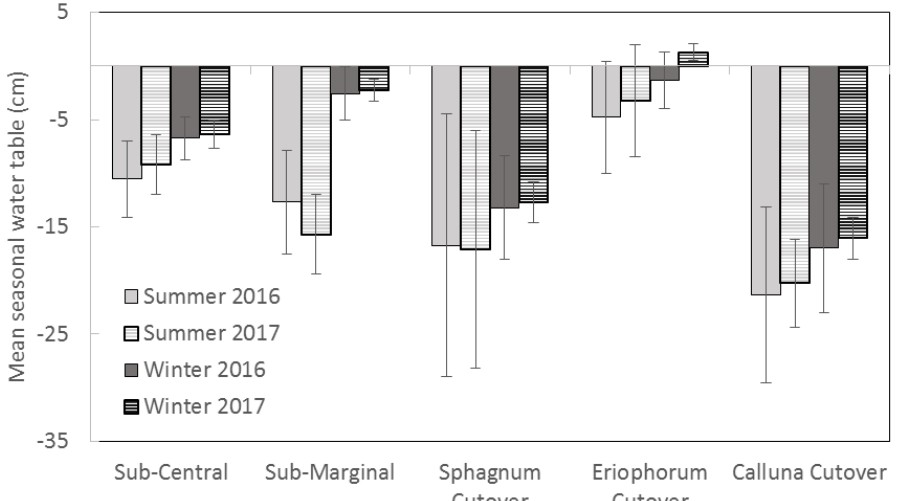

**Figure 3**. Mean seasonal water table for each of the ecotypes for summer (Apr-Sep) and winter (Oct-Mar), where the mean annual water table is measured with respect to the springtime peat surface or sphagnum surface (if present).

*3.2 CO$_2$ and CH$_4$ Gas Fluxes*
The modeled annual GPP, ER, and NEE for each collar is shown in (Table S5). The ecotype CO$_2$ fluxes
were calculated as the average of all collars in each ecotype. The seasonal trend in modeled
monthly GPP and ER were similar among all ecotypes increasing in the summer and decreasing in
the winter (Fig. 4a & 4b). The Sphagnum Cutover ecotype had the largest monthly GPP from
January to June both years. The monthly ER was highest at the Calluna Cutover ecotype, especially
during the summer months. The ecotypes show different seasonal trends in cumulative NEE (Fig.
4c). The Sphagnum Cutover and the Sub-Central ecotypes were net CO$_2$ sinks (negative slope) from
March (March 27 for Sub-central and March 4 for Sphagnum Cutover) to October 24, 2016 and
April 24 to October 7, 2017 and CO$_2$ sources the rest of the year, showing an overall similar pattern
to other studies of intact peatlands (e.g. Gažovič et al. 2013). By contrast, the Calluna Cutover
ecotype was the strongest CO$_2$ source during the summer months. The Sub-Marginal ecotype is an
overall moderate CO$_2$ source both years with a minor net CO$_2$ uptake occurring during summer of
2017. The Eriophorum Cutover ecotypes is approximately CO$_2$ neutral for much of the year with
short periods of CO$_2$ uptake during the summer months.






The temporal variation in methane flux was captured reasonably well ($r^2$ =0.61) by the model (Fig.
5). The methane data was extrapolated to an annual period using this model. Annual methane
fluxes by ecotype are shown in Figure 6 and annual methane emissions for each collar are shown in
Table S5. The methane emissions are highest for the Eriophorum Cutover (14.2 ±4.8 g C-$CH_4$ $m^{-2}$ $yr^{-1}$)
and Sub-Central ecotypes (12.6 ±7.9 g C-$CH_4$ $m^{-2}$ $yr^{-1}$), which have the highest mean annual water
table. The annual $CH_4$ flux at the Sub-Central ecotype is highly variable with a range of 1.2 to 19.3 g
C-$CH_4$ $m^{-2}$ $yr^{-1}$ between collars. The annual methane flux is lowest for the Calluna Cutover ecotype
(2.7 ±1.4 g C-$CH_4$ $m^{-2}$ $yr^{-1}$).



**Figure 4.** Monthly (a) GPP and (b) ER, and (c) cumulative NEE for each ecotype for 2016 and 2017, where the ecotype values are the average of all collars in the ecotype.



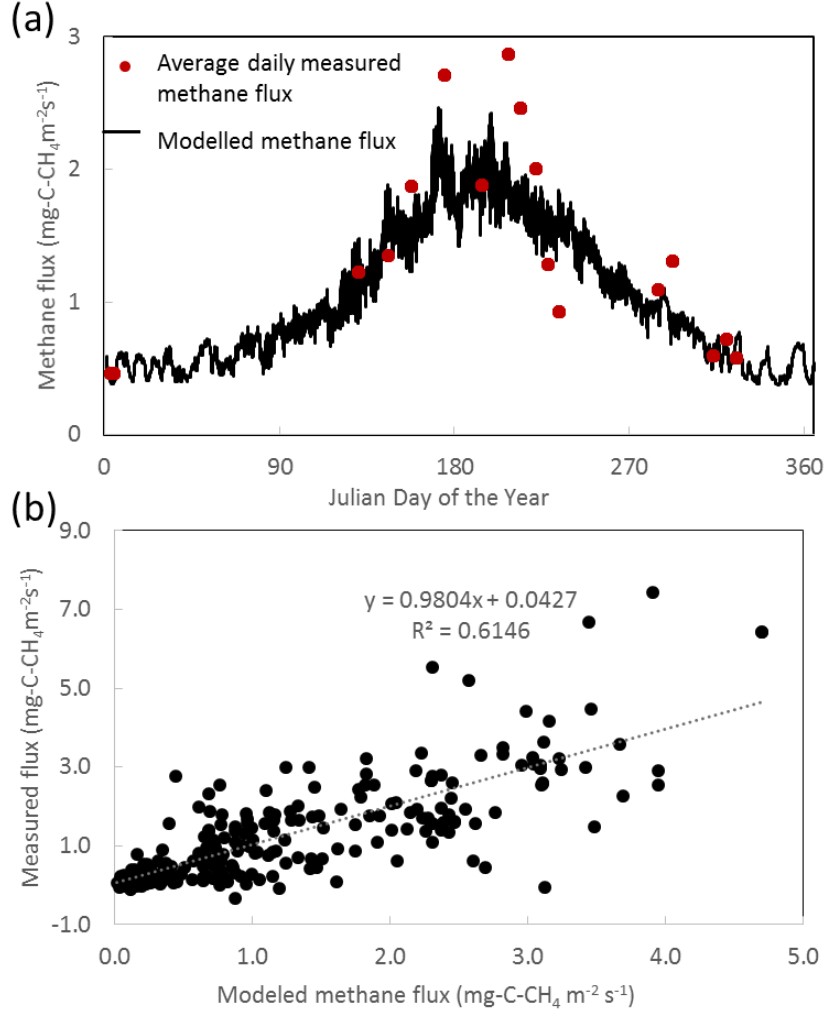

**Figure 5**. (a) The average daily methane flux compared to the modelled temporal fluctuations in methane flux, and (b) the modelled vs. measured methane flux when the temporal variation in multiplied by the collar average flux.



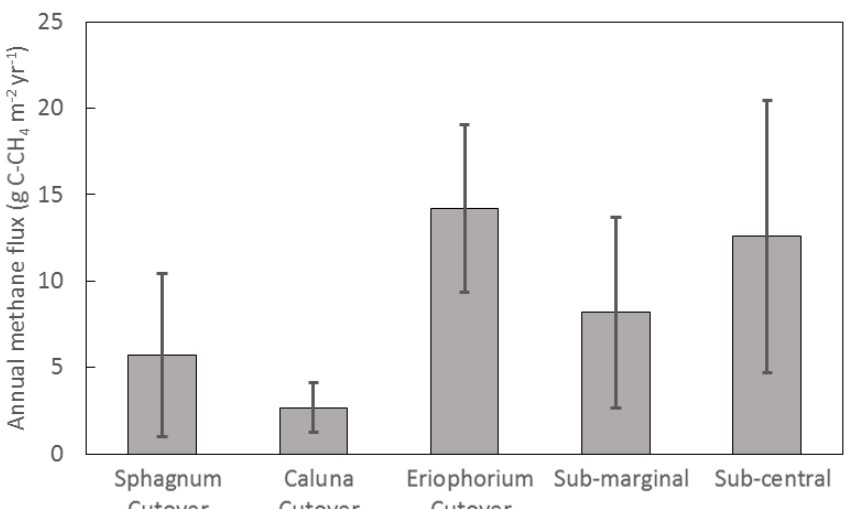

**Figure 6**. Annual methane flux for each ecotype averaged over all collars in the ecotype.


*3.3 Aquatic Carbon Losses*

The DOC concentrations showed a seasonal trend for both years - higher between approx. June and

November (46.0 ±3.0 mg L⁻¹) and lower between December and May (34.5 ±2.3 mg L⁻¹) (Fig. 7).

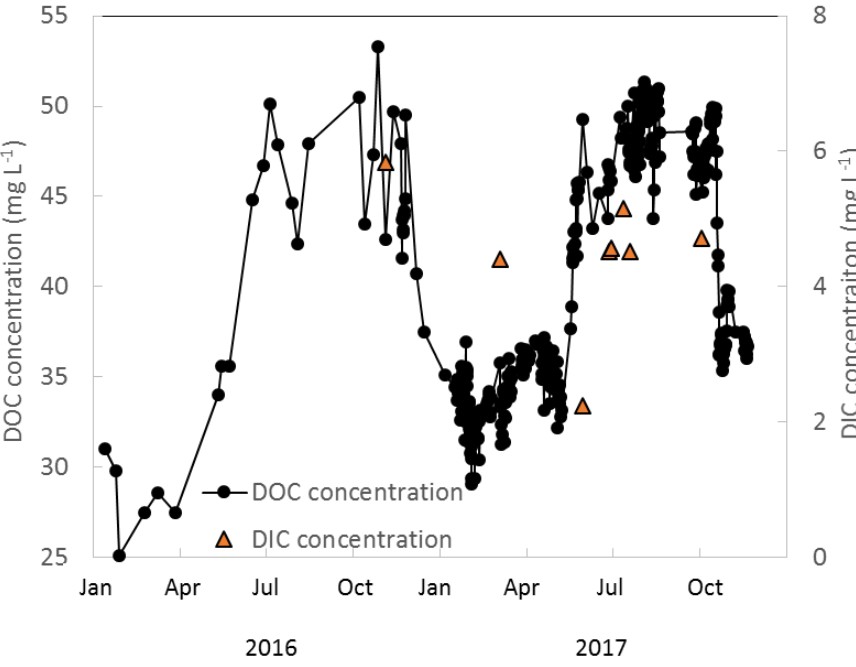





**Figure 7**. Measured DOC and DIC concentrations (mg L$^{-1}$) over a two year period (2016 and 2017) at the weir.


No trend was observed with respect to discharge. The discharge at the weir site was much higher in
the winter months, with a resulting higher total DOC flux over those months. Annual losses of DOC
were 8.0 ±1.6 and 12.8 ±2.5 g C m$^{-2}$ yr$^{-1}$ for 2016 and 2017, respectively. Seven DIC measurements
were conducted at the weir site between November 2016 and October 2017. The average DIC
concentration at the weir was 4.6 ±1.1 mg L$^{-1}$, excluding 1 low outlier (2.2 mg L$^{-1}$) on June 2, 2017
(Fig. 7). Based on this limited amount of data there is no significant trend in DIC concentration with
respect to season, temperature, or discharge, so it was assumed constant throughout the 2 year
study period. Annual carbon losses as DIC were 1.1 ±0.2 and 1.5 ±0.3 g C m$^{-2}$ yr$^{-1}$. These values of
annual aquatic carbon loss for DOC and DIC were applied to each of the ecotypes equally when
calculating the carbon balance and GWP. Open water $CO_2$ evasion was measured for two blocked
drains on the raised bog and just upstream of the weir. The average $CO_2$ evasion rate from the two
blocked drains (n=15) was 5.1 x 10$^{-3}$ ±2.9 x 10$^{-3}$ mg C-$CO_2$ m$^{-2}$ s$^{-1}$ and was somewhat higher at the
weir (n=8) as 9.2 x 10$^{-3}$ ±3.2 x 10$^{-3}$ mg C-$CO_2$ m$^{-2}$ s$^{-1}$ (Fig. 8).

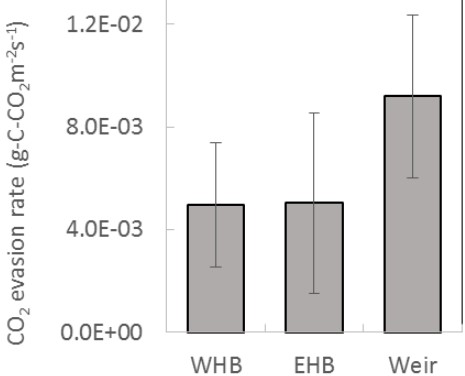

**Figure 8**. $CO_2$ evasion rate measured at two blocked drains on the high bog (WHB and EHB) and just upstream of the weir. Data was collected between March and July 2017 at the WHB location (n=7), November 2016 and July 2017 at the EHB location (n=8), and December 2016 and July 2017 at the weir location (n=8).



Based on this limited data set, there was no significant trend in evasion rate with respect to season,
temperature, or (at the weir site) discharge. $CO_2$ evasion rate was thus assumed constant and
extrapolated to give an annual carbon loss as $CO_2$ evasion of 162 ±91 g C-$CO_2$ m$^{-2}$ yr$^{-1}$ and 290 ±100
g C-$CO_2$ m$^{-2}$ yr$^{-1}$ for open water blocked ditches and active drain network of the weir, respectively.
The open water areas in the drain network contributing to the weir were ~0.9% of the total
catchment area to give a carbon loss of 2.7 ±0.9 g C-$CO_2$ m$^{-2}$ yr$^{-1}$ for the weir catchment area as a
whole. As above, this was applied equally all ecotypes. Open water areas of blocked drains only
occurred near one of the ecotypes (Sub-Marginal), where they were estimated to be 2.8% of the
total surface area. This gives an additional carbon loss in the Sub-Marginal ecotype of 4.5 ±2.6 g C-
$CO_2$ m$^{-2}$ yr$^{-1}$.
*3.4 Carbon Balance and GWP by Ecotype*
The NEE, $CH_4$ fluxes, and the aquatic losses of carbon were compiled to calculate the carbon balance
and GWP for each ecotype (Fig. 9). The Calluna Cutover ecotype was a substantial carbon source of
260 ±70 g C-$CO_2$ m$^{-2}$ yr$^{-1}$ and 218 ±78 g C-$CO_2$ m$^{-2}$ yr$^{-1}$ for 2016 and 2017, respectively. This ecotype
was significantly higher than all the other ecotypes in 2016 ($p < 0.001$) and 2017 ($p=0.011$). The
annual carbon balance for the other ecotypes was not significantly different from carbon neutral.
However, four of the six collars at the Sub-Central ecotype were significant carbon sinks both of the
years (range -25 to -97 g C-$CO_2$ m$^{-2}$ yr$^{-1}$). One collar in the Sub-Central ecotype was found to be a
significant carbon source both of the measured years (51 and 62 g C-$C0_2$ m$^{-2}$ yr$^{-1}$).There is
substantial variation between collars within each ecotype for NEE and $CH_4$ flux, which is the largest
source of error in ecotype carbon balance and GWP.
All ecotypes had an average positive GPW both years, with the lowest average GWP of 2.1 ± 2.4 tons
$CO_2$-eq m$^{-2}$ yr$^{-1}$ at the Sphagnum Cutover ecotype and the highest average GWP occurring at the
Calluna Cutover ecotype of 9.8 ± 3.5 tons $CO_2$-eq m$^{-2}$ yr$^{-1}$. The GWP at the Calluna Cutover ecotype
was significantly higher than the Sphagnum Cutover ($p = 0.002$) and Sub-Central ecotype ($p =

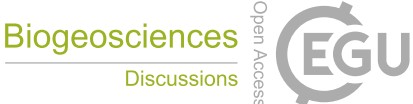

0.028) in 2016 and only the Sphagnum Cutover ecotype in 2017 (p = 0.018) (Fig. 9b). Methane
emissions account for 12% and 14% of the GWP at the Calluna Cutover ecotype in 2016 and 2017,
respectively. Methane emissions account for the majority of the total GWP in all other ecotypes (65-

414    146%).

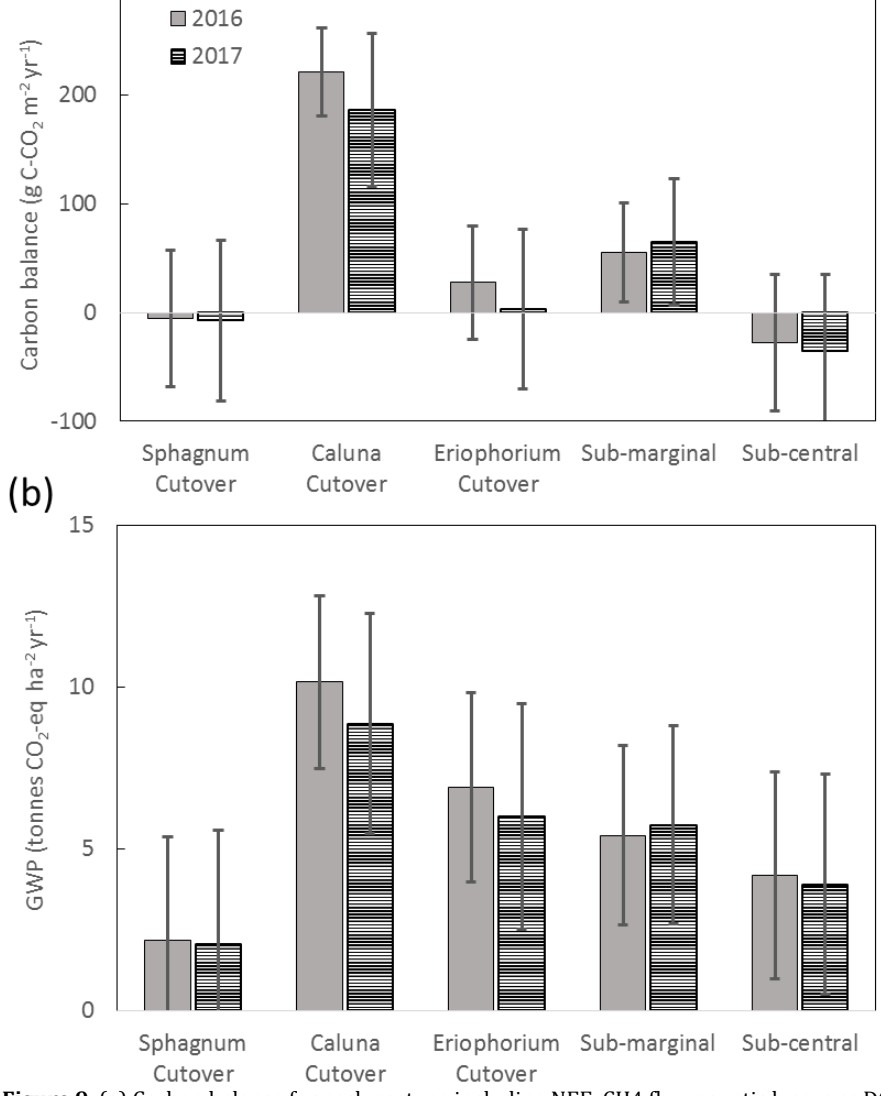

**Figure 9.** (a) Carbon balance for each ecotype including NEE, CH4 flux, aquatic losses as DOC and DIC, and open water CO₂ evasion. (b) Global warming potential for each ecotype.

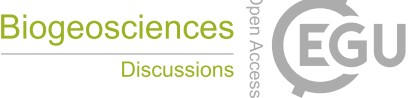



*3.5 Drivers of NEE and GWP*
Environmental drivers of the annual carbon balance, $CH_4$ flux, and GWP were analyzed by
comparing the data from each of the 29 collars. There is a significant (p=0.015) but weak ($r^2 = 0.20$)
negative linear correlation between the two year average annual carbon balance and the average
MAWT (Fig. 10a). This particular data set is skewed by the Sphagnum Cutover ecotype, where there
is a relatively low water table and an overall neutral carbon balance due to the presence of
*Sphagnum spp.* hummocks. If the Sphagnum Cutover ecotype is excluded, the linear regression
between average carbon balance and MAWT is highly significant (p<0.001) with a stronger
correlation ($r^2 = 0.41$). The annual $CH_4$ flux has a significant (p < 0.001) positive linear correlation
($r^2=0.51$) with the average MAWT (Fig. 10b). The trends in $CH_4$ flux and carbon balance with
respect to MAWT offset each other such that there is no trend (p = 0.91, $r^2 < 0.01$) in GWP with
respect to mean annual water table (Fig. 10c).

The collar annual average GWP has a highly significant (p < 0.001) negative linear correlation ($r^2 =$
0.58) with the percent *Sphagnum spp.* cover in the collar (Fig. 10f). The percentage *Sphagnum spp.*
cover and *Eriophorum spp.* cover in the collar seem to be correlated in a non-linear fashion with the
average annual carbon balance and the annual $CH_4$ flux, respectively (Fig. 10 d,e). In particular, the
annual $CH_4$ flux is greater than ~9 g $C\text{-}CH_4$ $m^{-2}$ $yr^{-1}$ for all collars where the percentage *Eriophorum*
*spp.* cover is higher than 10%.





**Figure 10.** Trends in collar annual carbon balance, CH$_4$ flux, and GWP plotted against mean annual water table (MAWT) (a-c) and percent genus cover (d-f). Data is averaged over the two year period.





**4. Discussion**
*4.1 Carbon balance and GWP*
For the first time, all of the major aspects of the carbon balance were measured simultaneously in a
recovering peatland. Also, the carbon balance of ecotypes with different degradation histories was
compared for a naturally recovering old cutover bog and a restored unharvested raised bog.
Although the NEE is the most variable component of the carbon balance and drives the trends in the
carbon balance, it is not necessarily the largest component of the carbon balance. Other aspects of
the carbon balance become proportionally more important when the NEE is near neutral. For
example, the NEE at the Eriophorum Cutover ecotype in 2016 was $+2 \pm 53$ g C-$CO_2$ m$^{-2}$ yr$^{-1}$. The
magnitude of the aquatic carbon loss in 2016 ($11.8 \pm 1.8$ g C m$^{-2}$ yr$^{-1}$) was actually larger than the
NEE at this ecotype. The total NECB was also measured for a Boreal oligotrophic mire in northern
Sweden (Nilsson et al. 2008) and Auchencorth Moss, a lowland bog in Scotland (Dinsmore et al.
2010). The average annual DOC losses found in this study (10.4 g C m$^{-2}$ yr$^{-1}$) are comparable to the
average annual losses reported in Nillson et al. (2008) of 13.0 g C m$^{-2}$ yr$^{-1}$ and lower than those
reported Dinsmore et al. (2010) of 25.4 g C m$^{-2}$ yr$^{-1}$. The DIC losses in this study (1.3 g C m$^{-2}$ yr$^{-1}$,
including super-saturated $CO_2$ as DIC) are lower than the values reported in Nilsson et al. (2008)
and Dinsmore et al. (2010) of 2.0 and 4.6 g C m$^{-2}$ yr$^{-1}$, respectively. This is partially because the
average DIC concentration measured in this study ($4.6 \pm 1.1$ mg C/L) is somewhat lower than that
reported in Nillson et al. (2008) of 9.6 mg C/L and at Auchencorth Moss (Dinsmore et al. 2013) of
8.65 mg C/L. The annual open water $CO_2$ evasion found in this study (2.7 or 7.2 g C m$^{-2}$ yr$^{-1}$) is lower
than what was reported in Dinsmore et al. (2010) (12.7 g C m$^{-2}$ yr$^{-1}$), but this is dependent on the
geometry of the system as water surface area is a factor in the calculation. Also, the floating
chamber method used in this study may underestimate total $CO_2$ evasion (Dinsmore et al. 2010).
The overall two year average aquatic carbon loss found in this study (14.4 g C m$^{-2}$ yr$^{-1}$) is





comparable to Nilsson et al. (2008) (17.8 g C m$^{-2}$ yr$^{-1}$) and lower than Dinsmore et al. (2010) (43.8 g
C m$^{-2}$ yr$^{-1}$).

The Calluna Cutover ecotype was found to be a substantial carbon source and this is likely due to a
lower water table and a plant ecology reflective of a degraded peatland. The Eriophorum Cutover
ecotype has the highest mean annual water table and the highest *Eriophorum spp.* cover; both of
which are related to an increase in the observed methane flux. Even with the increased methane
flux at the Eriophorum Cutover ecotype, the GWP at this ecotype was not higher than the Calluna
Cutover ecotype. This agrees with Wilson et al. (2016b), where a rewetted bog in Ireland was found
to have a lower GWP than a well-drained site even where *Eriophorum angustifolium* developed. The
*Sphagnum spp.* dominated ecotypes (Sphagnum Cutover and Sub-Central) were the lowest average
GWP sources, and *Sphagnum spp.* cover was negatively correlated to the GWP at the collar scale. In
terms of restoration, this suggests that there is GHG benefit for both raising the water table as well
as establishing high quality bog vegetation such as *Sphagnum spp.*

The eco-hydrological conditions seem to be what determines GHG emissions, rather than time since
restoration. The data here do not support the hypothesis that time since restoration/abondonment
*per se* is an important factor in the GHG emissions (once vegetation is established as discussed
below). For example, all three of the cutover sites were presumably abandoned at the same time
(circa 1960's). However, these three sites have very different $CO_2$ and $CH_4$ emissions despite their
close physical proximity (within 200 m), similarities in soil, and a shared site history. Similarly, the
raised bog ecotypes (Sub-Central and Sub-marginal) were restored more recently by drain blocking
in 2009. The average carbon balance and GWP of the Sub-Marginal ecotype falls within the range of
the much older cutover ecotypes, and the Sub-Central ecotype has a similar average GWP to the
Sphagnum Cutover ecotype. This hypothesis would only be true if is there is an eco-hydrological



trajectory in the years post restoration/abandonment where *Eriophorum spp.* cover decreases or
*Sphagnum spp.* cover increases, for example.  Further, although the Calluna Cutover location is
much higher carbon source than the Sub-Marginal or Sub-Central locations. This area is similar
ecologically (and presumably in terms of hydrologic conditions) to the large areas of the uncut
raised bog, which are heavily degraded. This type of habitat seems to be the most common habitat
in the cutover areas in Abbeyleix bog, and is probably similar to much of the degraded bog areas in
Ireland. In the absence of restoration works, this ecotype remains a large carbon source more than
5 decades after abandonment.

There is a need for simple methodologies to predict greenhouse emissions from peatlands for
policy and management, particularly from data that are available at the regional or national scale.
Water table, vegetation cover, and soil temperature have been previously suggested as potential
predictive metrics of GHG fluxes from peatlands (Strack et al. 2016). Hence, a simple linear
regression based on MAWT (in cm below ground level) and percent genus cover was fit to the data
from the 29 collars in this study to predict the annual carbon balance (Eq. (3), $r^2 = 0.71$) and $CH_4$
flux (Eq. (4), $r^2 = 0.56$).

Annual carbon balance = 117.9 - 6.23*(MAWT) - 2.1*(Percent *Sphagnum spp.*)        (3)

Annual $CH_4$ Flux = 12.23 + 0.440*(MAWT) + 0.0754*(Percent *Eriophorum spp.*)        (4)

where annual carbon balance and annual $CH_4$ flux are in units of g C m$^{-2}$ yr$^{-1}$. While these coefficients
are site specific, these metrics may be useful for comparison to future studies.

*4.2 Comparisons with Global Studies of Boreal and Temperate Peatlands*
The annual NEE, $CH_4$ flux, and water table data from the ecotypes in this study were compared to
global studies of boreal and temperate peatlands. The data from global studies was divided into
three generic categories as follows:

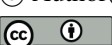


- • Pristine/Intact peatlands; - those peatlands that have not been harvested, undergone

- intensive agriculture or forestry, and are not heavily impacted by drainage or other

- disturbance;

- • Bare peat sites; - previous peat extraction sites where there is an absence of vegetation

- cover;

- • Restored/Degraded/Recovering peatlands; - all other peatlands are grouped into this

- category for this comparison.

This compilation of data focuses on low nutrient (if specified, pH<6) natural and semi-natural sites
and excludes sites that are actively used for intensive agriculture or forestry.

For both vegetated and bare peat sites, there is a negative correlation between MAWT and NEE
(Fig. 11). Both intact peatlands and variously degraded/recovering peatlands fall on the same trend
line, agreeing with Wilson et al. (2016a). Annual NEE for vegetated sites followed a linear trend
with respect to MAWT with slope of -4.5 g C-CO$_2$ m$^{-2}$ yr$^{-1}$ per cm rise in MAWT and an intercept of -
92 g C-CO$_2$m$^{-1}$yr$^{-1}$. The slope is similar to that reported from a review of studies of peatlands with
MAWT higher than -30 cm (Wilson et al. 2016a) of -2.0 ±1.0 and -5.0 ±2.0 g C-CO$_2$ m$^{-2}$ yr$^{-1}$ per cm
rise in MAWT for boreal and temperate peatlands, respectively. However, the trend in NEE with
respect to MAWT should be interpreted with some caution because of the difficulty of generalizing
across sites based on simple water table proxies (Wilson et al. 2016a). For example, there was a
"highly peatland-specific dependency (i.e., with different offsets and slopes) of the CO$_2$ response to
water table depth" for grassland peatlands in Germany (Tiemeyer et al. 2017), although, that study
looked at grasslands, which may have much more variability in soil type, land management,
nutrient status, etc. than the natural and semi-natural low nutrient sites shown in Figure 11. This
trend may also break down as MAWT becomes too low (e.g. Tiemeyer et al. 2017) because soil



respiration can be limited if the soil is too dry (Briones et al., 2014). Thus, climate patterns could be
an important factor in $CO_2$ response to water table (Tiemeyer et al. 2017).

Based on the data collected in Figure 11, intact peatlands occur at a narrower range of mean annual
water table and NEE. This is expected because degraded peatlands can have a wider range of site
histories and eco-hydrological conditions (Wilson et al. 2016a). This agrees with Strack et al.
(2016) who reported greater variation in $CO_2$ and $CH_4$ fluxes at restored plots when compared to
either unrestored or natural plots. As with data from this study, this may suggest that restoration of
high quality peatland ecology has an additional NEE benefit beyond raising the water table.

The Sub-Central ecotype in this study has continuous *Sphagnum spp.* lawns similar to an intact
peatland. This ecotype has a mean annual NEE of -57 g $C$-$CO_2$ m$^{-2}$ yr$^{-1}$ and a mean annual water table
of -8.2 cm. This is close to the overall average NEE (-60 g $C$-$CO_2$ m$^{-2}$ yr$^{-1}$) and mean annual water
table (-9 cm) for intact/pristine peatlands in this figure. This comparison is valuable for validating
the data for the other ecotypes because the carbon balance of natural bogs is comparatively better
characterized than degraded systems and the potential for systematic bias in chamber
measurements. The Calluna Cutover ecotype from this study has an exceptionally high NEE (222 g
$C$-$CO_2$ m$^{-2}$ yr$^{-1}$) for the mean annual water table (-18.6 cm) compared to the NEE (-5 g $C$-$CO_2$ m$^{-2}$ yr$^{-1}$)
predicted from the best fit trend line of vegetated sites.

Also, as shown in Figure 11, bare peat sites have a higher NEE than vegetated sites at a given
MAWT, and these trend lines diverge at higher MAWT. As it can take decades for vegetation to be
established in industrially harvested peatlands (Wilson et al., 2015), this data would suggest that
restoration to encourage plant colonization could reduce the short term $CO_2$ emissions even if no
other restoration works are undertaken. This data set could be used to predict the $CO_2$ reduction



from raising the water table as well as establishing vegetation on bare peat sites. Further, peatlands
may be large carbon sinks in the years immediately post restoration as vegetation recovers due to
the rapid, subsequent increase in vegetation biomass. For example, an annual NEE of -473 g C-$CO_2$
$m^{-2} yr^{-1}$ was reported by Waddington et al. (2010) one year post restoration for sites where
herbaceous vegetation increased dramatically. This may explain some of the low outliers in Figure
11 for restored/recovering sites. Three of the low outliers in Figure 11 are from Strack et al. (2014),
which is 4 years post restoration with a growing season NEE of -162, -121- and -126 g C-$CO_2$ $m^{-2}$ for
mean seasonal water tables of -21.3, -24.9 and -28.2 cm, respectively.

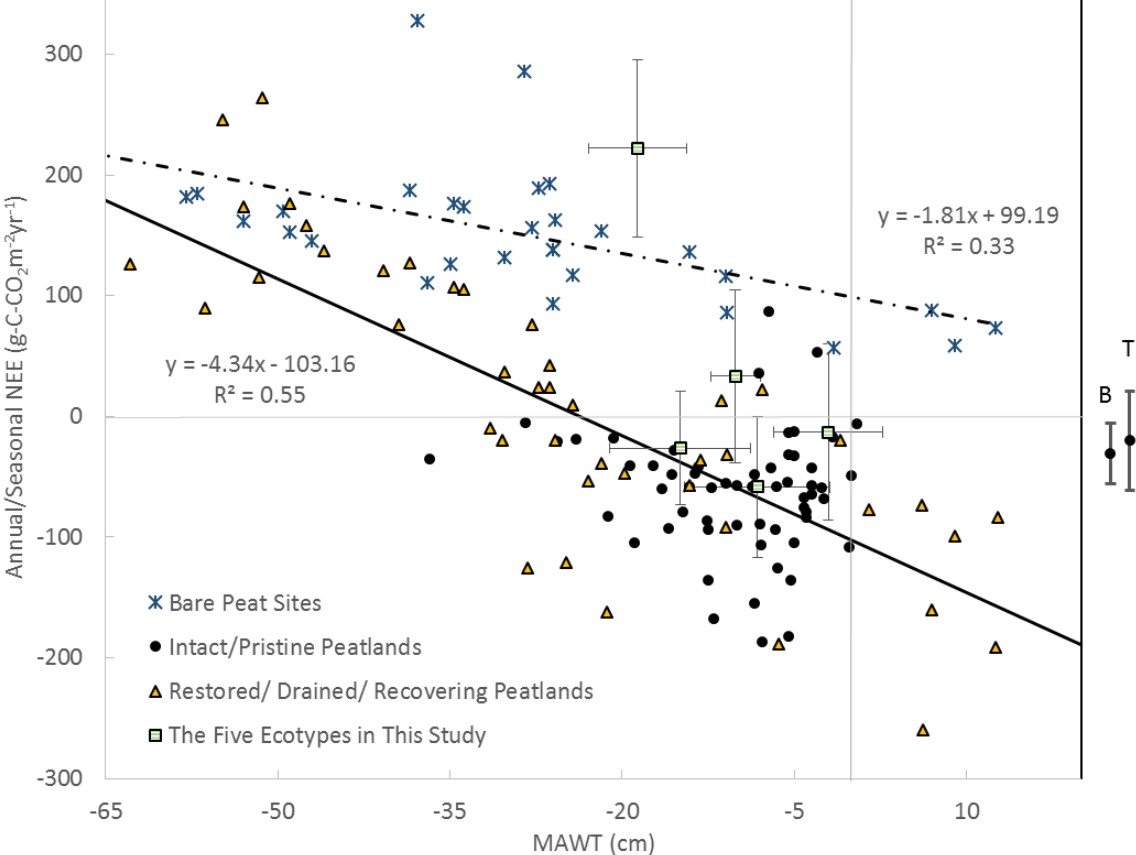

**Figure 11.** Mean annual water table vs. the annual NEE for the 5 ecotypes in this study (error bars are
standard deviation) compared to global studies from boreal and temperate peatlands. The solid line shows





the best fit linear trend line from all vegetated sites and the dashed line shows the best fit trend line for bare peat sites. (Data from: Wilson et al., 2015; Wilson et al., 2016; Vanslow-Algan et al., 2015; Tuittili et al., 1999; Waddington et al., 2010; Strack et al., 2014; Nilsson et al., 2008; Dinsmore et al., 2010; Koehler et al., 2011; Chimner et al., 2017; Gazovic et al., 2013; Lund et al., 2012; Levy and Grey et al., 2015; McVeigh et al., 2014; Helftler et al., 2015; Piechl et al., 2014; Stranchen et al., 2016; Roulet et al., 2007; Waddington and Roulet, 2000; for more details and additional studies see Supplemental Table S6 in Supplemental Section 3). Also, shown to the right of the figure is the mean and 95% CI NEE from nutrient poor, wet (MAWT >-30 cm) boreal (B) and temperate (T) peatlands (from the review paper, Wilson et al. 2016a).

There are a few cautionary notes that should accompany this plot. First, some of this data was
collected using the closed chamber method and some collected using eddy covariance methods.
Although both methods measure the same metric (NEE), closed chamber methods are inherently
micro-scale while eddy-covariance methods are inherently landscape scale, as are the water table
measurements accompanying them. Eddy-covariance measurements spatially integrate the micro-
variations within the landscape compared to closed chamber measurements, and much of the NEE
data reported in Figure 11 for intact peatlands is from eddy-covariance flux towers while there are
very few studies that have used this technique on degraded or recovering peatlands. This may
cause apparently higher variation in NEE for restored/recovering peatlands. Second, many of the
studies on boreal peatlands report only growing season NEE and water table because of frozen
winter conditions. Seasonal values from these studies are assumed to approximately represent
annual values because inter fluxes at boreal sites are probably of minor importance to the annual
fluxes. Third, this figure contains data points from different locations as well as the same location
over multiple years where data is available.

Similarly, annual/seasonal methane emissions are plotted against MAWT (Fig. 12). The data from
the ecotypes in this study fall well within the range of the $CH_4$ flux values in this compilation of data.
Reported methane emissions from drained peatlands are quite low and typically do not exceed 0.6
g C-$CH_4$ m$^{-2}$ yr$^{-1}$ when the mean annual water table is below -30 cm.



There is a high degree of variability in methane emissions in sites where the MAWT is higher than -
20 cm. Thus, a high MAWT seems to be a prerequisite for high methane emissions but does not
necessarily result in a high methane emissions, which agrees with Tiemeyer et al. (2017). As in
Wilson et al. (2016a), there does not seem to be a difference between restored and intact peatlands
for the $CH_4$ flux data presented in Figure 12, excluding infilled ditches, which can be hotspots for
methane emissions (Waddington and Day, 2000). For example, Cooper et al., (2014) reports 53.9 g
$C-CH_4 \, m^{-2} \, yr^{-1}$ for infilled ditches (Cooper et al. 2014). The low methane emissions from rewetted
bare peat soils suggests that the methanogenesis is limited by substrate availability in cutover
peatlands (Tuittila et al. 2000; Tuittila et al. 1999).





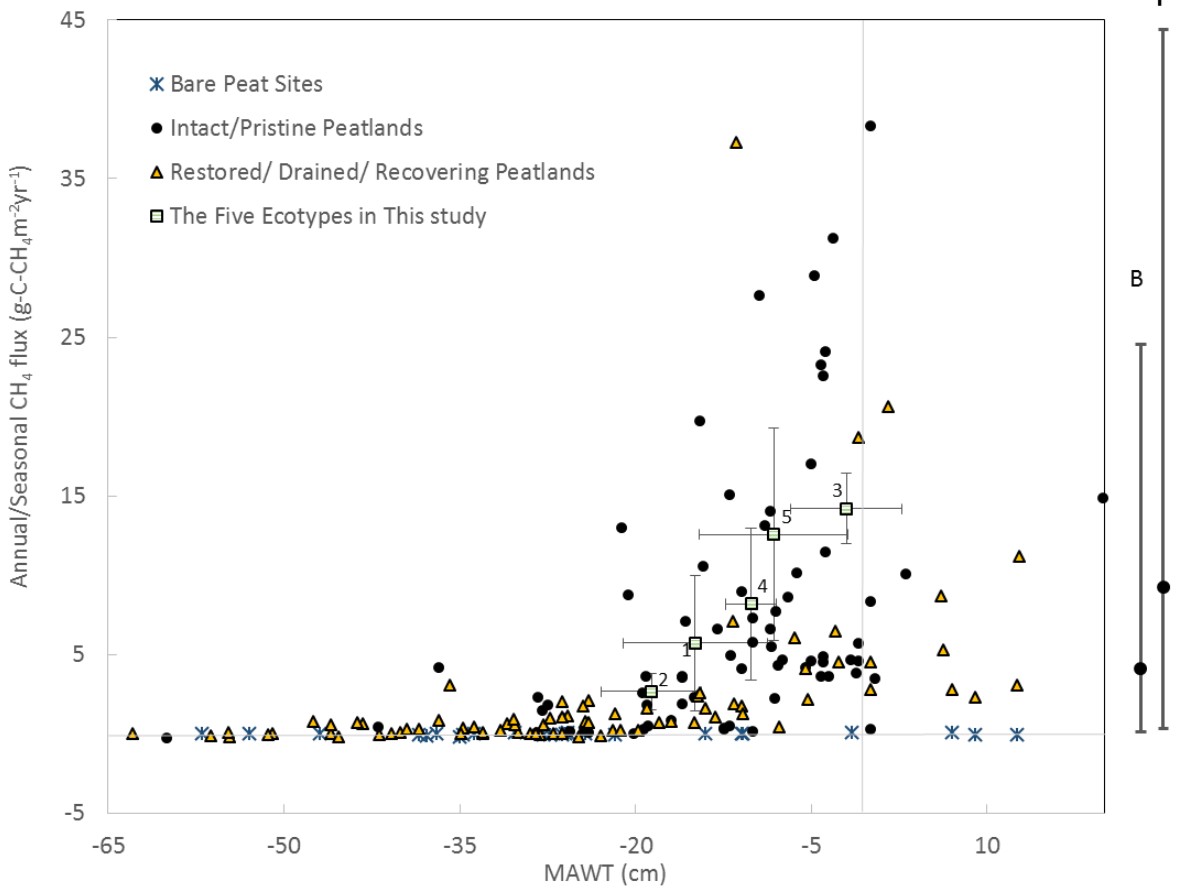

**Fig. 12.** This figure shows the mean annual water table plotted against the measured annual methane emissions for each ecotype for each ecotype in this study (error bars are standard deviations) and from global studies of temperate and boreal peatlands (Sources: Flessa et al. 1998, Fieldler et al. 1998, Wilson et al. 2016; Tuitili et al., 1999; Wilson et al., 2018; Danevic et al., 2010; Von Arnold et al., 2005; Laine et al., 1996; Yamulki et al., 2012; Nykanen et al., 1998; Fieldler et al., 2007; Cooper et al., 2014; Waddington and Day, 2007; Chimner et al., 2017; Waddington and Roulet 2000; for more details see Table S6 in Supplemental Section 3. Also, shown to the right of the figure is the mean and 95% CI of methane emissions from nutrient poor, wet (MAWT >-30 cm) boreal (B) and temperate (T) peatlands (from the review paper, Wilson et al. 2016a).

As with the NEE data in Figure 11, this figure contains both annual and seasonal fluxes, where
seasonal fluxes are more often reported for boreal sites. This figure excludes methane emissions
from infilled ditches. There are few studies that have reported methane emissions from bare peat
sites, and the results are generally low (mean of -0.03 g C-CH$_4$ m$^{-2}$ yr$^{-1}$) even at high water table. The



data used to compile Figs. 11 and 12 and additional studies can be found in Supplemental Section 3,
Table S6.

*4.3 Implications for Peatland Management and Restoration*
Peatland management and restoration is primarily able to alter 1) the hydrology, typically
managing the water table through drainage or drain blocking and 2) the plant ecology, through
revegetation efforts, managing water table, and controlling invasive species (Andersen et al. 2017).
If peatland management is used as a climate change mitigation tool [as suggested in Birkin et al.
(2011); Wilson et al. (2013); Leifeld and Menichetti, (2018)], the impact of these things must be
considered. The wide range of methane emissions reported in the literature at high water tables
means that generalizations cannot be made about GWP for restored vs. pristine peatlands or GWP
as a function of water table. For example, reported values of annual $CH_4$ fluxes in Fig. 12 from sites
with a MAWT above -10 cm range from 0.3 (Nykanen et al., 1995) to 38.3 g $C$-$CH_4$ $m^{-2}$ $yr^{-1}$
(calculated from Junkurst and Fieldler, 2007) for intact peatlands and 0.4 (Strack et al. 2014) to
20.6 g $C$-$CH_4$ $m^{-2}$ $yr^{-1}$ (Renou-Wilson et al. 2018a) for restored peatlands. This corresponds to a 100-
year GWP of 0.1 to 17.3 and 0.2 to 9.3 tonnes $CO_2$-eq $ha^{-1}$ $yr^{-1}$, respectively. This range is larger than
the largest reported $CO_2$ sink for intact peatlands of -6.9 tonnes $CO_2$-eq $ha^{-1}$ $yr^{-1}$ (calculated from
Levy and Grey 2015) and far larger than the average $CO_2$ sink for intact peatlands of -2.2 tonnes
$CO_2$-eq $ha^{-1}$ $yr^{-1}$ reported in Figure 11. Still, a GWP decrease is often observed following rewetting
(Wilson et al. 2016a, Wilson et al. 2016b, Renou-Wilson et al. 2018b). Additionally, the data from
this study would suggest that the presence of *Sphagnum spp.* corresponds to a decreased GWP.
Junkurst and Fielder (2007) conducted a review of $CO_2$ and $CH_4$ flux for boreal and temperate
peatlands. They state that the methane fluxes in temperate peatlands are "usually found to be three
orders of magnitude lower than simultaneously measured $CO_2$ emissions." Thus, they conclude that
the suppressed $CO_2$ emission from higher water table would outweigh the GWP effect of increased





methane emissions. This conclusion seems unlikely to be generally true based on the data shown
Figure 11 and 12.

There is some debate about the use of GWP as a metric for natural peatlands because this metric
focuses on a 100-year time window, which may not be appropriate. For example, to quote from
Evans et al. (2016), "as noted by Frolking et al. (2006), the long-term sequestration of $CO_2$ into
stable organic matter gradually outweighs the warming effect of $CH_4$, due to the shorter
atmospheric lifetime of the latter, so that natural peatlands exert a net cooling impact on the
atmosphere over longer periods." This means that the long term climate benefit of peatlands is
primarily controlled by NEE. However, this logic would only apply to restoration works if these
impact the eco-hydrological conditions on time scales >> 100 years.

The ecotypes of the uncut raised bog at Abbeyleix were mapped by Bord na Móna in 2009 just after
restoration works blocking surface drains and again in 2014. During this time the extent of Sub-
Central area increased by approx. 2.1 ha largely at the loss of the Sub-Marginal ecotype. Assuming
the values found in this study are representative of all years, the restoration works resulted in a
reduction of 7.0 ± 7.7 tonnes yr$^{-1}$ $CO_2$ although a smaller reduction (3.3 ± 7.6 tonnes yr$^{-1}$) of $CO_2$
equivalents. Additionally, there is a potential reduction in $CO_2$ emissions due to raising the water
table throughout the entire 108 ha of raised bog area. The change in water table from these
restoration works was not directly measured, but based on the typical depths of water in the
blocked drains, there was an estimated 10–40 cm rise in water table. For the 108 ha of raised bog
area, this could result in an additional reduction of 166–664 tonnes yr$^{-1}$ of $CO_2$ based on the trends
in Fig. 11. The impact of increased methane emissions in this case is probably minimal because the
majority (67%) of the raised bog area, although with a higher water table than previously, remains
as deeply drained ecotypes (Marginal or Facebank).



## 5. Conclusions

All the major components of the carbon balance were measured at several different ecotypes, on restored and cutover raised bog with different land use and degradation histories. Trends in annual NEE and $CH_4$ fluxes were observed with respect to both ecological and hydrological conditions. In particular, higher water level and intact *Sphagnum* vegetation seem to be related to higher carbon sink and lower GWP. The data from ecotypes in this study were compared to a large number of studies on boreal and temperate peatlands with respect to MAWT. In this broader comparison, negative trends were observed in NEE with respect to MAWT for both vegetated and bare peat sites, while $CH_4$ fluxes were more variable at high MAWT.

### Data availability

Much of the data on the various aspects of the annual carbon balance including all the data behind Fig. 6, Fig. 9, Fig. 10, Fig. 11, and Fig. 12 can be found in the supplemental material. All other data used in this study are archived by the authors and are available on request (swensonm@tcd.ie).

### Supplemental Information

*Section S1.* A description of the NEE and $CH_4$ flux models tested and the thought behind these models. Also, for each of the 29 collars in this study, the empirical fitting parameters, the $r^2$, and the standard deviation of the residuals is shown for the best GPP and ER models.

*Section S2.* Eco-hydrological conditions and carbon balance terms for all collars and both years of this study.

*Section S3.* Data collected from literature: measurements aspects of peatland greenhouse gas balance. This section includes the data behind Fig. 11 and Fig. 12 as well as other studies.



**Author contribution**


Michael Swenson collected and analyzed the majority of the field data and prepared the manuscript
with contributions from other co-authors. Shane Regan attained the grant award, determined the
field site location, and contributed to setting up the field equipment and measuring infrastructure.
Dirk Bremmers collected methane flux data in the field and analyzed gas samples in the lab. Jenna
Lawless collected field measurements of DIC and $CO_2$ evasion. Shane Regan, Matt Saunders and
Laurence Gill contributed technical advice and guidance throughout the project implementation
and manuscript writing stages.

**Competing interests**


The authors declare that they have no conflict of interest.

**Acknowledgements**


Environmental Protection Agency (Ireland) for funding the project (project ref: 2014-NC-MS-2);
Fernando Fernandez and Jim Ryan (National Parks and Wildlife Service, Ireland); Dr. Maria Strack
for providing the collar specific data of NEE, and $CH_4$ flux which are presented but not explicitly
reported in Strack et al., (2014) and were included in Fig. 11 and Fig. 12; Abbeyleix Bog Project,
LTD for endless encouragement and help; Trinity College lab technicians and support.

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
