# Peer review of "Title Page"

_Biogeosciences, 2018_

## Referee Comment (RC1) · Anonymous Referee #1 · 3 Oct 2018

What's the paper about: In this ms Swenson et al attempt to estimate C-balance of five managed raised bogs in Ireland. The five sites differ in their hydro-physical and ecological characteristics, as defined by the National Parks and Wildlife Servies of Ireland. The strength of their study is in their attempt to estimate total C-balance of each site based on measurements of various aspects of ecosystem carbon balance during the same season, including: ecosystem $CO_2$-gas flux, ecosystem $CH_4$-gas flux, aquatic fluxes of dissolved organic and inorganic carbon and $CO_2$ gas efflux from open water in drainage ditches. Furthermore, they provide a literature compilation and review of studies that have measured ecosystem $CO_2$ and methane flux in boreal and temperate, managed and intact, peatlands. They use the global synthesis to explore global patterns between the fluxes and mean annual water table. They discuss their sites

in light of the global patterns and argue that the most practical advice to policy makers on C-sequestration potential of restored peatlands should involve insights into the impact of water table and vegetation composition on C-fluxes - two aspects that are actually manageable at sites. Some caution should be taken when interpreting their annual sums of different component fluxes, as most are modelled based on point measurements throughout the year and in some cases models were used to extrapolate beyond the data range used to develop the empirical model! Lots of assumptions used in some cases.

Detailed Comments: 1. This is a bit lengthy ms, but given it has a synthesis review, maybe ok for length. Journal can decide if to have it split into ms a and b or one ms when published. ALternatively there may be enough material here to write two separate more focused ms: one using actual observed data, focusing on observed trends and one on estimated/modelled trends in light of the global synthesis work.

2. Title may be worth modifying to include "implications for restoration efforts" in it - or something along the way. Also use of "restored and cutover" is a bit confusing. It sounds like both sites are undergoing restoration, just for different amounts of time. Perhaps reword to be more clear: "raised bogs at different stages of restoration".

3. Keywords: may be add DIC, DOC, global synthesis, carbon balance

4. Abstract - you mention measuring along 5 different ecotypes, but only two are listed. Describe all 5 then. Also reword to include all 5 ecotype descriptions, avoid use of specific category name (ex. Sub-central) and just use general description in Abstract. Don't think that category name adds info for non-Irish readers or those unfamiliar with NPWS classifications.

5. line 105: unidentified acronym occurance? NECB.

6. lines 120-125. Objectives state two main research sites: 1960 cutover and 2009 cutover, but in Abstract and rest you discuss 5 sampling sites of different "eco-types".

[Figure]

So please adjust the wording to link your Objectives to the rest of your ms.

7. lines 142-145: Again there seems to be a mismatch in site description - here you say two sites were uncut and three cut, but in objectives on lines 120-122 it seems to suggest both sites were cut, just one left to recover since 1960 and the other site only since 2009. So please clarify your site descriptions. May be clarify that you have two main research sites and 5 sub-sites within them and how they all differ with respect to their management history. Also add reference to table 1 early in this paragraph.

8. lines 164 - you had trees on your sites? please include % cover in Table one or description. How large are they on average and species type, stem density.

9. LIne 167: collars were installed to represent ecolocigal variability - such as what?

10. lines 211: which light levels, not clear? Please list PPDF under which measured. Where measurements corrected for PAR reduction due to chamber transmittance reduction?

11. NEE modelling, section 2.5: modelled NEE to account for diurnal variability - based on what? did you measure diurnal variability in data? Line 243: PPFD used from which chamber? or outside? lines 245 and Table S3: when reporting stats it is insufficient to just present R2 values. Show also n-values and estimated coefficients. Also describe how many data points were used to fit the model and how many to test/validate the model fit? Did you do that?

12. Methane modelling: be very cautious in your "modelling" attempts and interpretations - you fit an empirical model to a small subset of observations and then attempt to use that to predict fluxes outside of the data range used to develop your empirical model. I don't think this is statistically sound, although I appreciate your attempt and sympathize with instrument malfunction. So highlight this limitation in your text and discussion. Line 265 - "collar average" probably should read "overall average from measurements at all collars".

13. Section 2.7, lines 278: - is that a valid assumption?

14. Lines 307-312: you discuss $CO_2$ evasion from drainage ditches, what about open-pond water on your bogs? Any present? what proportion? how much would the open ponds add to $CO_2$ efflux from open water?

15. Line 315: likely mean "DIC" instead of "DOC"?

16. LInes 321-322: Sentence makes no sense? what are you saying? "... from the sum of model error and error of input field variables"?

17. Lines 338: report st.dev on mean annual total from Ballyroan.

18. lines 340: 1978-2007 average taken from where? also report stdev on mean annual value.

19. Figure 4: hard to tell two blue colours apart, likewise for yellow and orange. suggest to change.

20. Figure 4 -7 - either show all modelled and observed fluxes like you do for CH4-fluxes in Figure 5a ( I WOULD HIGHLY RECOMMEND, OTHERWISE YOU SHOW TOO MUCH MODELLED RESULTS but little observed values used to derive the modelled values!) or as monthly bar plots or cumulative plots like you do for terrestrial CO2-fluxes in Fig 4. This is also probably where you can reduce your figure numbers. Fig 5b probably belongs to Supplement where you describe your model fitting.

21. Section 2.4 name - CO2-gas flux measurement, not GHG.

22. Aquatic carbon losses: Fig 7: how do you know point around 2 mg/L is an outlier for DIC to assume a constant flux throughout the year? Seems too few points to make such assumption.

23. Fig 8: what's WHB and EHB?

24. Section 3.4: not sufficient to report only p-values for statistical results, report also

associated n used, mean and st.dev. Lines 404-406: was spatial variability accounted for in your analysis?

25. Figure 9a: - units unclear gC-CO2? CO2-equivalent? so Fig a and b are the same, with a in g/m2/yr and b in tones/ha/yr? why include both?

26. line 416: "carbon flux" probably mean "CO2 flux", same comment on lines 424. lines 416-417: unclear - so you looked at environmental controls for each collar separately? why not together per eco-type?

27. Section 3.5 - again stats should show n, mean, stdev together with p-value, else meaningless.

28. lines 428-433: don't get this paragraph.

29. Fig. 10: What if you colour points by ecotype? would be nice to see how they fit on the scatter, if group or not. in plots a and d - units are in gC, so does this C include CH4-carbon? Legend says data was averaged over 2 years - why? was there no interannual variability?

30. Discussion: Lines 436-437: well, I would be cautious with such as statement, as nowhere in your paper do you present simultaneous measurements of all components of c-balance you meausured. If you do have observation days where you have all of the fluxes measured on the same day - those would be your key days to focus on and show when trying to figure-out contribution of each component flux to overall C-flux. Such comparison, even if only on few days would be more valuable than that gapfilled modelled comparisons to field-based research.

31. lines 449-452: so did Nilsson et al also take a single measurement in a year and assume DIC to be constant? If not, how many they took and how the differences in sampling between their study and yours impacts the results. Think of n-sampled.

32. LInes 458: lower than Dinsmore and Nilson by how much? make it easy on the reader, so less likely to flip back and forth. show both values or state % difference.

33. SECTION 4.2 - I THINK THIS BELONGS TO RESULTS - this section should be split with synthesis plots shown in results and their discussion left in Discussion. ALso how data was collected and filtered should be in methods.

34. lines 532: how dry is "too dry"? also Briones et al reference is missing.

35. Fig. 11 - add lines to legend

36. lines 568-574 discussion - so how many points were from EC-derived NEE and how many from chamber derived-NEE? how do they fit on your Fig 11/12. Is there a difference between the two? (ex. one method consistently lower, but trends same, or one falls on one end of trend and the other at the other end?

37. line 577: "inter" probably should read "winter"

38. Fig 12: you specify your data points with numbers 1-5. I assume that relates to your ecotypes - so which is which? add to fig description.

39. Lines 606: "managing water table" - that's repeat of point 1).

40. line 608: "... the impact of these things must be considered." which things? and their impact on what?

41. Lines 621-624, follow discussion about impact of Sphagnum presence on GWP. So what has the study of Junkurst and Fielder to do with Spagphum-GWP?

42: Lines 628-635 - unclear what you're trying to say. Are you attempting to say that peatlands lifespan is more than 100 years, so their assessment should be based on longer time scales? So how long?

43. Lines 649: what's Marginal and Facebank ecotype? references?

SUPPLEMENT:

S1. GPPmax is assumed constant throughout what? which metrics from Wilson et al 2007 where used? how many data points were used for model development at each

point? why biological zero is assumed at 0C? what reference you have for this? Label your equations consequitively and consistently. SHow your model comparisons, how good each fit is. Why model fit to all collars - so did you check they all behave the same? where are the results?

S2. Equation S10 - which one is that?

S3. Tables S1 and S2 - kind of useless statistically. Please add estimated parameter coefficient's stats - p and t-values, also model stats such as R2, n-observations.

S3. Table S3: p-values? n? F-stats? data ranges used to fit model? all needed to make sense of reported R2.

S4. methane modelling - what worries me most. you fit an empirical model to limited data range and then use that to extrapolate beyond that data range. Don't think that's good statistical practice. Be cautious of such things. If there's a physiologically based model to work with - try that instead. Was effect of Temp similar to that of ER - show? "Temporal variation in fluxes was extrapolated in this model" - what are you trying to say?

Also was there any model validation done for any of your modelling activities?

---

## Referee Comment (RC2) · Anonymous Referee #2 · 8 Oct 2018

General comments: The manuscript reports the results from a two year study at two peatlands in Ireland: an abandoned (but not rewetted) and a rewetted peatland. In both sites, a full carbon balance ($CO_2$, $CH_4$, DOC and DIC) was measured and calculated. The authors indicate that the abandoned site was a strong annual carbon source and that the rewetted site was a small carbon sink. The authors also compare their results with literature values (a very nice literature review is included in the Supplementary material). The manuscript is well written (although it would benefit from a spell-check and does feel a little long), tightly focused (except Discussion) and the results are clear. However, the Discussion section is disjointed and requires some surgery, and I also have some concerns in regard to the models used but this may just need clarification by the authors rather than any major reconstruction.

[Figure]

Specific comments:

L2 Add C after carbon and use thereafter in abstract

L2 Add methane before (CH4)

L2 Not necessary to add "losses"

L4 Harvested suggests a renewable fuel source. Peat removal for fuel is anything but sustainable. Please replace harvesting here (and throughout the manuscript) with either "extraction" or "mining".

L5 Please define what you mean by "historically abandoned"

L6/7 What do you mean by "high quality"?

L7 Calluna vulgaris

L14 Why upper cases for Temperate and Boreal?

L15 "...in this study and was..."

L18 Add C after carbon and use thereafter in the manuscript

L26 95% is very high – 80 to 85% is generally cited

L31 Throughout the manuscript, you use intact, natural, near-natural and pristine interchangeably - please select one and keep to it.

L54 Consider Fritz et al. (2011) New Phytologist. 190, 398-408.

L75 Please use primary source as reference; Myhre et al (2013) Climate Change 2013: The Physical Science Basis Contribution of Working Group I to the Fifth Assessment Report of the Intergovernmental Panel on Climate Change

L77 I would not be inclined to use specific data here; why Helfter et al. and not McVeigh et al. for instance?

**BGD**

L78 The CH4 values derived by Wilson et al (2016) include rewetted sites, so are not suitable here, however there are lots of CH4 studies you could cite instead, e.g. Laine et al. (2007) Plant and Soil. 299, 181-193; Green and Baird (2017) Mires and Peat. 19, Article 09.

L82 Change methane to CH4

L89 Bain et al (2011) is not in reference list

L90 Change methane to CH4

L101 Consider Barry et al. (2016) Aquatic Sciences. 78, 541-560

L109 "recovering" is a new term to me. Do you mean rehabilitated?

L129 What do you mean by "natural peatland area"? The site is obviously not natural and the surrounding landscape is mainly grassland and some forestry. Delete.

L130 See earlier comment regarding harvesting.

L132 Met station location? 1980-2010?

L134 1980s

L140 1970s and 1960s

Fig. 1 The quality of Fig. 1 is poor, although this might be due to the pdf. The legend on the elevation map is hard to determine.

Table 1. Check font sizes and spelling of Sphagnum and Eriophorum

L188 Where was the sensor located?

L196 Stainless steel collars – as written it appears as if you only had one

L197 Where was the water trough located?

L198 What does "constructed in house. . ." mean?

L207 and area, volume of collar/chamber?

L208 A constant temperature or a temperature similar to ambient temperature? The former could be 50C for example and fit your criteria but would be meaningless for gas flux calculations and subsequent modelling.

L213 State flux sign convention used in this study.

L214 describe criteria used for quality checking.

L212 How many samples?

L259-260 CH4 fluxes have a strong diurnal variability in some plant types.

L337 Unusual long-term dates; 1980-2010 more usual.

Fig. 2b Degree symbol missing on y axis.

Fig 5b The use of a 1:1 line would provide better information as to the performance of the model

Figs. 4,6, 9 and 10 Check spelling of plant names.

Section 3.4 Only use "significant" when related to statistical comparisons.

L407 GWP

L407 tonnes (and thereafter)

Fig. 10 Given that MAWT is used as a predictor variable in the models, these observations are not independent (especially as the collars were lumped together for modelling) and I am far from convinced as to their value in this manuscript.

L436 Consider Nugent et al (2018) Global Change Biology, https://doi.org/10.1111/gcb.14449

L477 1960s

L484/485 join the sentences

L490 five decades

Section 4.2 I am not sure of the value of this section. The manuscript is already quite long and this seems superfluous (especially given the extensive data set in the Supplementary). If it really must be kept, then it should be moved to the Results section and then discussed here.

L515 Natural or semi-natural = intact?

L539 What does "Restoration of high quality peatland ecology" mean?

L576 Not so: In Saarnio (2007) Boreal Environment Research. 12, 101-113, 15% of growing season flux is emitted in the non-growing season. This approach was also used by IPCC Wetlands Supplement (2014)

L577 What is "inter flux"?

L608/609 "The impact of these things.." is very vague.

L628 and also due the pulse effect.

L630 Why not quote Frolking directly?

Conclusion This reads as a summary not as a conclusion; what does your study mean for land managers, policy makers etc?

L668 "best models" = the ones you used?

Supplementary: Check spellings throughout

Tables S1 and S2 Please provide the standard error (SE) associated with each parameter estimate. Given the large number of parameters in the models, I would suspect that the SE will be very high and would invalidate your approach.

---

## Author Comment (AC1) · 26 Nov 2018

Reviewer 1 General Comments: What's the paper about: In this ms Swenson et al attempt to estimate C-balance of five managed raised bogs in Ireland. The five sites differ in their hydro-physical and ecological characteristics, as defined by the National Parks and Wildlife Servies of Ireland. The strength of their study is in their attempt to estimate total C-balance of each site based on measurements of various aspects of ecosystem carbon balance during the same season, including: ecosystem CO2-gas flux, ecosystem CH4-gas flux, aquatic fluxes of dissolved organic and inorganic carbon and CO2 gas efflux from open water in drainage ditches. Furthermore, they provide a literature compilation and review of studies that have measured ecosystem CO2 and methane flux in boreal and temperate, managed and intact, peatlands. They use the

global synthesis to explore global patterns between the fluxes and mean annual water table. They discuss their sites in light of the global patterns and argue that the most practical advice to policy makers on C-sequestration potential of restored peatlands should involve insights into the impact of water table and vegetation composition on C-fluxes - two aspects that are actually manageable at sites. Some caution should be taken when interpreting their annual sums of different component fluxes, as most are modelled based on point measurements throughout the year and in some cases models were used to extrapolate beyond the data range used to develop the empirical model! Lots of assumptions used in some cases.

Response: The authors agree that more cautionary notes should be given to extrapolating beyond data collection periods. Responses are below for the specific "assumptions" that have been pointed out in the detailed comments.

Manuscript changes: In general, the manuscript has been changed to clarify the study site description, which seems to have caused some confusion for Reviewer 1. Also, in line with Reviewer 2, the manuscript is a little too long. To address this issue, much of the discussion Section 4.2 has been trimmed down as it tends to be a bit discursive, and the first paragraph in discussion Section 4.1 has been shortened by presenting data comparisons with other studies in table format.

More caution has been given to extrapolating the modelled CH4 and CO2 flux data beyond the data collection periods. The request for more statistical information on the models and the apparent need for a clearer description of the modeling process have been included in the main body of the manuscript as well as in the supplemental section.

Reviewer 1 Detailed Comments: 1. This is a bit lengthy ms, but given it has a synthesis review, maybe ok for length. Journal can decide if to have it split into ms a and b or one ms when published. ALternatively there may be enough material here to write two separate more focused ms: one using actual observed data, focusing on observed

trends and one on estimated/modelled trends in light of the global synthesis work.

Response: Agreed that the manuscript is too long, and some parts can be shortened, particularly the discussion section 4.2, on comparisons to literature. It probably works better as one manuscript (if the Editor agrees).

Manuscript changes: As per comments from Reviewer 2, much of the global synthesis discussion will be cut from the text as the Figs. 11 and 12 can speak largely for themselves. This should help keep to a more focused discussion and shorten the manuscript as a whole.

R1 comment 2. Title may be worth modifying to include "implications for restoration efforts" in it - or something along the way. Also use of "restored and cutover" is a bit confusing. It sounds like both sites are undergoing restoration, just for different amounts of time. Perhaps reword to be more clear: "raised bogs at different stages of restoration".

Response: The cutover area is not really undergoing restoration as such. Thus, it doesn't make sense to include "different stages of restoration" in the title.

Manuscript changes: Title has been changed to: Carbon balance of a restored and cutover raised bog: Implications for restoration and comparison to global trends

R1 comment 3. Keywords: may be add DIC, DOC, global synthesis, carbon balance

Response: Good suggestions.

Manuscript changes: DIC, DOC and carbon balance will be added to the key words.

R1 comment 4. Abstract - you mention measuring along 5 different ecotypes, but only two are listed. Describe all 5 then. Also reword to include all 5 ecotype descriptions, avoid use of specific category name (ex. Sub-central) and just use general description in Abstract. Don't think that category name adds info for non-Irish readers or those unfamiliar with NPWS classifications.

Response: OK, this comment and comments 6 and 7 suggest that Reviewer 1 is getting confused on the site description. We agree that this needs to be clarified both in the abstract and the main body of the paper. However, the Reviewer is asking to include details of all five the ecotypes in the abstract, but then not to use the names. To fully describe the 5 different ecotypes, without using the names will be far too wordy for an abstract.

Manuscript changes: A much clearer and still concise description of the study site is included in the abstract, which should help with the Reviewer's confusion. Names of specific ecotypes have been removed from the abstract. All of the ecotypes have not been fully described individually, but this general description has been included: "were measured for five distinct ecotypes ranging from those with high quality peat forming vegetation down to communities indicative of degraded, drained conditons."

R1 comment 5. line 105: unidentified acronym occurrence? NECB.

Response: OK

Manuscript changes: "net ecosystem carbon balance" (NECB) has been added to define the acronym.

R1 comment 6. lines 120-125. Objectives state two main research sites: 1960 cutover and 2009 cutover, but in Abstract and rest you discuss 5 sampling sites of different "eco-types".

Response: Yes, the description of sampling locations and study site needs to be clarified. Five ecotypes were located in areas with two different site histories.

Manuscript changes: A clearer description of the ecotypes and site history has been included in both the abstract and the main body of the text. Also, in the words "... including detailed comparisons between five distinct ecotypes" has been added to the objectives.

R1 comment 7. lines 142-145: Again there seems to be a mismatch in site description

- here you say two sites were uncut and three cut, but in objectives on lines 120-122 it seems to suggest both sites were cut, just one left to recover since 1960 and the other site only since 2009. So please clarify your site descriptions. May be clarify that you have two main research sites and 5 sub-sites within them and how they all differ with respect to their management history. Also add reference to table 1 early in this paragraph.

Response: Agreed, this can be clarified as per the above comments. The term recently restored is used in the last paragraph of the Introduction, this might imply that the bog had been harvested (and has now being restored) whereas it just been drained in preparation for being cut (but not actually harvested). The site history needs to be clarified earlier on.

Manuscript changes: See previous comments.

R1 comment 8. lines 164 - you had trees on your sites? please include % cover in Table one or description. How large are they on average and species type, stem density

Response: Yes, there were trees on the site as a whole. For example, the weir catchment area contained "lightly forested drains along a bog road (<10%)" Line 276. However, all of the ecotypes were treeless. Line 164 reads "open areas, excluding any large trees". In this case, "excluding any large trees" is a qualifier describing the open areas i.e. there were no trees at our monitoring sites.

Manuscript changes: "excluding any large trees" is changed to "excluding any trees"

R1 comment 9. LIne 167: collars were installed to represent ecolocigal variability - such as what?

Response: Again, it seems that the site description needs to be clearer in the manuscript. Within the ecotype definitions, there can be variability for the specific species coverage.

Manuscript changes: See above changes to line 7.

R1 comment 10. lines 211: which light levels, not clear? Please list PPDF under which measured. Where measurements corrected for PAR reduction due to chamber transmittance reduction?

Response: The light sensor was located inside of the chamber during measurements, so there was no need to correct for chamber transmittance as the light level was directly measured.

Manuscript changes: The phrases "generally under full ambient light, 1-2 light other partial shading light levels, and a completely shaded measurement" and "located inside the chamber" have been added to the manuscript.

R1 comment 11. NEE modelling, section 2.5: modelled NEE to account for diurnal variability – based on what? did you measure diurnal variability in data? Line 243: PPFD used from which chamber? or outside? lines 245 and Table S3: when reporting stats it is insufficient to just present r2 values. Show also n-values and estimated coefficients. Also describe how many data points were used to fit the model and how many to test/validate the model fit? Did you do that?

Response: The gross primary production (that is CO2 uptake by photosynthesis) is the largest component of net ecosystem exchange. In practically every system, the gross primary production is strongly controlled by the light intensity, which obviously has diurnal variations. Thus, the hourly light intensity (measured in the field at the weather station) was input into models of NEE, which results in expected diurnal fluctuations of the modelled results. Measurement of light intensity is described in Lines 186–191 of the manuscript.

All of the field data was used to calibrate the models. Validation was not done explicitly.

Manuscript changes: Further, statistical information is included in Table S3. Supplemental section S1 has been revised to show a clearer description of the modeling process, including (among other things) a description of the number of data points used

to fit the models.

R1 comment 12. Methane modelling: be very cautious in your "modelling" attempts and interpretations - you fit an empirical model to a small subset of observations and then attempt to use that to predict fluxes outside of the data range used to develop your empirical model. I don't think this is statistically sound, although I appreciate your attempt and sympathize with instrument malfunction. So highlight this limitation in your text and discussion. Line 265 - "collar average" probably should read "overall average from measurements at all collars".

Response: This is a good point. More cautionary notes should be included in describing the limitations of this data. All of the methane field measurements were used rather than "a small subset." "Collar average" is the correct terms here.

Manuscript changes: More cautionary notes have been included in describing this data throughout the manuscript, particularly the assumption that methane flux was the same both years even though it was only measured one of the years.

R1 comment 13. Section 2.7, lines 278: - is that a valid assumption?

Response: This is a big assumption, that the aquatic carbon losses were the same for each of the ecotypes, but it is an assumption that we had to make due to resource limitation. (It would be an interesting topic for a future research project.) Many studies on peatland carbon balance balance have made bigger assumptions about the magnitude of the DOC losses. For example, Wilson et al., 2016b estimate DOC losses from a peatland "based on guidance provided by IPCC" rather than field measurements. In this study, the DOC/DIC flux was at least measured directly on-site.

Manuscript changes: None.

R1 comment 14. Lines 307-312: you discuss $CO_2$ evasion from drainage ditches, what about openpond water on your bogs? Any present? what proportion? how much would the open ponds add to $CO_2$ efflux from open water?

Response: These are valid questions, but in this case, the only open water areas at the study site were associated with drainage ditches.

Manuscript changes: None.

R1 comment 15. Line 315: likely mean "DIC" instead of "DOC"?

Response: No, this is correct. Here the DOC flux is calculated differently for the carbon balance of the system, which includes all carbon losses from the catchment area and for the global warming potential, which is the greenhouse gas effect. This is because some of the DOC lost from the system may be stored in longer term sediment and not contribute to the GWP.

Manuscript changes: Line 313 to 315 has been clarified.

R1 comment 16. LInes 321-322: Sentence makes no sense? what are you saying? "... from the sum of model error and error of input field variables"?

Response: This sentence is describing the NEE model statistics.

Manuscript changes: This sentence has been re-worded for clarity.

R1 comment 17. Lines 338: report st.dev on mean annual total from Ballyroan.

Response: OK

Manuscript changes: These details have been added to the manuscript.

R1 comment 18. lines 340: 1978-2007 average taken from where? also report stdev on mean annual value.

Response: OK.

Manuscript changes: The weather station details have been added as well as stdev of weather station data.

R1 comment 19. Figure 4: hard to tell two blue colours apart, likewise for yellow and

orange. suggest to change.

Response: OK

Manuscript changes: The colors have been changed in Figure 4 to improve readability.

R1 comment 20. Figure 4 -7 - either show all modelled and observed fluxes like you do for CH4- fluxes in Figure 5a ( IWOULD HIGHLY RECOMMEND, OTHERWISE YOU SHOWTOO MUCH MODELLED RESULTS but little observed values used to derive the modelled values!) or as monthly bar plots or cumulative plots like you do for terrestrial CO2- fluxes in Fig 4. This is also probably where you can reduce your figure numbers. Fig 5b probably belongs to Supplement where you describe your model fitting.

Response: Moving Fig. 5b to the supplemental section is a good suggestion. Otherwise, this comment is a little perplexing: on the one hand, Reviewer 1 mentions reducing the number of figures; on the other hand, Reviewer 1 asks for more figures showing modelled and measured results. As the GPP and ER were modelled separately for each of the 29 collars, it would be confusing/misleading to show the entire field dataset and modelled results in a single plot. The modelled and measured data is shown together in the attached figures for one particular collar as an example. For all 29 collars, 77 such plots (3 for each collar) would be needed, which may be excessive even for the supplemental section (although if the Editor thinks this is valuable, these plots can be included). The r2 values, although not statistically sufficient by themselves, demonstrate the modelled data correlation with field data in a much smaller format.

"like you do for CH4- fluxes" Showing all of the modelled and measured CH4 flux data was possible because the field data from all collars was lumped together before modelling.

Manuscript changes: Fig 5b has been moved to supplemental section. Otherwise, no changes will be made unless the Editor requests plots like the example above to be in

the supplemental section for each collar.

R1 comment 21. Section 2.4 name - CO2-gas flux measurement, not GHG.

Response: Yes, that is better.

Manuscript changes: The section heading name changed to "CO2 and CH4 gas flux measurements"

R1 comment 22. Aquatic carbon losses: Fig 7: how do you know point around 2 mg/L is an outlier for DIC to assume a constant flux throughout the year? Seems too few points to make such assumption.

Response: This is a fair point, but it is just an assumption and has a very minor effect on the overall results, as this is the smallest component of the C balance. Based on Dixon's Q test, this point can be excluded as an outlier to 95% confidence.

Manuscript changes: None.

R1 comment 23. Fig 8: what's WHB and EHB?

Response: These points need to be labeled on Fig.1

Manuscript changes: These points have been labeled on Fig.1

R1 comment 24. Section 3.4: not sufficient to report only p-values for statistical results, report also associated n used, mean and st.dev. Lines 404-406: was spatial variability accounted for in your analysis?

Response: OK, on the stats. It is not clear what is meant by spatial variability; this is exactly what is being described here i.e. difference between collars within the same ecotype.

Manuscript changes: Additional statistical information added throughout this section.

R1 comment 25. Figure 9a: - units unclear gC-CO2? CO2-equivalent? so Fig a and b are the same, with a in g/m2/yr and b in tones/ha/yr? why include both?

Response: The carbon balance and the global warming potential (GWP) as shown in Fig. 9a and 9b, respectively, are fundamentally different quantities with correspondingly different units. Although the change in units may be confusing, the units chosen are standard units for reporting these types of measurements.

Manuscript changes: None.

R1 comment 26. line 416: "carbon flux" probably mean "CO2 flux", same comment on lines 424. lines 416-417: unclear - so you looked at environmental controls for each collar separately? why not together per eco-type?

Response: The phrase "carbon flux" is not used. The phrase "carbon balance" is the correct wording here and refers to multiple aspects of the carbon balance, including CO2 flux. The data are presented together by ecotype in Fig. 4, Fig. 6, and Fig. 9. As data (flux & divers) is available in detail at each collar, there is no reason not to include these comparisons.

Manuscript changes: The section has been changed to describe the patterns in collar CO2 flux data instead of patterns in carbon balance data, to reduce confusion.

R1 comment 27. Section 3.5 - again stats should show n, mean, stdev together with p-value, else meaningless.

Response: This is a good suggestion.

Manuscript changes: The additional statistical information is included throughout this section.

R1 comment 28. lines 428-433: don't get this paragraph.

Response: The percent genus cover within the collars is compared to the GWP, C-balance, and CH4 flux.

Manuscript changes: This paragraph has been re-written to clarify.

R1 comment 29. Fig. 10: What if you colour points by ecotype? would be nice to see how they fit on the scatter, if group or not. in plots a and d - units are in gC, so does this C include CH4-carbon? Legend says data was averaged over 2 years - why? was there no interannual variability?

Response: Yes, good idea to colour by ecotype (and would also address point 26 somewhat). Data was averaged over the two years for clarity in the plots because these plots are focused on spatial rather than temporal variability; the longest data set available was included. Yes, in the first draft of the manuscript the carbon balance includes all aspects of the carbon balance. However, plots a and d of this figure have been changed in the revised manuscript to include only CO2 flux (NEE) rather than carbon balance.

Manuscript changes: The collars were coded by ecotype in this figure as per the Reviewer's suggestion. Plots a and d of figure 10 have been changed in the revised manuscript to include only CO2 flux (NEE) rather than carbon balance.

R1 comment 30. Discussion: Lines 436-437: well, I would be cautious with such as statement, as nowhere in your paper do you present simultaneous measurements of all components of c-balance you meausred. If you do have observation days where you have all of the fluxes measured on the same day - those would be your key days to focus on and show when trying to figure-out contribution of each component flux to overall C-flux. Such comparison, even if only on few days would be more valuable than that gapfilled modelled comparisons to field-based research.

Response: Being cautious about this statement is a fair point with, as the word "simultaneously" is a little misleading. This line could be re-stated to be more precise. However, adding in description about specific days where all the fluxes were measured together with all of the other data (flows etc.) would probably not help the structure of the paper (and would make it a lot longer).

Manuscript changes: The word "simultaneously" has been removed from this paragraph. Instead, this line has been changed to "annual fluxes of all major aspects of the carbon balance have been quantified over the same 2 year period".

R1 comment 31. lines 449-452: so did Nilsson et al also take a single measurement in a year and assume DIC to be constant? If not, how many they took and how the differences in sampling between their study and yours impacts the results. Think of n-sampled.

Response: Agreed that more discussion is needed on the causes of differences between studies although it is unlikely that the differences observed are due primarily to the number of samples.

Manuscript changes: Further discussion is included on the differences between these study sites and results. See also to response to comment 32 below.

R1 comment 32. LInes 458: lower than Dinsmore and Nilson by how much? make it easy on the reader, so less likely to flip back and forth. show both values or state % difference. Response: Yes, this paragraph is a bit burdensome with a lot of data comparisons. The comparison between studies may be easier to read in table format. Then, this paragraph can be focused more on the differences in methods, etc. causing the variation in results (Partially, addressing Comment 31).

Manuscript changes: A table of the various components of the carbon balance from these other two studies is included in Section 4.1.

R1 comment 33. SECTION 4.2 - I THINK THIS BELONGS TO RESULTS - this section should be split with synthesis plots shown in results and their discussion left in Discussion. ALso how data was collected and filtered should be in methods.

Response: That is a good suggestion and agrees with Reviewer 2.

Manuscript changes: The figures 11 and 12 have been moved to the results section and relevant pieces have been moved to the methods section or kept in the discussion. On the whole, the discussion section here has been shortened as per response to

Reviewer 2.

R1 comment 34. lines 532: how dry is "too dry"? also Briones et al reference is missing.

Response: The reader can go to the relevant reference if interested in this question. The details on this point not too relevant to the current manuscript.

Manuscript changes: This reference has been added to the references section.

R1 comment 35. Fig. 11 - add lines to legend

Response: OK

Manuscript changes: Lines were added to the legend in Fig. 11.

R1 comment 36. lines 568-574 discussion - so how many points were from EC-derived NEE and how many from chamber derived-NEE? how do they fit on your Fig 11/12. Is there a difference between the two? (ex. one method consistently lower, but trends same, or one falls on one end of trend and the other at the other end?

Response: The primary difference between the use of these two methods is based on the land use type as described in the text. The curious reader can pull out this information in the Supplemental tables, but this plot with numerous symbols already, may become overcrowded with this information.

Manuscript changes: None.

R1 comment 37. line 577: "inter" probably should read "winter"

Response: Yes.

Manuscript changes: "inter" changed to "winter"

R1 comment 38. Fig 12: you specify your data points with numbers 1-5. I assume that relates to your ecotypes - so which is which? add to fig description.

Response: Yes, this needs to be added.

Manuscript changes: The numbers 1- 5 have been specified in the figure caption.

R1 comment 39. Lines 606: "managing water table" - that's repeat of point 1).

Response: That is a fair point.

Manuscript changes: The words "managing water table" were deleted from line 606.

R1 comment 40. line 608: "... the impact of these things must be considered." which things? and their impact on what?

Response: Yes, this line is "vague" as per Reviewer 2, and may not be necessary to include.

Manuscript changes: This line has been removed from the manuscript.

R1 comment 41. Lines 621-624, follow discussion about impact of Sphagnum presence on GWP. So what has the study of Junkurst and Fielder to do with Spagphum-GWP?

Response: OK, further discussion can be included in the Sphagnum effect on GWP. Junkurst and Fielder is a little off topic here. This reference may be removed.

Manuscript changes: As per above responses, the entire discussion section has been streamlined and clarified.

R1 comment 42: Lines 628-635 - unclear what you're trying to say. Are you attempting to say that peatlands lifespan is more than 100 years, so their assessment should be based on longer time scales? So how long?

Response: This sentence needs to be clarified, but is important. Peatland preservation is beneficial (in terms of greenhouse gasses) despite methane emissions because the long term sequestration and storage of carbon outweighs methane emissions. However, for peatland restoration, the greenhouse gas impact depends on the time scale

that restoration works effect the eco-hydrological trajectory. For example, if restoration works only impact the short term (decadal) eco-hydrological trajectory, then methane emissions may be proportionally more important to consider for the overall greenhouse gas budget.

Manuscript changes: These lines have been clarified.

R1 comment 43. Lines 649: what's Marginal and Facebank ecotype? references?

Response: Yes, this is unclear.

Manuscript changes: This sentence is re-written to remove the undefined and Irish specific terms "facebank" and "marginal".

R1 comment S1. GPPmax is assumed constant throughout what? which metrics from Wilson et al 2007 where used? how many data points were used for model development at each point? why biological zero is assumed at 0C? what reference you have for this? Label your equations consequitively and consistently. SHow your model comparisons, how good each fit is. Why model fit to all collars - so did you check they all behave the same? where are the results?

Response: Section S1 describes the modelling process. The number of questions surrounding this section suggests that this entire section could be clarified. Yes, some assumptions were made. However, not too many major assumptions were made because several different empirical models were fit to the field data, and then, these models were compared. That is, the model choices were data driven. The actual biological zero has a minor impact on the model results because of the nature of the temperature effect in these models; a biological zero of 0 C for GPP in bogs is supported by the observations of Peichl et al., 2014. Why model fit to all collars - so did you check they all behave the same? Quite the opposite, I did not assume that all collars behave the same way because of ecological and hydrological differences between collars. Thus, empirical models were fit to the field $CO_2$ flux data from each of the 29

collars individually.

Manuscript changes: This section has been re-written and clarified. where are the results? See response to Comment 20. More field data could be included at the Editor's request as described above.

R1 comment S2. Equation S10 - which one is that?

Response: This section has been re-written and clarified as per above comment.

Manuscript changes: As above.

R1 comment S3. Tables S1 and S2 - kind of useless statistically. Please add estimated parameter coefficient's stats - p and t-values, also model stats such as r2, n-observations.

Response: That is fine.

Manuscript changes: More statistical information has been included in these tables.

R1 comment S3. Table S3: p-values? n? F-stats? data ranges used to fit model? all needed to make sense of reported r2.

Response: That is a good suggestion.

Manuscript changes: More statistical information will be included in Table S3.

R1 comment S4. methane modelling - what worries me most. you fit an empirical model to limited data range and then use that to extrapolate beyond that data range. Don't think that's good statistical practice. Be cautious of such things. If there's a physiologically based model to work with - try that instead. Was effect of Temp similar to that of ER - show? "Temporal variation in fluxes was extrapolated in this model" - what are you trying to say?

Response: This is a good point that more caution needs to be added around the model results. Though, perhaps the purpose of modelling methane fluxes for this study needs

to be clarified a bit more in the paper as well. The purpose of this model was not to predict the methane flux at a particular point in time. Rather, the purpose of this model was to estimate annual methane fluxes. Essentially the average methane flux at each collar was scaled by a factor to account for the fact that the field data was not collected over the entire year. The model was used to predict that scaling factor. Further, the major hole in this data is that methane measurements were only collected during 1 of 2 years. This limitation in the data needs to be highlighted more explicitly in the manuscript. The assumption that the methane flux was the same both years is partially verified by the fact that the model gave very similar results for both years. This is the other purpose of this model.

"limited data range" or "beyond the data range" This phrase is used several times by the Reviewer; I am not entirely sure of the meaning. If the Reviewer is referring to a limited data collection period, than this is a fair comment. However, if the Reviewer is referring to a limited data range, than, it seems that the reviewer is mistaken. The modelled results were not extended beyond the measured data range.

I am not aware of an appropriate physiologically based model to use in this case because methane fluxes from peatlands can be highly variable both within and between sites.

"Temporal variation in fluxes was extrapolated in this model- what are you trying to say?" Yes, this needs to be clarified.

Manuscript changes: More caution needs to be applied when presenting methane modelling results, particularly for 2016 when no field data was collected. Also, the purpose methane modeling has been clarified in the manuscript.

R1 comment Also was there any model validation done for any of your modelling activities?

Response: No, in this case, all of the field data was used in calibrating the models.

Manuscript changes: None.

Please also note the supplement to this comment:
https://www.biogeosciences-discuss.net/bg-2018-350/bg-2018-350-AC1-
supplement.pdf

[Figure]

[Figure]

[Figure]

**Fig. 1.** Figure caption. This figure shows the modelled and measured data for collar EC13 for ER (top), GPP (middle), and for clarity (because GPP drops to 0 every night with light level), GPP normalized by li

---

## Author Comment (AC2) · 26 Nov 2018

Author response to Reviewer 2 comment on "Carbon balance of a restored and cutover raised bog: Comparison to global trends" by Michael M. Swenson et al.

R2 General comments: The manuscript reports the results from a two year study at two peatlands in Ireland: an abandoned (but not rewetted) and a rewetted peatland. In both sites, a full carbon balance ($CO_2$, $CH_4$, DOC and DIC) was measured and calculated. The authors indicate that the abandoned site was a strong annual carbon source and that the rewetted site was a small carbon sink. The authors also compare their results with literature values (a very nice literature review is included in the Supplementary material). The manuscript is well written (although it would benefit from a spell-check

and does feel a little long), tightly focused (except Discussion) and the results are clear. However, the Discussion section is disjointed and requires some surgery, and I also have some concerns in regard to the models used but this may just need clarification by the authors rather than any major reconstruction.

Response: The Reviewer comments that the paper is "well written" but "little long" and the Discussion section is "disjointed and requires some surgery." I agree with this comment in general. After a fresh read-through, I think that the discussion section gets a little off topic from the study results by including lengthy discussions of data from literature. The comparison to literature data is valuable, which includes Figures 11 and 12 and the extensive tables of literature data in the Supplemental Section 3. However, the discussion text surrounding these figures can be greatly shortened. The Reviewer comments, "I also have some concerns in regard to the models used but this may just need clarification by the authors rather than any major reconstruction." The only "concern" raised explicitly is the number of fitting parameters (an in comment below on Tables S1 and S2). The Reviewer raises a valid point for the ecosystem respiration model, in particular, which arguably had too many empirical fitting parameters. This model has been replaced by a simpler model with fewer fitting parameters, as described in more detail in the response to the comment on Table S1 and S2. This change had a minor effect on the results and conclusions of the paper. For "clarification," the Reviewer requests that the SE of model fitting parameters be included. Additional statistical information on the modelling has been included in the Tables S1 and S2.

Manuscript changes: The discussion section has been shortened and streamlined by cutting out much of the text describing the comparison to literature. The Figures 11 and 12 will be left in the body of the text, but moved to the results (as in comment on Section 4.2, below).

The modelling of ecosystem respiration has been redone using a simpler model, which was taken directly from Wilson et al., 2016b.

R2 COMMENT: Specific comments:

R2 COMMENT: L2 "Add C after carbon and use thereafter in abstract" "Add methane before (CH4)" "Not necessary to add "losses""

Response: These are good suggestions.

Manuscript changes: The recommended changes will be made in the manuscript abstract.

R2 COMMENT: L4 Harvested suggests a renewable fuel source. Peat removal for fuel is anything but sustainable. Please replace harvesting here (and throughout the manuscript) with either "extraction" or "mining".

Response: OK, yes, this is a good suggestion.

Manuscript changes: The term "harvested" has been changed to "mined" in the manuscript when in reference to peatlands impacted by peat extraction. R2 COMMENT: L5 Please define what you mean by "historically abandoned"

Response: OK, this is potentially confusing.

Manuscript changes: "historically" has been changed to "abandoned x years ago"

R2 COMMENT: L6/7 What do you mean by "high quality"?

Response: That is a good question; high quality is referring to the site most similar to the ecology and hydrology of undisturbed open raised bog habitat in Ireland.

Manuscript changes: As per Reviewer 1's comments, the site description has been clarified in the abstract.

R2 COMMENT: L7 Calluna vulgaris

Response: The Calluna vulgaris refers to a species of plant. In the manuscript the Calluna Cutover refers to specific ecotype.

Manuscript changes: The named of specific ecotypes have been removed from the abstract to reduce confusion, as per Comment 1 from Reviewer 1.

R2 COMMENT: L14 Why upper cases for Temperate and Boreal?

Response: Indeed.

Manuscript changes: Temperate and boreal will be changed to lower case throughout the manuscript where appropriate.

R2 COMMENT: L15 ": : :in this study and was: : :"

Response: Ah, there is an extra word in this line.

Manuscript changes: The word "was" has been removed from this line.

R2 COMMENT: L18 Add C after carbon and use thereafter in the manuscript

Response: Agreed.

Manuscript changes: The suggested change has been applied to the manuscript.

R2 COMMENT: L26 95% is very high – 80 to 85% is generally cited

Response: OK, 95% is referring to the percent of raised bogs that have been degraded. 80-85% refers to the overall peatlands in Ireland. The way that the sentence currently reads is thus incorrect.

Manuscript changes: This line has been clarified to refer more specifically to raised bogs.

R2 COMMENT: L31 Throughout the manuscript, you use intact, natural, near-natural and pristine interchangeably - please select one and keep to it.

Response: This is a good suggestion.

Manuscript changes: The term "intact" will be used in reference to peatlands which have not been mined or drained. The term "natural" will be used in reference to those

peatlands which are not actively being used for agriculture, intensive grazing, mining, forestry, etc.

R2 COMMENT: L54 Consider Fritz et al. (2011) New Phytologist. 190, 398-408.

Response: I was not aware of this publication. The findings from this publication are somewhat contradictory with this line of the introduction.

Manuscript changes: This line has been changed to include the findings from Fitz et al., 2011.

R2 COMMENT: L75 Please use primary source as reference; Myhre et al (2013) Climate Change 2013: The Physical Science Basis Contribution of Working Group I to the Fifth Assessment Report of the Intergovernmental Panel on Climate Change

Response: Yes, good point.

Manuscript Changes: The suggested primary reference is used here.

R2 COMMENT: L77 I would not be inclined to use specific data here; why Helfter et al. and not McVeigh et al. for instance?

Response: Including specific data here is helpful for putting the present work in context. Helftler et al. is used as a source because they include a table of reported literature values.

Manuscript changes: The use of this source is clarified by adding "literature compilation from Helftler..."

R2 COMMENT: L78 The CH4 values derived by Wilson et al (2016) include rewetted sites, so are not suitable here, however there are lots of CH4 studies you could cite instead, e.g. Laine et al. (2007) Plant and Soil. 299, 181-193; Green and Baird (2017) Mires and Peat. 19, Article 09.

Response: That is a good point.

Manuscript changes: The suggested citations have been used here.

R2 COMMENT: L82 Change methane to CH4; L90 Change methane to CH4

Response: OK

Manuscript changes: The use of "methane" has been replaced by the chemical symbol throughout the manuscript after initially defining it.

R2 COMMENT: L89 Bain et al (2011) is not in reference list

Response: OK

Manuscript changes: This citation has been added to the reference list.

R2 COMMENT: L101 Consider Barry et al. (2016) Aquatic Sciences. 78, 541-560

Response: OK

Manuscript changes: This article has added as a reference here.

R2 COMMENT: L109 "recovering" is a new term to me. Do you mean rehabilitated?

Response: Recovering means bogs which have had no definite action taken to rehabilitate them. They have just stopped being mined and left. So, the term "rehabilitated" is not correct.

Manuscript changes: In this line "abandoned" will be used in place of "recovering". In the manuscript, this term has now been defined as the "'spontaneous revegetation of mined peatlands' (From Poulin et al., 2005), which have had no definite action taken to rehabilitate them."

R2 COMMENT: L129 What do you mean by "natural peatland area"? The site is obviously not natural and the surrounding landscape is mainly grassland and some forestry. Delete.

Response: OK, The site is a natural area with a variety of plant communities some of

which is peatland in various degrees of degradation.

Manuscript changes: The wording has been changed from "natural peatland area" to "peatland and natural area".

R2 COMMENT: L130 See earlier comment regarding harvesting.

Response: As in above comment.

Manuscript changes: As in above comment.

R2 COMMENT: L132 Met station location? 1980-2010?

Response: The unusual dates are due to the available data from the met station.

Manuscript changes: The met station location has been added to the manuscript.

R2 COMMENT: L134 1980s; L140 1970s and 1960s

Response: OK

Manuscript changes: The apostrophe has been removed from all decade numbers.

R2 COMMENT: Fig. 1 The quality of Fig. 1 is poor, although this might be due to the pdf. The legend on the elevation map is hard to determine.

Response: Noted.

Manuscript changes: A higher resolution figure will be used for the final submission.

R2 COMMENT: Table 1. Check font sizes and spelling of Sphagnum and Eriophorum

Response: OK, good catch.

Manuscript changes: These typo mistakes have been corrected.

R2 COMMENT: L188 Where was the sensor located?

Response: Sensor was located inside of the chamber.

Manuscript changes: This detail is added to the manuscript.

R2 COMMENT: L196 Stainless steel collars – as written it appears as if you only had one

Response: Agreed.

Manuscript changes: "collars" instead of "collar"

R2 COMMENT: L197 Where was the water trough located?

Response: Along the top edge.

Manuscript changes: Added "along the top edge" to this sentence

R2 COMMENT: L198 What does "constructed in house: : :" mean?

Response: This phrase means that the chambers were built by the authors and department technicians.

Manuscript changes: The phrase "in house" was changed to "in-house" as it should be.

R2 COMMENT: L207 and area, volume of collar/chamber?

Response: The dimensions of the chamber are given above in line 198, but the area and volume can be stated more explicitly.

Manuscript changes: This information has been added to the manuscript.

R2 COMMENT: L208 A constant temperature or a temperature similar to ambient temperature? The former could be 50C for example and fit your criteria but would be meaningless for gas flux calculations and subsequent modelling.

Response: Reviewer 2 seems to have mis-understood the meaning of this sentence. The temperature was kept constant over the chamber closure equal to the initial ambient temperature.

Manuscript changes: The phrase "over the chamber closure time" was added to the

sentence as a clarifier.

R2 COMMENT: L213 State flux sign convention used in this study.

Response: Yes, that needs to be included.

Manuscript changes: The following sentence was added after line 213 "For this study, a positive sign convention is indicates a net loss of carbon to the peatland."

R2 COMMENT: L214 describe criteria used for quality checking.

Response: This data was quality checked to ensure that the change in $CO_2$ concentration over the chamber closure was monotonic, and physical parameters such as temperature and PPFD did not change substantially over the closure.

Manuscript changes: The following sentence describing quality checkin criteria was added to the manuscript: "...quality checked to ensure that the change in $CO_2$ concentration over the chamber closure was monotonic and that the PPFD did not change by more than 50 $\mu$mol m-2 s-1 over the chamber closure."

R2 COMMENT: L212 How many samples?

Response: "...generally under full ambient light, 1-2 light other partial shading light levels, and a completely shaded measurement."

Manuscript changes: The sentence in the response was added to the manuscript.

R2 COMMENT: L259-260 CH4 fluxes have a strong diurnal variability in some plant types.

Response: The data from Pypker et al., 2013 supports the claim of a low diurnal variability in CH4 flux compared to CO2 flux for bog species. This line (259-260) was removed because it does not seem necessary here. The point is that CH4 measurements taken during the daytime only were used to represent the overall CH4 fluxes. By contrast, the CO2 flux is strongly controlled by light level and thus had to be modelled

on a shorter (hourly) time step because light level is obviously changing throughout the day.

Manuscript changes: The first sentence of section 2.6 (Line 259-260) was removed because it is not really necessary for the method description.

R2 COMMENT: L337 Unusual long-term dates; 1980-2010 more usual.

Response: This is the data range available at the nearby weather station.

Manuscript changes: No changes.

R2 COMMENT: Fig. 2b Degree symbol missing on y axis.

Response: Good catch.

Manuscript changes: The suggested changes are made to the figure.

R2 COMMENT: Fig 5b The use of a 1:1 line would provide better information as to the performance of the model

Response: That is a good idea.

Manuscript changes: The suggested changes are made to the figure.

R2 COMMENT: Figs. 4,6, 9 and 10 Check spelling of plant names.

Response: OK

Manuscript changes: Plant name spelling mistakes are corrected in the figures.

R2 COMMENT: Section 3.4 Only use "significant" when related to statistical comparisons.

Response: Yes, that is already the case. Every instance of the word "significant" in the manuscript refers to statistical comparisons.

Manuscript changes: None in response to this comment, but more statistical information is included in the text in line with Reviewer 1's comments. This makes the use of the term significant is clearer.

R2 COMMENT: L407 GWP

Response: Yes, Good catch.

Manuscript changes: One instance of "GPW" changed to "GWP."

R2 COMMENT: L407 tonnes (and thereafter)

Response: Yes, Good catch.

Manuscript changes: Two instances of "tons" changed to "tonnes" in the manuscript.

R2 COMMENT: Fig. 10 Given that MAWT is used as a predictor variable in the models, these observations are not independent (especially as the collars were lumped together for modelling) and I am far from convinced as to their value in this manuscript.

Response: Hourly water table (as opposed to MAWT) data was used as a parameter in modelling NEE, but not CH4 flux. The NEE was modelled using collar specific empirical models fit to field data, and water table had a minor impact on the over modelled results. Thus, the changes in hourly water table help explain the variability in NEE but do not strongly control the modelled annual NEE. I argue that these are independent variables and the trends in these plots are useful results for comparing with other studies. These plots are valuable because they show the interaction between ecotype, genus percent coverage, MAWT and NEE, CH4 flux, GWP.

Manuscript changes: The plots in figure 10 were changed to include different symbols for the five ecotypes (as per comment by Reviewer 1), which improves the value of this figure in the manuscript.

R2 COMMENT: L436 Consider Nugent et al (2018) Global Change Biology, https://doi.org/10.1111/gcb.14449

Response: This study was not published at the time of initial submission, but is very relevant.

Manuscript changes: This suggested publication has been included and cited in section 4.1.

R2 COMMENT: L477 1960s

Response: OK

Manuscript changes: "1960's" changes to "1960s" as per previous comment

R2 COMMENT: L484/485 join the sentences

Response: Yes, this should be one sentence.

Manuscript changes: Lines 484/485 have been joined as one sentence.

R2 COMMENT: L490 five decades

Response: OK

Manuscript changes: "5 decades" changed to "five decades"

R2 COMMENT: Section 4.2 I am not sure of the value of this section. The manuscript is already quite long and this seems superfluous (especially given the extensive data set in the Supplementary). If it really must be kept, then it should be moved to the Results section and then discussed here.

Response: After re-reading the manuscript, I agree that much of this discussion is unnecessary and a little off topic for the main findings of the paper. However, the literature comparisons in fig 11 and 12 are valuable as are the extensive data sets behind them included in the supplementary tables. This information puts the study in a broader context.

Manuscript changes: Figure 11 and Figure 12 have been moved to the results section, and the discussion in section 4.2 has been substantially shortened.

R2 COMMENT: L515 Natural or semi-natural = intact?

Response: See response to comment on Line 31, above.

Manuscript changes: As per above comment.

R2 COMMENT: L539 What does "Restoration of high quality peatland ecology" mean?

Response: I can see how this is a bit confusing.

Manuscript changes: This "high quality" has been changed to "Sphagnum dominated." Though, much of this section has been re-written.

R2 COMMENT: L576 Not so: In Saarnio (2007) Boreal Environment Research. 12, 101-113, 15% of growing season flux is emitted in the non-growing season. This approach was also used by IPCC Wetlands Supplement (2014)

Response: That is a good point.

Manuscript changes: The results reported in figure 11 and 12 will remain the same, but more caution has been given to the comparison between growing season and year round data, including reference to Saarnio (2007).

R2 COMMENT: L577 What is "inter flux"?

Response: A typo: winter flux

Manuscript changes: This has been changed in the manuscript.

R2 COMMENT: L608/609 "The impact of these things.." is very vague.

Response: Yes, this line is "vague" and may not be necessary to include. It does not add anything to the manuscript.

Manuscript changes: This line has been removed from the manuscript.

R2 COMMENT: L628 and also due the pulse effect.

Response: ??? This comment does not seem match the text in this line.

Manuscript changes: No changes.

R2 COMMENT: L630 Why not quote Frolking directly?

Response: Yes, that is a good point.

Manuscript changes: The quote from Evans et al. 2016 is removed, and the primary reference is cited directly to make a similar point.

R2 COMMENT: Conclusion This reads as a summary not as a conclusion; what does your study mean for land managers, policy makers etc?

Response: That is good advice.

Manuscript changes: The conclusion has been re-written to focus less on summary and more on important findings form this work.

R2 COMMENT: L668 "best models" = the ones you used?

Response: Yes.

Manuscript changes: The wording "best models" has been changed in this line to "the models used."

R2 COMMENT: Supplementary: Check spellings throughout

Response: OK

Manuscript changes: Section S1 has been re-worked and edited in response to both Reviewers' comments as there seems to be some confusion from both Reviewers. The supplemental section previously read as a prosaic exposition of the various models tested in the development of this work. It has been changed to simply give detailed information on the models used.

R2 COMMENT: Tables S1 and S2 Please provide the standard error (SE) associated

with each parameter estimate. Given the large number of parameters in the models, I would suspect that the SE will be very high and would invalidate your approach.

Response: Including the SE of the model parameters was also suggested by Reviewer 1. That will be included in these tables.

Yes, there are a large number of fitting parameters. The same GPP and ER models were used for all 29 collars, but the model fit parameters were determined empirically for each of the collars individually. For the GPP modeling, there was sufficient field data to justify a model with 5 empirical fitting parameters. For the majority of the collars, all of the GPP model fit parameters are significant to 95% confidence. This model was designed in such a way that the effect of modelled parameters reduces to zero as the explanatory value of the additional variables decreases such that insignificant model parameters have a minor impact on the modelled results. For the ER modeling, the sample size is smaller and the point that Reviewer 2 is making is quite valid. Previously, there had been some debate among the authors as to which of two models to use for ER, so much so, that information was included on both of these models in the supplemental section in the original manuscript draft. Thus, the simpler ER model (with 3 fitting parameters compared to 5 fitting parameters) has been used to calculate ER in the updated manuscript. This model was taken directly from Wilson et al., 2016b and developed from the same type of data, collected in Ireland.

Manuscript changes: Additional statistical information has been included in the table S1 and S2 of the model fit parameters. The ER has been calculated by a different and simpler model in the revised manuscript, which had been previously described in the supplemental section but not used in the manuscript and was taken directly from Wilson et al., 2016b. The change in ER model had a minor effect on the overall conclusions of the paper. Also, the text of this supplemental section has been substantially revised to clarify confusion expressed by both Reviewers.

Please also note the supplement to this comment:

https://www.biogeosciences-discuss.net/bg-2018-350/bg-2018-350-AC2-supplement.pdf

---

## Author Response (AR1)

**Point by Point Reviewer Response 1**

2

**3 **Reviewer 1**

**4 General comments:**

- 5 What's the paper about: In this ms Swenson et al attempt to estimate C-balance of five
- 6 managed raised bogs in Ireland. The five sites differ in their hydro-physical and ecological
- 7 characteristics, as defined by the National Parks and Wildlife Servies of Ireland.
- 8 The strength of their study is in their attempt to estimate total C-balance of each site
- 9 based on measurements of various aspects of ecosystem carbon balance during the
- 10 same season, including: ecosystem CO2-gas flux, ecosystem CH4-gas flux, aquatic 11 fluxes of dissolved organic and inorganic carbon and CO2 gas efflux from open water
- in drainage ditches. Furthermore, they provide a literature compilation and review of
- 12 13 studies that have measured ecosystem CO2 and methane flux in boreal and temperate,
- managed and intact, peatlands. They use the global synthesis to explore global 14
- 15 patterns between the fluxes and mean annual water table. They discuss their sites in light of the
- global patterns and argue that the most practical advice to policy makers 16
- 17 on C-sequestration potential of restored peatlands should involve insights into the
- impact of water table and vegetation composition on C-fluxes two aspects that are 18
- 19 actually manageable at sites. Some caution should be taken when interpreting their
- 20 annual sums of different component fluxes, as most are modelled based on point
- 21 measurements throughout the year and in some cases models were used to extrapolate
- 22 beyond the data range used to develop the empirical model! Lots of assumptions used
- 23 in some cases.

Response: The authors agree that more cautionary notes should be given to exptratolating 24 beyond data collection periods. For the specific assumptions that have been pointed out in the 25 detailed comments, responses are below. The request for more statistical information on the 26 27 models and the apparent need for a clearer description of the modeling process has been 28 included in the main body of the manuscript as well as in the supplemental section.

29 Manuscript changes: More caution has been given to extrapolating the modelled CH4 and CO2 30 flux data beyond the data collection periods. The request for more statistical information on the

- 31 models and the apparent need for a clearer description of the modeling process has been
- included in the main body of the manuscript as well as in the supplemental sections. 32

33

- Detailed Comments: 1. This is a bit lengthy ms, but given it has a synthesis review, 34
- 35 maybe ok for length. Journal can decide if to have it split into ms a and b or one ms
- when published. ALternatively there may be enough material here to write two separate 36

- more focused ms: one using actual observed data, focusing on observed trends and 37
- one on estimated/modelled trends in light of the global synthesis work. 38
- 39

40 Response: Agreed that the manuscript is too long, and some parts can be shortened, 41 particularly the discussion section on comparisons to literature. It probably works better as one manuscript (if the Editor agrees). 42 43 44 Manuscript changes: As per comments from Reviewer 2, much of the global synthesis discussion has been cut from the text as the Figs. 11 and 12 can speak largely for themselves. 45 Figures 11 and 12 have been moved to the results section. This keeps the discussion section 46 47 more focused and shortens the manuscript as a whole. 48 2. Title may be worth modifying to include "implications for restoration efforts" in it -49 or something along the way. Also use of "restored and cutover" is a bit confusing. It 50 sounds like both sites are undergoing restoration, just for different amounts of time. 51 52 Perhaps reword to be more clear: "raised bogs at different stages of restoration". 53 54 Response: The cutover area is not really undergoing restoration as such. Thus, it doesn't make 55 sense to include "different stages of restoration" in the title. 56 57 Manuscript changes: Title has been changed to: Carbon balance of a restored and cutover 58 raised bog: Implications for restoration and comparison to global trends 59 3. Keywords: may be add DIC, DOC, global synthesis, carbon balance 60 61 Response: Good suggestions. 62 63 64 Manuscript changes: DIC, DOC and carbon balance has been added to the key words. 65 4. Abstract - you mention measuring along 5 different ecotypes, but only two are listed. 66 Describe all 5 then. Also reword to include all 5 ecotype descriptions, avoid use of 67 68 specific category name (ex. Sub-central) and just use general description in Abstract. 69 Don't think that category name adds info for non-Irish readers or those unfamiliar with 70 NPWS classifications. 71 72 Resonse: OK, this comment and comments 6 and 7 suggest that Reviewer 1 is getting confused 73 on the site description. We agree that this needs to be clarified both in the abstract and the main 74 body of the paper. However, the Reviewer is asking to include details of all five the ecotypes in 75 the abstract, but then not to use the names. To fully describe the 5 different ecotypes, without using the names would be far to wordy for an abstract. 76 77 78 Manuscript changes: A much clearer and still concise description of the study site is included in the abstract. Names of specific ecotypes have been removed from the abstract. All of the 79 80 ecotypes have not been described individually, but this general description has been included: "were measured for five distinct ecotypes ranging from those with high quality peat forming 81 82 vegetation down to communities indicative of degraded, drained conditons." Also, a brief but clear description of the site has been added in the abstract. 83 84

2

85 5. line 105: unidentified acronym occurrence? NECB.

87 Response: OK 88 89 Manuscript changes: "net ecosystem carbon balance" (NECB) has been added to define the 90 acronym. 91 92 6. lines 120-125. Objectives state two main research sites: 1960 cutover and 2009 93 cutover, but in Abstract and rest you discuss 5 sampling sites of different "eco-types". 94 95 Response: Yes, the description of sampling locations and study site needs to be clarified. Five 96 ecotypes were located in areas with two different site histories. 97 Manuscript changes: A clearer description of the ecotypes and site history has been included in 98 99 both the abstract and the main body of the text. The wording in the objectives was changed to 100 "for five distinct peatland ecotypes, which are located in two adjacent areas with contrasting site 101 histories." 102 7. lines 142-145: Again there seems to be a mismatch in site description - here you 103 104 say two sites were uncut and three cut, but in objectives on lines 120-122 it seems to 105 suggest both sites were cut, just one left to recover since 1960 and the other site only since 2009. So please clarify your site descriptions. May be clarify that you have two 106 main research sites and 5 sub-sites within them and how they all differ with respect to 107 108 their management history. Also add reference to table 1 early in this paragraph. 109 110 Response: Agreed, this can be clarified as per the above comments. The term recently restored 111 is used in the last paragraph of the Introduction, this might imply that the bog had been harvested (and has now being restored) whereas it just been drained in preparation for being 112 113 cut (but not actually harvested). The site history needs to be clarified earlier on. 114 115 Manuscript changes: See previous comments. 116 117 8. lines 164 - you had trees on your sites? please include % cover in Table one or 118 description. How large are they on average and species type, stem density 119 120 Response: Yes, there were trees on the site as a whole. For example, the weir catchment area 121 contained "lightly forested drains along a bog road (<10%)" Line 276. However, all of the ecotypes were treeless. Line 164 reads "open areas, excluding any large trees". In this case, 122 123 "excluding any large trees" is a qualifier describing the open areas i.e. there were no trees at our 124 monitoring sites. 125 126 Manuscript changes: "excluding any large trees" is changed to "excluding any trees" 127 128 9. Line 167: collars were installed to represent ecolocigal variability - such as what? 129 130 Response: Again, it seems that the site description needs to be clearer in the manuscript. Within

Response: Again, it seems that the site description needs to be clearer in the manuscript. With
 the ecotype definitions, there can be variability for the specific species coverage.

3

133 Manuscript changes: See above changes to line 7. 134 135 10. lines 211: which light levels, not clear? Please list PPDF under which measured. Where measurements corrected for PAR reduction due to chamber transmittance reduction? 136 137 138 Response: The light sensor was located inside of the chamber during measurements, so there 139 was no need to correct for chamber transmittance as the light level was directly measured. 140 141 Manuscript changes: The phrases "generally under full ambient light, 1-2 light other partial 142 shading light levels, and a completely shaded measurement" and "located inside the chamber" have 143 been added to the manuscript. 144 145 11. NEE modelling, section 2.5: modelled NEE to account for diurnal variability - based 146 on what? did you measure diurnal variability in data? Line 243: PPFD used from which chamber? or outside? lines 245 and Table S3: when reporting stats it is insufficient to 147 148 just present R2 values. Show also n-values and estimated coefficients. Also describe how many data points were used to fit the model and how many to test/validate the 149 model fit? Did you do that? 150 151 Response: The gross primary production (that is CO2 uptake by photosynthesis) is the largest 152 153 component of net ecosystem exchange. In practically every system, the gross primary 154 production is strongly controlled by the light intensity, which obviously has diurnal variations. Thus, the hourly light intensity (measured in the field at the weather station) was input into 155 models of NEE, which results in expected diurnal fluctuations of the modelled results. 156 157 Measurement of light intensity is described in Lines 186-191 of the manuscript. 158 All of the field data was used to calibrate the models. Validation was not done explicitly. 159 160 161 Manuscript changes: Further, statistical information is included in Table S3. Supplemental 162 section S1 has been revised to show a clearer description of the modeling process, including (among other things) a description of the number of data points used to fit the models. 163 164 165 12. Methane modelling: be very cautious in your "modelling" attempts and interpretations - you fit an empirical model to a small subset of observations and then attempt 166 to use that to predict fluxes outside of the data range used to develop your empirical 167 model. I don't think this is statistically sound, although I appreciate your attempt and 168 169 sympathize with instrument malfunction. So highlight this limitation in your text and discussion. Line 265 - "collar average" probably should read "overall average from 170 171 measurements at all collars". 172 173 Response: This is a good point. More cautionary notes should be included in describing the 174 limitations of this data. All of the methane field measurements were used rather than "a small 175 subset." "Collar average" is the correct terms here. 176 Manuscript changes: More cautionary notes have been included in describing this data 177

178 throughout the manuscript.

13. Section 2.7, lines 278: - is that a valid assumption? Response: This is a big assumption, that the aquatic carbon losses were the same for each of the ecotypes, but it is an assumption that we had to make due to resource limitation. (It would be an interesting topic for a future research project.) Many studies that have measured other aspects of the carbon balance have made bigger assumptions about the magnitude of the DOC losses. For example, Wilson et al., 2016b estimate DOC losses from a peatland "based on guidance provided by IPCC [report]" rather than field measurements. In this case, the DOC/DIC flux was at least measured directly on-site. Manuscript changes: None. 14. Lines 307-312: you discuss CO2 evasion from drainage ditches, what about openpond water on your bogs? Any present? what proportion? how much would the open ponds add to CO2 efflux from open water? Response: These are valid questions, but in this case, the only open water areas at the study site were associated with drainage ditches. Manuscript changes: None. 15. Line 315: likely mean "DIC" instead of "DOC"? Response: No, this is correct. Here the DOC flux is calculated differently for the carbon balance of the system, which includes all carbon losses from the catchment area and for the global warming potential, which is the greenhouse gas effect. This is because some of the DOC lost from the system may be stored in longer term sediment and not contribute to the GWP. Manuscript changes: None. 16. Lines 321-322: Sentence makes no sense? what are you saying? "... from the sum of model error and error of input field variables"? Response: This sentence is describing the model statistics. Manuscript changes: This sentence has been re-worded for clarity. Also, partially because of this confusion expressed by the Reviewer here and partially because the modelling of ER was redone as per comments by Reviewer 2, the statistics have been redone using simpler and more straight forward methods. 17. Lines 338: report st.dev on mean annual total from Ballyroan. Response: OK Manuscript changes: These details have been added to the manuscript.  226 227 18. lines 340: 1978-2007 average taken from where? also report stdev on mean 228 annual value. 229 230 Response: OK. 231 232 Manuscript changes: The mean annual temperature is changed to the 30 year average 1981-233 2010 based on Walsh, 2012 as per comments by reviewer 1. Walsh (2012) does not report the 234 stdev so this is not included. 235 19. Figure 4: hard to tell two blue colours apart, likewise for yellow and orange, suggest 236 237 to change. 238 239 240 Response: OK 241 Manuscript changes: The colors have been changed in Figure 4 to improve readability. 242 243 244 20. Figure 4 -7 - either show all modelled and observed fluxes like you do for CH4-245 fluxes in Figure 5a ( IWOULD HIGHLY RECOMMEND, OTHERWISE YOU SHOWTOO MUCH MODELLED RESULTS but little observed values used to derive the modelled 246 values!) or as monthly bar plots or cumulative plots like you do for terrestrial CO2-247 248 fluxes in Fig 4. This is also probably where you can reduce your figure numbers. Fig 249 5b probably belongs to Supplement where you describe your model fitting. 250 251 Response: Moving Fig. 5b to the supplemental section is a good suggestion. Otherwise, this 252 comment is a little perplexing: on the one hand, Reviewer 1 mentions reducing the number of figures; on the other hand, Reviewer 1 asks for more figures showing modelled and measured 253 254 results. As the GPP and ER were modelled separately for each of the 29 collars, it would be confusing/misleading to show the entire field dataset and modelled results in a single plot. The 255 modelled and measured data is shown together here for one particular collar as an example. 256 257 For all 29 collars, 77 such plots (3 for each collar) would be needed, which may be excessive 258 even for the supplemental section (although if the Editor thinks this is valuable, these plots can be included). The  $r^2$  values, although not statistically sufficient by themselves, demonstrate the 259 260 modelled data correlation with field data in a much smaller format. 261 262 *"like you do for CH4- fluxes"* Showing all of the modelled and measured CH4 flux data was

263 possible because the field data from all collars was lumped together before modelling.

264

Hours after 1st Jan. 2016
Figure caption. This figure shows the modelled and measured data for collar EC13 for ER (top), GPP
(middle), and for clarity (because GPP drops to 0 every night with light level), GPP normalized by light
level to show the seasonal fluctuations independent of light level (bottom).

Manuscript changes: Fig 5b has been moved to supplemental section. Otherwise, no changes
 has been made unless the Editor requests plots like the example above to be in the
 supplemental section.

**274 21. Section 2.4 name - CO2-gas flux measurement, not GHG.**

- 275 Response: Yes, that is better.
- 276

273

- 277 Manuscript changes: The section heading name changed to " $CO_2$  and  $C_{H4}$  gas flux
- 278 measurements"

22. Aquatic carbon losses: Fig 7: how do you know point around 2 mg/L is an outlier for DIC to assume a constant flux throughout the year? Seems too few points to make such assumption. Response: This is a fair point, but it is just an assumption and has a very minor effect on the overall results, as this is the smallest component of the C balance. Based on Dixon's Q test, this point can be excluded as an outlier to 95% confidence. Manuscript changes: None. 23. Fig 8: what's WHB and EHB? Response: These points need to be labeled on Fig.1 Manuscript changes: These points have been labeled on Fig.1 24. Section 3.4: not sufficient to report only p-values for statistical results, report also associated n used, mean and st.dev. Lines 404-406: was spatial variability accounted for in your analysis? Response: OK, on the stats. It is not clear what is meant by spatial variability; this is exactly what is being described here i.e. difference between collars within the same ecotype. Manuscript changes: Additional statistical information added throughout this section, where applicable. It is not always appropriate to include n, mean, and stdev for every statistical analysis. 25. Figure 9a: - units unclear gC-CO2? CO2-equivalent? so Fig a and b are the same, with a in g/m2/yr and b in tones/ha/yr? why include both? Response: The carbon balance and the global warming potential (GWP) as shown in Fig. 9a and 9b, respectively, are fundamentally different quantities with correspondingly different units. Although the change in units may be confusing, the units chosen are standard units for reporting these types of measurements. Manuscript changes: None. 26. line 416: "carbon flux" probably mean "CO2 flux", same comment on lines 424. lines 416-417: unclear - so you looked at environmental controls for each collar separately? why not together per eco-type? Response: The phrase "carbon flux" is not used. The phrase "carbon balance" is the correct wording here and refers to multiple aspects of the carbon balance, including CO2 flux. The data

are presented together by ecotype in Fig. 4, Fig. 6, and Fig. 9. As data (flux & divers) is 326 327 available in detail at each collar, there is no reason not to include these comparisons. 328 329 Manuscript changes: This section has been changed to describe the patterns in collar CO2 flux 330 instead of patterns in carbon balance data, to reduce confusion. 331 332 27. Section 3.5 - again stats should show n, mean, stdev together with p-value, else 333 meaningless. 334 335 336 Response: This is a good suggestion. 337 338 Manuscript changes: The additional statistical information is included throughout this section. 339 340 28. lines 428-433: don't get this paragraph. 341 342 343 Response: The percent genus cover within the collars is compared to the GWP. C-balance, and 344 CH4 flux. 345 Manuscript changes: This paragraph is guite short and seems fairly clear. Updated statistical 346 information has been included, which may help. Also, the updated version of Fig. 10, which is 347 348 referenced in the paragraph, may help clarify. 349 350 29. Fig. 10: What if you colour points by ecotype? would be nice to see how they fit on the scatter, if group or not, in plots a and d - units are in gC, so does this C 351 352 include CH4-carbon? Legend says data was averaged over 2 years - why? was there no interannual variability? 353 354 Response: Yes, good idea to colour by ecotype (and would also address comment 26 355 somewhat). Data was averaged over the two years for clarity in the plots because these plots 356 357 are focused on spatial rather than temporal variability; the longest data set available was 358 included. Yes, in the first draft of the manuscript the carbon balance includes all aspects of the 359 carbon balance. However, a and d of this plot have been changed in the revised manuscript to 360 include only CO2 flux (NEE) only rather than carbon balance. 361 362 Manuscript changes: The collars were coded by ecotype as per the Reviewer's suggestion in 363 this plot. Plots a and d of this figure 10 have been changed in the revised manuscript to include only CO2 flux (NEE) only rather than carbon balance. 364 365 366 30. Discussion: Lines 436-437: well, I would be cautious with such as statement, as nowhere in your paper do you present simultaneous measurements of all components 367 368 of c-balance you meausured. If you do have observation days where you have all of the fluxes measured on the same day - those would be your key days to focus on and 369 370 show when trying to figure-out contribution of each component flux to overall C-flux.

- 371 Such comparison, even if only on few days would be more valuable than that gapfilled
- 372 modelled comparisons to field-based research.

| 373 |                                                                                                    |
|-----|----------------------------------------------------------------------------------------------------|
| 374 | Response: This is a fair point with being cautious about this statement, as the word               |
| 375 | "simultaneously" is a little misleading. This line could be re-stated to be more precise. However, |
| 376 | adding in description about specific days where all the fluxes were measured together with all of  |
| 377 | the other data (flows etc.) would probably not help the structure of the paper (and would make it  |
| 378 | a lot longer).                                                                                     |
| 379 |                                                                                                    |
| 380 | Manuscript changes: The word "simultaneously" has been removed from this paragraph.                |
| 381 | Instead, this line has been changed to "concurrently quantified annual fluxes of all major aspects |
| 382 | of the C balance"                                                                                  |
| 383 |                                                                                                    |
| 384 | 31. lines 449-452: so did Nilsson et al also take a single measurement in a year and               |
| 385 | assume DIC to be constant? If not, how many they took and how the differences in                   |
| 386 | sampling between their study and yours impacts the results. Think of n-sampled.                    |
| 387 |                                                                                                    |
| 388 | Response: Agreed that more discussion is needed on the causes of differences between               |
| 389 | studies although it is unlikely that the differences observed are due primarily to the number of   |
| 390 | samples.                                                                                           |
| 391 |                                                                                                    |
| 392 | Manuscript changes: Further discussion is included on the differences between these study          |
| 393 | sites and results. See also to response to comment 32 below.                                       |
| 394 |                                                                                                    |
| 395 | 32. Lines 458: lower than Dinsmore and Nilson by how much? make it easy on the                     |
| 396 | reader, so less likely to flip back and forth. show both values or state % difference.             |
| 397 | Response: Yes, this paragraph is a bit burdensome with a lot of data comparisons. The              |
| 308 | comparison between studies has been put into tablular format to be assign to read format. Then     |
| 399 | this paragraph can be focused more on the differences in methods, etc. causing the variation in    |
| 400 | results (Partially, addressing Comment 31)                                                         |
| 400 |                                                                                                    |
| 402 | Manuscript changes: A table of the various components of the carbon balance from these other       |
| 403 | two studies is included in Section 4.1 (now section 4.2 in the revised manuscript). This           |
| 404 | naragraph is also substantially shorter than previously                                            |
| 405 |                                                                                                    |
| 406 | 33. SECTION 4.2 - I THINK THIS BELONGS TO RESULTS - this section should be                         |
| 407 | split with synthesis plots shown in results and their discussion left in Discussion. ALso          |
| 408 | how data was collected and filtered should be in methods.                                          |
| 409 |                                                                                                    |
| 410 | Response: That is a good suggestion and agrees with Reviewer 2.                                    |
| 411 |                                                                                                    |
| 412 | Manuscript changes: The figures 11 and 12 have been moved to the results section and               |
| 413 | relevant pieces have been moved to the methods section or kept in the discussion. On the           |
| 414 | whole, the discussion section has been shortened and re-written for clarity, as per response to    |
| 415 | Reviewer 2.                                                                                        |
| 416 |                                                                                                    |
| 417 | 34. lines 532: how dry is "too dry"? also Briones et al reference is missing.                      |
| 418 |                                                                                                    |

Response: The reader can go to the relevant reference if interested in this question: Briones, M. J., McNamara, N. P., Poskitt, J., Crow, S., and Ostle. N. J. Interactive biotic and abiotic regulators of soil carbon cycling: evidence from controlled climate experiments on peatland and boreal soils. Global Change Biology, doi: 10.1111/gcb.12585, 2014. Manuscript changes: This line has actually been removed from the manuscript. This line was deemed to be an unnecessary tangent. As this is the only line that references Briones et al., 2014, the reference was not included in the reference list. 35. Fig. 11 - add lines to legend Response: OK Manuscript changes: Lines were added to the legend in Fig. 11. 36. lines 568-574 discussion - so how many points were from EC-derived NEE and how many from chamber derived-NEE? how do they fit on your Fig 11/12. Is there a difference between the two? (ex. one method consistently lower, but trends same, or one falls on one end of trend and the other at the other end? Response: The primary difference between the use of these two methods is for the land use type as described in the text. The curious reader can pull out this information in the Supplemental tables, but this plot with numerous symbols already, may become overcrowed with this information. Manuscript changes: None. 37. line 577: "inter" probably should read "winter" Response: Yes. Manuscript changes: "inter" changed to "winter" 38. Fig 12: you specify your data points with numbers 1-5. I assume that relates to your ecotypes - so which is which? add to fig description. Response: Yes, this needs to be added. Manuscript changes: The numbers 1-5 have been specified in the figure caption. 39. Lines 606: "managing water table" - that's repeat of point 1). Response: That is a fair point. Manuscript changes: The words "managing water table" were deleted from line 606.

40. line 608: "... the impact of these things must be considered." which things? and 466 467 their impact on what? 468 469 Response: Yes, this line is "vague" as per Reviewer 2, and can be clarified. 470 Manuscript changes: This line has been changed to "the impact of these actions on C balance, CH4 471 472 flux, and GWP must be considered" 473 474 41. Lines 621-624, follow discussion about impact of Sphagnum presence on GWP. 475 So what has the study of Junkurst and Fielder to do with Spagphum-GWP? 476 477 Response: OK, further discussion can be included on the Sphagnum effect on GWP. Junkurst 478 and Fielder is a little off topic here. This reference may be removed. 479 480 Manuscript changes: As per above responses the entire discussion section has been 481 streamlined and clarified. These particular line referring to Junkurst and Fielder have been 482 removed. 483 484 42: Lines 628-635 - unclear what you're trying to say. Are you attempting to say that peatlands lifespan is more than 100 years, so their assessment should be based on 485 longer time scales? So how long? 486 487 488 Response: This sentence needs to be clarified, but is important. Peatland preservation is beneficial (in terms of greenhouse gasses) despite methane emissions because of long term 489 490 sequestration and storage of carbon out balances methane emissions. However, for peatland 491 restoration, the greenhouse gas impact depends on the time scale that restoration works effect 492 the eco-hydrological trajectory. For example, if restoration works only impact the short term (decadal) eco-hydrological trajectory, then methane emissions may be proportionally more 493 494 important to consider for the overall greenhouse gas budget. 495 496 Manuscript changes: These lines have been clarified. 497 498 43. Lines 649: what's Marginal and Facebank ecotype? references? 499 500 Response: Yes, this is unclear. 501 502 Manuscript changes: These lines were ultimately removed from the manuscript along with the undefined and Irish specific terms "facebank" and "marginal". These lines were removed simply 503 to shorten and streamline the discussion section as they were not central to the overall points of 504 505 the discussions section. 506 S1. GPPmax is assumed constant throughout what? which metrics from Wilson et al 507 508 2007 where used? how many data points were used for model development at each point? why biological zero is assumed at 0C? what reference you have for this? Label 509

- 510 your equations consequitively and consistently. SHow your model comparisons, how
- 511 good each fit is. Why model fit to all collars so did you check they all behave the

**512 same? where are the results? 513 Response: Section S1 describes the modelling process. The number of questions surrounding 514 this section suggests that this entire section could be clarified. Yes, some assumptions were 515 516 made. However, not too many major assumptions were made because several different empirical models were fit to the field data and then compared. The actual biological zero has a 517 minor impact on the model results. An exponential increase with temperature is quite common 518 519 to be applied in biological systems at the temperature ranges of the Irish climate. Why model fit to all collars - so did you check they all behave the same? Quite the opposite, I did not assume 520 521 that all collars behave the same way because of ecological and hydrological differences 522 between collars. Thus, empirical models were fit to the field CO2 flux data from each of the 29 collars individually. 523 524 525 Manuscript changes: This section has been re-written and clarified. where are the results? See 526 response to Comment 20. More field data could be included at the Editor's request as described 527 above. 528 529 S2. Equation S10 - which one is that? 530 531 Response: This section has been re-written and clarified as per above comment. 532 533 Manuscript changes: As above. 534 535 S3. Tables S1 and S2 - kind of useless statistically. Please add estimated parameter 536 coefficient's stats - p and t-values, also model stats such as R2, n-observations. 537 538 Response: That is fine. 539 540 Manuscript changes: More statistical information has been included in these tables. 541 542 S3. Table S3: p-values? n? F-stats? data ranges used to fit model? all needed to 543 make sense of reported R2. 544 545 Response: That is a good suggestion. 546 547 Manuscript changes: More statistical information has been included in Table S3. 548 549 S4. methane modelling - what worries me most, you fit an empirical model to limited 550 data range and then use that to extrapolate beyond that data range. Don't think that's good statistical practice. Be cautious of such things. If there's a physiologically based 551 552 model to work with - try that instead. Was effect of Temp similar to that of ER - show? 553 "Temporal variation in fluxes was extrapolated in this model" - what are you trying to 554 say? 555 556 Response: This is a good point that more caution needs to be added around the model results. 557 Though, perhaps the purpose of modelling methane fluxes for this study needs to be clarified a**

558 bit more in the paper. The purpose of this model was not to predict the methane flux at a

particular point in time. Rather, the purpose of this model was to estimate annual methane 559 560 fluxes. Essentially the average methane flux at each collar was scaled by a factor to account for 561 the fact that the field data was not collected over the entire year. The model was used to predict 562 that factor. Further, the major hole in this data is that methane measurements were only 563 collected during one of 2 years. This limitation in the data needs to be highlighted more explicitly in the manuscript. The assumption that the methane flux was the same is partially 564 verified by the fact that the model gave very similar results for both years. This is the other 565 566 purpose of this model. 567

568 "limited data range" This phrase is used several times by the Reviewer; I am not entirely sure of the meaning. If the Reviewer is referring to a limited data **collection period**, than this is a fair comment. However, if the Reviewer is referring to a limited **data range**, than, it seems that the reviewer is mistaken. The modelled results were not extended beyond the data range.

I am not aware of an appropriate physiologically based model to use in this case because
 methane fluxes from peatlands can be highly variable both within and between sites.

576 "Temporal variation in fluxes was extrapolated in this model" Yes, this needs to be clarified.

578 Manuscript changes: More caution needs to be applied when presenting methane modelling
579 results, particularly for 2016 when no field data was collected. Also, the purpose methane
580 modeling has been clarified in the manuscript.

582 Also was there any model validation done for any of your modelling activities?

584 Response: No, in this case, all of the field data was used in calibrating the models.

586 Manuscript changes: None.

587 588

572

577

581

583

585

589

590 **Reviewer 2**:

591 General Comments:

592 General comments: The manuscript reports the results from a two year study at two 593 peatlands in Ireland: an abandoned (but not rewetted) and a rewetted peatland. In both 594 sites, a full carbon balance (CO2, CH4, DOC and DIC) was measured and calculated. 595 The authors indicate that the abandoned site was a strong annual carbon source and 596 that the rewetted site was a small carbon sink. The authors also compare their results 597 with literature values (a very nice literature review is included in the Supplementary 598 material). The manuscript is well written (although it would benefit from a spell-check 599 and does feel a little long), tightly focused (except Discussion) and the results are clear. 600 However, the Discussion section is disjointed and requires some surgery, and I also 601 have some concerns in regard to the models used but this may just need clarification 602 by the authors rather than any major reconstruction.

**603 Response:**

The Reviewer comments that the paper is a "little long" and the Discussion section is "disjointed
and requires some surgery." I agree with this comment in general. After a fresh read-through, I
think that the discussion section gets a little off topic from the study results by including lengthy
discussions of data from literature. The comparison to literature data is valuable, including Figures
and 12 and the extensive tables of literature data in the Supplemental Section 3. However, the
discussion text surrounding these figures can be greatly shortened, while the figures would remain

understandable and useful. After some major revisions, the discussion section has been shortened
 from 3013 words to 1975 words in length and we feel that is much clearer and to the point. The

- overall important points of the discussion section have remained largely the same; the text has
- 613 been clarified, shortened, and restructured.

614 Further, the Reviewer comments, "I also have some concerns in regard to the models used but this

may just need clarification by the authors rather than any major reconstruction." The only
 "concern" raised explicitly is the number of fitting parameters (an in comment below on Tables S1

and S2) and for "clarification," the Reviewer requests that the SE of model fitting parameters be

618 included. Additional statistical information on the modelling has been included in the Tables S1 and

619 S2. The Reviewer raises a valid point; the ecosystem respiration model, in particular, arguably had

too many empirical fitting parameters. This model was replaced by a simpler model with fewer

621 fitting parameters, as described in more detail in the response to the comment on Table S1 and S2.

622 Manuscript changes:

623 The discussion section has been shortened and streamlined by cutting out much of the text

describing the comparison to literature. The Figures 11 and 12 has been left in the body of the text,

but moved to the results (as in comment on Section 4.2, below).

626

The model used for ecosystem respiration has been redone using a simpler model, which was takendirectly from Wilson et al., 2016a.

629

630 Specific comments:

L2 "Add C after carbon and use thereafter in abstract" "Add methane before (CH4)" "Not
 necessary to add "losses""

633 Response: These are good suggestions.

634 Manuscript changes: The recommended changes have been made in the manuscript abstract.

635 L4 Harvested suggests a renewable fuel source. Peat removal for fuel is anything

636 but sustainable. Please replace harvesting here (and throughout the manuscript) with

637 either "extraction" or "mining".

- 638 Response: OK, yes, this is a good suggestion.
- Manuscript changes: The term "harvested" has been changed to "mined" in the manuscriptwhen in reference to peatlands impacted by peat extraction.
- 641 L5 Please define what you mean by "historically abandoned"
- 642 Response: OK, this is potentially confusing.
- 643 Manuscript changes: "historically" has been changed to "abandoned x years ago"
- 645 L6/7 What do you mean by "high quality"?
- Response: That is a good question; high quality is referring to the site most similar to the ecology and hydrology of undisturbed open raised bog habitat in Ireland.
- 649 Manuscript changes: As per Reviewer 1's comments, the site description has been clarified in the650 abstract.
- 651 L7 Calluna vulgaris

648

- Response: The *Calluna vulgaris* refers to a species of plant. In the manuscript the CallunaCutover refers to specific ecotype.
- Manuscript changes: The named of specific ecotypes have been removed from the abstract toreduce confusion, as per Comment 1 from Reviewer 1.
- 656 L14 Why upper cases for Temperate and Boreal?
- 657 Response: Indeed.
- Manuscript changes: Temperate and boreal has been changed to lower case throughout themanuscript where appropriate.
- 660 L15 ": : : in this study and was: : :"
- 661 Response: Ahh, there is an extra word in this line.
- 662 Manuscript changes: The word "was" has been removed from this line.
- 663 L18 Add C after carbon and use thereafter in the manuscript
- 664 Response: Agreed.
- 665 Manuscript changes: The suggested change has been applied to the manuscript.
- 666 L26 95% is very high 80 to 85% is generally cited
- 667 95% refers to the percent of raised bogs that have been degraded in Ireland rather than the total
- amount of peatlands, this has been clarified in the manuscript.

- 669 L31 Throughout the manuscript, you use intact, natural, near-natural and pristine interchangeably
- 670 please select one and keep to it.
- 671 Response: This is a good suggestion.
- Manuscript changes: The term "intact" has been used in reference to peatlands which have not been
   mined or drained. The term "natural" has been used in reference to those peatlands which are not
- 674 actively being used for agriculture, intensive grazing, mining, forestry, etc.
- 675 L54 Consider Fritz et al. (2011) New Phytologist. 190, 398-408.
- Response: I was not aware of this publication. The findings from this publication are somewhatcontradictory with this line of the introduction.
- 678 Manuscript changes: This line has been changed to include the findings from Fitz et al., 2011.
- 679 L75 Please use primary source as reference; Myhre et al (2013) Climate Change 2013:
- 680 The Physical Science Basis Contribution of Working Group I to the Fifth Assessment
- 681 Report of the Intergovernmental Panel on Climate Change
- 682 Response: Yes, good point.
- 683 Manuscript changes: The suggested primary reference is used
- L77 I would not be inclined to use specific data here; why Helfter et al. and not McVeighet al. for instance?
- Response: Including specific data here is helpful for putting the present work in context. Helftler etal. is used as a source because they include a table of reported literature values.
- Manuscript changes: The use of this source is clarified by adding "literature compilation fromHelftler..."
- 690 L78 The CH4 values derived by Wilson et al (2016) include rewetted sites, so are not
- 691 suitable here, however there are lots of CH4 studies you could cite instead, e.g. Laine
- 692 et al. (2007) Plant and Soil. 299, 181-193; Green and Baird (2017) Mires and Peat.
- 693 **19**, Article 09.
- 694 Reponse: That is a good point.
- 695 Manuscript changes: The value of CH4 emissions for intact peatlands in this line of the
- introduction has been removed based on comments on L77 and L78. Assigning a typical CH4
   flux value may not be helpful in this line of the introduction because of the high variability reported
   in literature.
- 699 L82 Change methane to CH4; L90 Change methane to CH4
- 700 Response: OK

- Manuscript changes: The use of "methane" has been replaced by the chemical symbol throughout
   the manuscript after initially defining it.
- 703 L89 Bain et al (2011) is not in reference list
- 704 Response: OK
- 705 Manuscript changes: This citation has been added to the reference list.
- 706 L101 Consider Barry et al. (2016) Aquatic Sciences. 78, 541-560
- 707 Response: OK
- 708 Maunscript changes: This article has been added as a reference here.
- 709 L109 "recovering" is a new term to me. Do you mean rehabilitated?
- Response: Recovering means bogs which have had no definite action taken to rehabilitate them.
   They have just stopped being mined and left. So, the term "rehabilitated" is not correct.
- 712
  713 Manuscript changes: In this line "abandoned" has been used in place of "recovering". In the
  714 manuscript, this term has now been defined here as the "spontaneous revegetation of mined
- peatlands" (From Poulin et al., 2005), which have had no definite action taken to rehabilitate them.

L129 What do you mean by "natural peatland area"? The site is obviously not natural
 and the surrounding landscape is mainly grassland and some forestry. Delete.

- 718 Response: OK, The site is a natural area with a variety of plant communities some of which is 719 peatland in various degrees of degradation.
- Manuscript changes: The wording has been changed from "natural peatland area" to "peatlandand natural area".
- 722 L130 See earlier comment regarding harvesting.
- 723 Response: As in above comment.
- 724 Manuscript changes: As in above comment.
- 725 L132 Met station location? 1980-2010?
- 726 Response: Ok, this could be clarified.

727 Manuscript changes: The 30 year average (1981-2010) meteorological mean annual rainfall and 728 temp from (Walsh, 2012). Have been used in the site description instead of the shorter period of 729 record from nearby weather stations.

- 730 L134 1980s; L140 1970s and 1960s 731
- 732 Response: OK

| 722        |                                                                                         |
|------------|-----------------------------------------------------------------------------------------|
| 734        | Manuscript changes: The apostrophe has been removed from all decade numbers.            |
| 735        |                                                                                         |
| 736        | Fig. 1 The quality of Fig. 1 is poor, although this might be due to the pdf. The legend |
| 737        | on the elevation map is hard to determine.                                              |
| 738        |                                                                                         |
| 739        | Response: Noted.                                                                        |
| 740        |                                                                                         |
| 741        | Manuscript changes: A higher resolution figure has been used for the final submission.  |
| 742        |                                                                                         |
| 743        | Table 1. Check font sizes and spelling of Sphagnum and Eriophorum                       |
| 744        |                                                                                         |
| 745        | Response: OK, good catch.                                                               |
| 746        |                                                                                         |
| 747        | Manuscript changes: These typo mistakes have been corrected.                            |
| 748        |                                                                                         |
| 749        | L188 Where was the sensor located?                                                      |
| 750        |                                                                                         |
| 751        | Response: Sensor was located inside of the chamber.                                     |
| 752        |                                                                                         |
| /53        | Manuscript changes: This detail is added to the manuscript.                             |
| 754
755 | 106 Stainlean staal collers                                                             |
| 755        | L 196 Stainless steel collars – as written it appears as it you only had one            |
| 750        | Response: Agreed                                                                        |
| 758        | Response. Agreed.                                                                       |
| 759        | Manuscript changes: "collars" instead of "collar" The word was pluralized               |
| 760        |                                                                                         |
| 761        | 1 197 Where was the water trough located?                                               |
| 762        |                                                                                         |
| 763        | Response: Along the top edge.                                                           |
| 764        |                                                                                         |
| 765        | Manuscript changes: Added "along the top edge" to this sentence                         |
| 766        |                                                                                         |
| 767        | L198 What does "constructed in house: : :" mean?                                        |
| 768        |                                                                                         |
| 769        | Response: This phrase means that the chambers were built by the authors and department  |
| 770        | technicians.                                                                            |
| 771        |                                                                                         |
| 772        | Manuscript changes: The phrase "in house" was changed to "in-house" as it should be.    |
| 773        |                                                                                         |
| 774        | L207 and area, volume of collar/chamber?                                                |
| 775        |                                                                                         |
| 776        | Response: The dimensions of the chamber are given above in line 198, but the area and   |
| 777        | volume can be stated more explicitly.                                                   |
| 778        | ··· ··· -··· · · · · · · · ·                                                            |
| 779        | Manuscript changes: This information has been added to the manuscript.                  |

L208 A constant temperature or a temperature similar to ambient temperature? The former could be 50C for example and fit your criteria but would be meaningless for gas flux calculations and subsequent modelling. Response: Reviewer 2 seems to have mis-understood the meaning of this sentence. The temperature was kept constant over the chamber closure equal to the initial ambient temperature. Manuscript changes: The phrase "over the chamber closure time" was added to the sentence as a clarifier. L213 State flux sign convention used in this study. Response: Yes, that needs to be included. Manuscript changes: The following sentence was added after line 213 "For this study, a positive sign convention is indicates a net loss of carbon from the peatland." L214 describe criteria used for quality checking. Response: This data was quality checked to ensure that the change in CO2 concentration over the chamber closure was monotonic, and physical parameters such as temperature and PPFD did not change substantially over the closure. Manuscript changes: The following sentence describing guality checkin criteria was added to the manuscript: "...quality checked to ensure that the change in CO2 concentration over the chamber closure was monotonic and that the PPFD did not change by more than 50  $\mu mol\ m^{-2}\,s^{-1}$ over the chamber closure." L212 How many samples? Response: "...generally under full ambient light, 1-2 light other partial shading light levels, and a completely shaded measurement." Manuscript changes: The sentence in the response was added to the manuscript. L259-260 CH4 fluxes have a strong diurnal variability in some plant types. Response: The data from Pypker et al., 2013 supports the claim of a low diurnal variability in CH4 flux compared to CO2 flux for bog species. This line (259-260) was removed because it does not seem necessary here. The point is that CH4 measurements taken during the daytime only were used to represent the overall CH4 fluxes. By contrast, the CO2 flux is strongly controlled by light level and thus had to be modelled on a shorter (hourly) time step because light level is obviously changing throughout the day.

Manuscript changes: The first sentence of section 2.6 (Line 259-260) was removed because it is not really necessary for the method description. L337 Unusual long-term dates; 1980-2010 more usual. Response: This is the data range available at the nearby weather station. Manuscript changes: The data used for annual rainfall comparisons to study period was from the Ballyroan (oatlands) Met Éirrean daily rainfall station with a period of record from 2001 to present. The average annual temperature value was changed in the manuscript to the 1980 to 2010 average temperature based on Walsh (2012). Fig. 2b Degree symbol missing on y axis. Response: Good catch. Manuscript changes: The suggested changes are made to the figure. Fig 5b The use of a 1:1 line would provide better information as to the performance of the model Response: That is a good idea. Manuscript changes: The suggested changes are made to the figure. Figs. 4,6, 9 and 10 Check spelling of plant names. Response: OK Manuscript changes: Plant name spelling mistakes are corrected in the figures. Section 3.4 Only use "significant" when related to statistical comparisons. Response: Yes, that is already the case. Every instance of the word "significant" in the manuscript refers to statistical comparisons. Manuscript changes: None in response to this comment but more statistical information is included in the text in line with Reviewer 1's comments. This makes the use of the term significant is clearer. L407 GWP Response: Yes, Good catch. Manuscript changes: One instance of "GPW" changed to "GWP." L407 tonnes (and thereafter)

Response: Yes, Good catch. Manuscript changes: Two instances of "tons" changed to "tonnes" in the manuscript. Fig. 10 Given that MAWT is used as a predictor variable in the models, these observations are not independent (especially as the collars were lumped together for modelling) and I am far from convinced as to their value in this manuscript. Response: Hourly water table (as opposed to MAWT) data was used as a parameter in modelling NEE, but not CH4 flux. The NEE was modelled using collar specific empirical models fit to field data, and water table had a minor impact on the overall modelled results. Thus, the changes in hourly water table help explain the variability in NEE but do not strongly control the modelled annual NEE. I argue that these are independent variables and the trends in these plots are useful results for comparing with other studies. Manuscript changes: The plots in figure 10 were changed to include different symbols for the five ecotypes (as per comment by Reviewer 1), which improves the value of this figure in the manuscript. L436 Consider Nugent et al (2018) Global Change Biology, https://doi.org/10.1111/gcb.14449 Response: This study was not published at the time of initial submission, but is very relevant. Manuscript changes: This suggested publication has been included and cited in section 4.1. L477 1960s Response: OK Manuscript changes: "1960's" changes to "1960s" as per previous comment L484/485 join the sentences Response: Yes, this should be one sentence. Manuscript changes: Line 484/485 have been joined as one sentence. L490 five decades Response: OK Manuscript changes: "5 decades" changed to "five decades" Section 4.2 I am not sure of the value of this section. The manuscript is already quite long and this seems superfluous (especially given the extensive data set in the Supplementary). If it really must be kept, then it should be moved to the Results section

**and then discussed here. Response: After re-reading the manuscript, I agree that much of this discussion is unnecessary and a little off topic for the main findings of the paper. However, the literature comparisons in fig 11 and 12 are valuable as are the extensive data sets behind them included in the supplementary section. This information puts the study in a broader context. Manuscript changes: Figure 11 and Figure 12 have been moved to the results section, and the discussion in section 4.2 has been substantially shortened. L515 Natural or semi-natural = intact? Response: See response to comment on Line 31, above. Manuscript changes: As per above comment. L539 What does "Restoration of high quality peatland ecology" mean? Response: I can see how this is a bit confusing. Manuscript changes: This "high quality" has been changed to "Sphagnum dominated." Though, much of this section has been re-written. L576 Not so: In Saarnio (2007) Boreal Environment Research. 12, 101-113, 15% of growing season flux is emitted in the non-growing season. This approach was also used by IPCC Wetlands Supplement (2014) Response: That is a good point. Manuscript changes: The results reported in figure 11 and 12 remain the same, but a cautionary note has been included on the comparison between growing season and year round data, including reference to Saarnio (2007). L577 What is "inter flux"? Response: A typo: winter flux Manuscript changes: This has been changed in the manuscript. L608/609 "The impact of these things.." is very vague. Response: Yes, this line is "vague" and can be clarified. Manuscript changes: This line has been changed to "the impact of these actions on C balance, CH4 flux, and GWP must be considered" L628 and also due the pulse effect. Response: This comment does not seem match the text in this line.**

966 Manuscript changes: No changes. 967 968 L630 Why not quote Frolking directly? 969 Response: Yes, that is a good point. I like how it is phrased by Evans et al. 970 971 Manuscript changes: The quote from Evans et al., is left in the text but this line has been 972 shortened to exclude 'to quote from Evans et al. (2016), "as noted by Frolking et al. (2006)," The 973 quotation marks remain to indicate a direct quote and both citations are listed as references. 974 975 Conclusion This reads as a summary not as a conclusion; what does your study mean 976 for land managers, policy makers etc? 977 978 Response: That is good advice. 979 980 Manuscript changes: The conclusion has been re-written to focus less on summary and more 981 on important findings form this work. 982 983 L668 "best models" = the ones you used? 984 Response: Yes. 985 986 Manuscript changes: The wording "best models" has been changed in this line to "the models 987 used." 988 989 990 Supplementary: Check spellings throughout 991 992 Response: OK 993 994 Manuscript changes: Section S1 has been re-worked and edited in response to both Reviewers' 995 comments as there seems to be some confusion from both Reviewers. The supplemental 996 section previously read as a prosaic exposition of the various models tested as part of this work. 997 It has been changed to simply give detailed information on the models used, and other models 998 tested are listed in a table. 999 1000 Tables S1 and S2 Please provide the standard error (SE) associated with each parameter 1001 estimate. Given the large number of parameters in the models, I would suspect 1002 that the SE has been very high and would invalidate your approach. 1003 1004 Response: Including the SE of the model parameters was also suggested by Reviewer 1. That 1005 has been included in these tables. 1006 1007 Yes, there are a large number of fitting parameters. The same GPP and ER models were used 1008 for all 29 collars, but the model fit parameters were determined empirically for each of the collars individually. For the GPP modeling, there was sufficient field data to justify a model with 1009 1010 5 empirical fitting parameters. For the majority of the collars, all of the GPP model fit parameters are significant to 95% confidence. This model was designed in such a way that the effect of 1011

1012 modelled parameters reduces to zero as the explanatory value of the additional variables

1013 decreases such that insignificant model parameters have a minor impact on the modelled

1014 results. For the ER modeling, the sample size is smaller and the point that Reviewer 2 is making 1015 is quite valid. There had been some debate among the authors as to which of two models to use

1015 for ER, so much so, that information was included on both of these models in the supplemental

1017 section in the original manuscript draft. Thus, the simpler ER model (with 3 fitting parameters

1018 compared to 5 fitting parameters) has been used to calculate ER in the updated manuscript.

1019
1020 Manuscript changes: Additional statistical information has been included in the table S1 and S2
1021 of the model fit parameters. The ER has been calculated by a different and simpler model in the
1022 revised manuscript, which had been previously described in the supplemental section but not
1023 used in the manuscript and was taken directly from Wilson et al., 2016. Also, the text of this
1024 supplemental section has been substantially revised to clarify confusion expressed by both
1025 Reviewers.

- 1025 1
- 1027

| 1028 | List o | f Major Changes to Manuscript:                                                                   |
|------|---------------|--------------------------------------------------------------------------------------------------|
| 1029 | 1.            | The ER was modelled using a simpler model with fewer fitting parameters based on the             |
| 1030 |               | suggestion of Reviewer 2. This resulted in minor changes in the reported NEE. However, the       |
| 1031 |               | major conclusions of the paper remained unchanged. All NEE, C balance, and GWP data was          |
| 1032 |               | updated based on this different modelling approach.                                              |
| 1033 | 2.            | The process for modelling for NEE and CH4 flux was clarified in the main body of the text        |
| 1034 |               | and in Supplemental Section 1, based on the Reviewers comments. Further statistical              |
| 1035 |               | information was included in the Supplemental material for the models used.                       |
| 1036 | 3.            | Cautionary notes were included for CH 4 data in 2016, which was estimated rather than |
| 1037 |               | measured because of equipment issues, based on suggestions from Reviewer 1.                      |
| 1038 | 4.            | The site description including site history and ecotype description was clarified in the         |
| 1039 |               | manuscript based on comments from both Reviewers.                                                |
| 1040 | 5.            | Although the main points in the discussion section remain largely unchanged, the                 |

1041discussion section was re-written to be much more concise and clear based on comments1042from both Reviewers.

**1045 **Revised Manuscript with track changes**

Title Page

Title: Carbon balance of a restored and cutover raised bog: Implications for restoration and

comparisonComparison to global trends

Running Head: C BALANCE Of A RESTORED AND CUTOVER BOG

10436. The abstract and conclusion sections were overhauled based on comments from both1044Reviewers.

Authors: Michael M. Swenson1; Shane Regan1; Dirk T. H. Bremmers1; Jenna Lawless1; Matthew Saunders2; Laurence W. Gill1

Institution:

[revised manuscript text omitted]

| -                 | Formatted:
style | Add space between paragraphs of the same    |
|-------------------|---------------------|---------------------------------------------|
| λ                 | Formatted:          | Font color: Text 1                          |
|                   | Formatted:          | Font color: Text 1                          |
|                   | Formatted:          | Font color: Text 1, English (United States) |
|                   | Formatted:          | Font color: Text 1, English (United States) |
|                   | Formatted:          | Font color: Text 1                          |
|                   | Formatted:          | Font color: Text 1                          |
|                   | Formatted:          | Font color: Text 1                          |
|                   | Formatted:          | Font color: Text 1, English (United States) |
|                   | Formatted:          | Font color: Text 1                          |
|                   | Formatted:          | Font color: Text 1, English (United States) |
| //                | Formatted:          | Font color: Text 1                          |
| //                | Formatted:          | Font color: Text 1, English (United States) |
| 1                 | Formatted:          | Font color: Text 1                          |
| Ŋ                 | Formatted:          | Font color: Text 1, English (United States) |
| 1                 | Formatted:          | Font color: Text 1                          |
| //                | Formatted:          | Font color: Text 1, English (United States) |
| λ                 | Formatted:          | Font color: Text 1                          |
| λ                 | Formatted:          | Font color: Text 1                          |
| 1                 | Formatted:          | Font color: Text 1                          |
| λ                 | Formatted:          | Font color: Text 1                          |
| Λ                 | Formatted:          | Font: Not Bold, Font color: Text 1          |
| 1                 | Formatted:          | Font color: Text 1                          |
| 1                 | Formatted:          | Font: Not Bold, Font color: Text 1          |
|                   | Formatted:          | Font color: Text 1                          |
|                   | Formatted:          | Font: Not Bold, Font color: Text 1          |
|                   | Formatted:          | Font color: Text 1                          |
|                   | Formatted:          | Font: Not Bold, Font color: Text I          |
|                   | Formatted:          | Font color: Text 1                          |
| $\langle \rangle$ | Formatted:          | Font: Not Bold, Font color: Text I          |
|                   | Formatted:          | Font Color: Text 1                          |
|                   | Formatted:          | Font color: Text 1                          |
|                   | Formatted:          | Font: Not Rold Font color: Text 1           |
| 1                 | Formatted:          | Font color: Text 1                          |
| 1                 | Formatted           | Font: Not Bold Font color: Text 1           |
| Y                 | Formatted:          | Font color: Text 1                          |
| 1                 | i ormatted:         | TOTIL COLOR. TEXT I                         |

| 1069 | water table (MAWT) and the plot scale NEE but not the global warming potential (GWP). However,                                                                |
|------|---------------------------------------------------------------------------------------------------------------------------------------------------------------|
| 1070 | a <del>). Also,</del> significant negative correlation was observed between the plot scale global warming                                                     |
| 1071 | potential and percent Sphagnum moss cover and the GWP, highlighting the importance of                                                                         |
| 1072 | regenerating this keystone genus as a climate change mitigation strategy in peatland restoration.                                                             |
| 1073 | The data from this study was then compared to the rapidly growing number of peatland $\underline{C}$ carbon                                                   |
| 1074 | balance studies across $\underline{boreal} \underline{Boreal}$ and $\underline{temperate} \overline{Temperate}$ regions. The trend in NEE and CH 4 |
| 1075 | flux with respect to MAWT was compared for the five ecotypes in this study was and literature data                                                            |
| 1076 | from degraded/restored /recovering peatlands, intact peatlands, and bare peat sites.                                                                   |
| 1077 | 1. Introduction                                                                                                                                               |
| 1078 | Peatlands are important to the global carbon cycle as they act as significant important stores of                                                             |
| 1079 | carbon (C) and sources or sinks of carbon dioxide ( $CO_2$ ) and methane ( $CH_4$ ) (Gorham 1991). Despite                                                    |
| 1080 | covering only ${\sim}3\%$ of the earth's terrestrial surface, it is estimated that between 500 and 700                                                        |
| 1081 | billion tonnes of C are stored as organic soil within the global peatland expanse (Leifeld and                                                                |
| 1082 | Menichetti, 2018; Paige and Baird, 2016; Yu et al., 2010; Paige and Baird, 2016; Leifeld and                                                                  |
| 1083 | Menichetti. 2018). However, at present, human activity is either draining or mining $\sim 10\%$ of global                                                     |
| 1084 | peatlands, transforming them from long-term C sinks into sources (Joosten, 2010; Leifeld and                                                                  |
| 1085 | Menichetti $_{a}$ 2018). In Europe, a high percentage (~46%) of the remaining peatlands are degraded                                                          |
| 1086 | to the point whereby peat is no longer actively being formed (Tanneberger et al. 2017), and in                                                                |
| 1087 | Ireland whilst $\sim$ 20% of the land area is peatland, over 95% of $\frac{14}{100}$ raised bogs has been degraded                                            |
| 1088 | through anthropogenic activities such as drainage for agriculture, forestry and peat extraction                                                               |
| 1089 | (Connolly and Holden, 20092017; Connolly and Holden, 20172009 ).                                                                                |
| 1090 |                                                                                                                                                               |

| 1091 | The Ccarbon cycle and greenhouse gas (GHG) dynamics of degraded peatlands areis often                                           |           |          |
|------|-----------------------------------------------------------------------------------------------------------------------------------------------|-----------|----------|
| 1092 | substantially different compared to intact<del>pristine</del> peatlands ( Baird et al., 2009; Blodau, 2002 <del>; Baird</del>   |           |          |
| 1093 | et al., 2009) making them significant with respect to national and global GHG budgets and emission                                            |           |          |
| 1094 | reporting ( Billet et al., 2010; Wilson et <del>al., al., 2013<del>; Billet et al. 2010)</del>. Moreover, degraded</del> |           |          |
| 1095 | peatlands can continue to emit C for decades to centuries following drainage, and current estimates                                           |           |          |
| 1096 | are that degraded peatlands store globally ${\sim}80.8$ Gt soil C and emit ${\sim}1.91$ (0.31–3.38) Gt CO $_2$ -eq.                           |           |          |
| 1097 | yr -1 (Leifeld and Menichetti-2018). Soil Cearbon sequestration through peatland restoration is                             |           |          |
| 1098 | increasingly recognized as an important strategy to tackle climate change (Dise, 2009; Leifeld and                                            |           |          |
| 1099 | Menichetti, 2018), and in recent years there has been a substantial increase in money being                                                   |           |          |
| 1100 | invested in peatland projects across the world (Anderson et al., 2017). With the increase in global                                           |
Forma | atted: F |
| 1101 | active peatland management, there is a need for more studies examining how drainage and                                                       |           |          |
| 1102 | restoration alters the eco-hydrology of degraded peatlands systems and their C earbon balances                                         |           |          |
| 1103 | (Baird et al., 2009; Young et al., 2017).                                                                                                     |           |          |
| 1104 |                                                                                                                                               |           |          |
| 1105 | The land atmosphere $CO_2$ flux, or net ecosystem exchange (NEE) in peatlands is related to water                                             |           |          |
| 1106 | table level, as inundation creates anaerobic conditions which suppresses the decomposition of soil                                            |           |          |
| 1107 | organic matter (Lain et al., 1996). This-High water table can result in a net $CO_2$ sink (negative NEE)                                      |           |          |
| 1108 | whereas a low water table can result in a net $\text{CO}_2$ source (positive NEE). Thus, water table has been                                 |           |          |
| 1109 | correlated to spatial (<del>{Strack et al., 2014;</del> J unkurst and Fielder, 2007; Silvola et al., 1996 ; Strack et           |           |          |
| 1110 | al., 2014) and temporal ( Helftler et al., 2015; Lund et al., 2012; McVeigh et <del>al.,</del> 2014; Peichl et al.,                    |           |          |
| 1111 | 2014; <del>Lund et al., 2012; </del> Strachan et al., 2016 <del>; Helftler et al., 2015</del> ) variation in the NEE of both                  |           |          |
| 1112 | intact and degraded peatlands. However, anaerobic conditions due to a high water table can also                                               |           |          |
| 1113 | increase the land atmosphere CH $_4$ flux (Frenzel and Karofeld, 2000). Both NEE and CH $_4$ flux are also                                    |           |          |
| 1    |                                                                                                                                               |           |          |

| 1114 | affected by plant ecology, as the extent of aerenchymatous vegetation cover such as Eriophorum                                             |                          |
|------|--------------------------------------------------------------------------------------------------------------------------------------------|--------------------------|
| 1115 | spp. is correlated with increased CH 4 flux (Cooper et- alal 2014; Frenzel and Karofeld, 2000 <del>, ; Gray</del> |                          |
| 1116 | et al., 2013: McNamera et al., 2008: Waddington and Day, 2007 <del>, McNamera et al. 2008, Gray et al.</del>                        |                          |
| 1117 | 2013), although this effect can possibly be reversed if aerenchymatous vegetation aerates the                                              |                          |
| 1118 | saturated soil (Fritz et al., 2011). Sphagnum spp., however, often exhibit lower CH4 fluxes (Frenzel                                       |                          |
| 1119 | and Rudolph et-al., 1998) due to a symbiotic relationship with methanotrophic bacteria                                                     |                          |
| 1120 | (Raghoebarsing et al., 2005). Also, Sphagnum spp. coverage may correspond to an increase in the                                            |                          |
| 1121 | CO 2 sink function of " natural" sites (Strack et- al. 2016) as much of the peat in northern                      |                          |
| 1122 | peatlands is derived from this genus ( Bacon et al., 2017;  Vitt et al., 2000 <del>, Bacon et al., 2017</del> ).                    |                          |
| 1123 | Furthermore, the extent of vegetation cover is an important factor affecting the NEE (Strack et al.,                                       |                          |
| 1124 | 2016; Tuitili et al., 1999; Waddington and Day, 2010 <del>; Tuitili et al., 1999, Strack et al., 2016</del> ). This is                     |                          |
| 1125 | relevant to degraded and restored peatlands because minedharvested peatlands can have large                                                |                          |
| 1126 | areas of bare peat (Wilson et al., 2015).                                                                                                  |                          |
| 1127 |                                                                                                                                            |                          |
| 1128 | Climatic variables such as the frequency of cloudiness, temperature, and length of growing season                                          |                          |
| 1129 | have also been found to be important controlling factors of NEE (Charman et al., 2013; : Helftler et                                       |                          |
| 1130 | al., 2015: McVeigh et al., 2014: Zhaojun et al., 2011 <del>; McVeigh et al., 2014; Helftler et al., 2015</del> ).                   |                          |
| 1131 | However, climate variables cannot be controlled at a specific site, and therefore, may not be as                                           |                          |
| 1132 | relevant when considering climate change mitigation actions.                                                                               |                          |
| 1133 |                                                                                                                                            |                          |
| 1134 | Although $N_2O$ emissions can be an important aspect of the GHG emissions from organic soils (Pärn                                         | Formatted: Font: Cambria |
| 1135 | et <del>.al., al.,</del> 2018), this study focuses only on aspects of the Cearbon balance. In low nutrient, non-             |                          |
| 1136 | agricultural,semi-natural sites like in this study, N2O emissions are typically low (Haddaway et al.,                                      |                          |
|      |                                                                                                                                            |                          |

| 1137 | 2014) but can be higher for deeply drained (Vanselow-Algan et al., 2015) or high nutrient sites                                                                        |
|------|------------------------------------------------------------------------------------------------------------------------------------------------------------------------|
| 1138 | (Danevčič et al., 2010). The radiative impact of different GHGs can be normalized by converting                                                                        |
| 1139 | them into a $CO_2$ equivalents in terms of the 100-year global warming potential (GWP) in tonnes                                                                       |
| 1140 | $CO_2$ -eq ha -1 yr -1 : over a hundred year horizon, $CO_2 = 1$ , $CH_4 = 34$ , and $N_2O = 298$ . (after Wilson et al.,                        |
| 1141 | 2016b from-IPCC 2013 recommendations (Myhre and Shindell, 20132014).                                                                                                   |
| 1142 |                                                                                                                                                                        |
| 1143 | Intact peatlands are a net $CO_2$ sink [typical annual average NEE range -31.9 to -66 g C- $CO_2$ m -2 yr -1 ,                                   |
| 1144 | from literature data compiled by Helftler et <del>al., (al. (</del> 2015)] and a CH 4 source . <del>[average of 9.2 g C-CH</del> 4 |
| 1145 | m -2 yr -1 , (95% Cl 0.3 to 44.5 g C-CH 4 m -2 yr -1 ) for low nutrient temperate peatlands from Wilson et      |
| 1146 | al., (2016a)]. By contrast, drained peatlands are a $CO_2$ source [the average annual NEE of +81 to                                                                    |
| 1147 | +151 g C-CO 2 m -2 yr -1 reported in Renou-Wilson et al., (al. (2018a) is typical] with very low $CH_4$                               |
| 1148 | emissions (Baird et al., 2009). However, it should be noted that this can be offset by high                                                                            |
| 1149 | $\underline{CH_4}$ methane emissions from active drains of ~60 g CH 4 m -2 yr -1 (Evans et al., 2016).                                |
| 1150 | Degraded/drained peatlands typically have a larger GWP compared to intactnatural sites or                                                                              |
| 1151 | rewetted sites because a large positive NEE outweighs the reduced $CH_4$ emissions (RenauRenou-                                                                        |
| 1152 | Wilson et al., 2018a). The NEE and $CH_4$ fluxes from restored peatlands can be similar to                                                                             |
| 1153 | intactpristine peatlands, but exhibit greater variability ( Strack et al., 2016; Wilson et al., 2016a <del>;</del>                                       |
| 1154 | Strack et al., 2016).                                                                                                                                                  |
| 1155 |                                                                                                                                                                        |
| 1156 | Several studies have suggested the hypothesis that time since restoration is an important factor in                                                                    |
| 1157 | the GWP of peatlands (Augustin & Joosten, 2007; Bain et al., 2011; Waddington and Day 2007). In                                                                        |

particular, the restored sites may go through an initial period of high  $\underline{CH_4}$  methane production and

high GWP because restored peatlands are often rapidly colonized by aerenchymatous vegetation,

1158

1159

| 1160 | such as Eriophorum spp. ( Cooper et al., 2014; Waddington and Day, 2007 <del>, Cooper et al. 2014</del> ). This             |                                   |
|------|-------------------------------------------------------------------------------------------------------------------------------------------|-----------------------------------|
| 1161 | is followed by a period of decreasing GWP as mosses and other peatland species become                                                     |                                   |
| 1162 | established (Augustin & Joosten, 2007; Bain et al., 2011). To test this hypothesis, more data is                                          |                                   |
| 1163 | needed for peatlands "restored more than 10 years previously" (Bacon et al., 2017). Also, it is                                           |                                   |
| 1164 | valuable to have studies which directly compare adjacent raised bog and cutover bogsites with                                             |                                   |
| 1165 | contrasting different site histories.                                                                                              |                                   |
| 1166 |                                                                                                                                           |                                   |
| 1167 | Aquatic losses of Cearbon include dissolved organic carbon (DOC) and dissolved inorganic carbon                                    |                                   |
| 1168 | (DIC) in runoff as well as $CO_2$ evasion from open water. These have not been measured as                                                |                                   |
| 1169 | frequently as NEE and CH $_4$ flux (Dinsmore et al., 2010), but can represent a key component of the                                      |                                   |
| 1170 | net ecosystem Ccarbon budget (NECB) ( Barry et al., 2016; Kindler et al., 2011 ). , Ignoring the                     |                                   |
| 1171 | aquatic C carbon losses would result in an overestimate of the C carbon sink function of peatlands                          |                                   |
| 1172 | (Billet et al., al., 2010). The DOC losses from temperate peatlands range from 5-36 g C m -2 yr -1 and              |                                   |
| 1173 | are lower for boreal peatlands (Range: 4-13 g C m -2 yr -1 ) (Evans et al., 2016). Few studies have                 |                                   |
| 1174 | simultaneously concurrently measured a complete NECB for a peatland including the DIC flux                                                |                                   |
| 1175 | (Nilsson et alal 2008) and CO 2 evasion from open water (Dinsmore et alal 2010), even though                     |                                   |
| 1176 | $CO_2$ evasion has been found to be important to the overall Cearbon balance <del>, with a reported 2-year</del>                   |                                   |
| 1177 | average of 12.8 g C m -2 yr -1 for an intact peatland in Scotland (Dinsmore et <del>al., al.,</del> 2010). Further, |                                   |
| 1178 | these studies have focused on intact rather than degraded or restored <del>or recovering</del> peatlands.                          |                                   |
| 1179 | Despite the considerable amount of recent scientific work on the greenhouse emissions from                                                | Formatted: Font color: Light Blue |
| 1180 | peatlands, few previous studies have quantified the carbon balance of old abandoned cutover                                               |                                   |
| 1181 | peatlands (Bacon et al., 2017). Many studies have focused on more recently abandoned/restored                                             |                                   |
| 1182 | degraded bogs. This may be because of the recent change in attitudes towards peatland                                                     |                                   |
| 1183 | conservation and restoration (Holden et al., 2004), which means that many bog restoration projects                                        |                                   |
| 1    |                                                                                                                                           |                                   |

| 1184 | are relatively recent. The focus on the cutover ecotypes investigated in this study is therefore                        |                            |
|------|-------------------------------------------------------------------------------------------------------------------------|----------------------------|
| 1185 | valuable because it the potential future climate impact of abandoned cutover bogs under the "do-                        |                            |
| 1186 | nothing" restoration approach (Holden et al., 2004). This is especially true in Ireland because                         |                            |
| 1187 | substantial areas of the Irish midland bogs are currently used in industrial peat harvesting, which is                  |                            |
| 1188 | scheduled to cease in the coming decades (Bord na Móna website).                                                        |                            |
| 1189 |                                                                                                                         |                            |
| 1190 |                                                                                                                         |                            |
| 1191 | The growing body of scientific research on the GHG and Cearbon balance of peatlands and the                      |                            |
| 1192 | importance to global climate change means that it is increasingly important to consider new data in                     |                            |
| 1193 | the context of global studies <del>. Often, boreal and temperate peatlands have similar conditions:</del>               | Formatted: Font color: Red |
| 1194 | hydrological (consistently high water table), chemical (high carbon, often acidic peat soils),                          |                            |
| 1195 | ecological (often ground cover of Sphagnum mosses, with low nutrient sedges and ericaceous                              |                            |
| 1196 | shrubs). As similar factors (i.e. water table, plant ecology, growing season length, soil temperature,                  |                            |
| 1197 | etc.) are often cited as controlling factors for greenhouse gas fluxes, it may be possible to identify                  |                            |
| 1198 | global trends across boreal and temperate peatlands (e.g. Junkurst and Fieldler 2007).                                  |                            |
| 1199 |                                                                                                                         |                            |
| 1200 | The goal of this work is to quantify all of the major aspects of the Cearbon balance (NEE, CH 4 flux, |                            |
| 1201 | and aquatic losses as DOC, DIC, and CO 2 evasion ) over a two year period for five distinct peatland  |                            |
| 1202 | ecotypes, which are located in two adjacent areas with contrasting site histories: a, for a historically                | Formatted: Font color: Red |
| 1203 | <del>(ca. 1960) abandoned peat extraction <mark>cutover bog</mark>, which was abandoned ca. 1960 compared to and</del>  | Formatted: Font color: Red |
| 1204 | an ambratraphic an adjacent mare recently rectared (2000) raised bog which was previously                               | Formatted: Font color: Red |
| 1204 | an onior ou opinc e m aujacent more recently restored (2009) raised bog , which was previously        | Formatted: Font color: Red |
| 1205 | impacted by drainage but not peat extraction, and then recently restored (in 2009), This study also                     | Formatted: Font color: Red |
|      |                                                                                                                         | Formatted: Font color: Red |
| 1206 | presents the measurements in the context of global studies on boreal and temperate peatlands with                       | Formatted: Font color: Red |

| 1207 | the aim of identifying trends in NEE and $\mathrm{CH}_4$ flux based on land condition (drained, restored,                           |
|------|-------------------------------------------------------------------------------------------------------------------------------------|
| 1208 | intactpristine), mean annual water table, and vegetation cover (presence/lack of vegetation).                                       |
| 1209 |                                                                                                                                     |
| 1210 | 2. Materials and Methods                                                                                                            |
| 1211 | 2.1 Site Description                                                                                                                |
| 1212 | Abbeyleix Bog (N 52.89714, W 7.35022, elevation approx. 90 m) is a <del>natural</del> peatland and natural                   |
| 1213 | area in Co. Laois, Ireland. This site is located in a temperate, oceanic climate with a 30 year (1981–                              |
| 1214 | 2010) mean annual rainfall of <del>844</del>923 mm and a mean annual temperature of 9.95° C (Walsh,                          |
| 1215 | 2012). Acidic, low nutrient, histosol, peat soils remain throughout the raised and cutover bog with                                 |
| 1216 | 5.0-8.5 m depth on the raised bog and 1-3 m depth on the cutover bog.                                                               |
| 1217 |                                                                                                                                     |
| 1218 | Abbeyleix Bog containing contains both areas that were historically mined for peat (referred to here                                |
| 1219 | as cutover bog) as well as un-harvested raised ombrotrophic bog <del> and</del> , which was never mined for           |
| 1220 | peat historically harvested cutover bog (Fig. 1). This site is located in a temperate, oceanic climate                              |
| 1221 | with a mean annual rainfall of 844 mm and a mean annual temperature of 9.9° C. Acidic, low                                          |
| 1222 | nutrient, histosol, peat soils remain throughout the raised and cutover bog with 5.0–8.5 m depth on                                 |
| 1223 | the raised bog and 1–3 m depth on the cutover bog. The areas of cutover bog were domestically                                       |
| 1224 | mined for peat by hand cutting between the 1870s and 1960s, and then abandoned (i.e. no                                             |
| 1225 | restoration or management works have occurred in this area post-extraction) (Ryle, 2013). Peat                                      |
| 1226 | extraction never occurred on the remaining areas of raised bog; however, these areas The areas of                                   |
| 1227 | <del>raised bog w</del> ere impacted by a surface drainage network installed in the 1980s <del>1980's</del> in |
| 1228 | preparation for industrial extraction although the plans for industrial extraction of the peat were                          |
| 1229 | later abandoned due to resistance from the local community. Throughout the raised bog, Ssurface 34                                  |